# Predictive Understanding of Socioeconomic Flood Impact in Data-Scarce Regions Based on Channel Properties and Storm Characteristics: Application in High Mountain Asia (HMA)

Mariam Khanam[1, 2], Giulia Sofia[1], Wilmalis Rodriguez[1], Efthymios I. Nikolopoulos[3], Binghao Lu[4], Dongjin Song[4] and Emmanouil N. Anagnostou[1]

[1]Civil & Environmental Engineering, University of Connecticut, Storrs, 06269, USA
[2]Water Resources Science and Engineering, Oak Ridge National Laboratory, Oak Ridge, 37771, USA
[3]Civil & Environmental Engineering, Rutgers University, Piscataway, 08854, USA
[4]Computer Science and Engineering, University of Connecticut, Storrs, 06269, USA

*Correspondence to*: Mariam Khanam (khanamm@ornl.gov), Giulia Sofia (giulia.sofia@uconn.edu)

**Abstract.** High Mountain Asia (HMA) faces heightened vulnerability to natural disasters due to its extreme conditions and the escalating impacts of climate change. Understanding the long-term response of this landscape to hydroclimatic fluctuations is imperative, given the profound effects these changes have on millions of people annually. Heavy rain and monsoon season bring forth floods and debris flows, resulting in significant damage to crops, infrastructure, and communities, causing widespread human impacts. Despite efforts to estimate flood risk locally, traditional techniques often fall short due to the scarcity of high-quality, consistent data, especially in ungauged basins. To overcome this challenge, we propose a novel approach: a geomorphologically guided machine learning (ML) method for mapping flood effects across HMA. Central to our methodology is the Lifeyear Index (LYI), a systematic measure that quantifies both the financial and human losses incurred by disasters, specifically for this study, fluvial and pluvial flooding. Our model was trained using a dataset comprising over 6000 flood events spanning from 1980 to 2020, along with their corresponding five-year and ten-year LYI. Key predictors included: (1) five-year rainfall concentrations derived from ERA5 daily data, (2) a geomorphic classifier based on hydraulic scaling functions derived from high-resolution digital elevation models (DEM), and (3) population density. Results demonstrate the model's effectiveness in identifying flood susceptibility hotspots at a national scale and delineating their evolution from 1980 to 2020. Moreover, the study underscores the severity of hydroclimatic extremes across the entire HMA region. Importantly, the proposed framework is versatile and can be adapted to generate various pluvial and fluvial flood vulnerability and risk maps in ungaged regions.

## 1 Introduction

High Mountain Asia (HMA) presents complex terrain characterized by dynamic hydrological and geomorphological processes. Over recent years, the region has been significantly affected by climate change, notably witnessing accelerated glacial melts (Shrestha and Aryal, 2011; Byers et al., 2022) and shifts in precipitation patterns and intensity (Haag et al., 2019; Kirschbaum et al., 2020). These environmental changes, compounded by anthropogenic influences such as landscape alterations, have escalated the region's susceptibility to flooding (Byers et al., 2022; Pervin et al., 2020; Shrestha et al., 2010; Zheng, G. et al., 2021), with consequent increasing threats to lives, agriculture, and critical infrastructure (Fischer et al., 2022; Pervin et al., 2020; Rentschler et al., 2022; Sharma et al., 2019; Torti, 2012). The direct impacts caused by the flood are only part of the picture; the enduring socioeconomic repercussions further compound the crisis. These include loss of livelihoods, the urgent need for rehabilitation efforts, and the psychological toll exacted on affected communities.

Flood disasters are generally associated with hydroclimatic extremes. The variability of precipitation patterns over time, space, and intensity is indeed crucial to their occurrence, but changes in catchment characteristics can also alter flood magnitude and frequency. The complex geomorphology and orographic characteristics in the HMA region cause significant spatiotemporal heterogeneity of precipitation patterns and extremes (Haag et al., 2019). Furthermore, the geomorphic structure of basins in HMA can influence the flood characteristics more than land cover does (Marston et al., 1996). Many floods in HMA carry huge amounts of sediment and water that adversely affect downstream areas where most population resides and can remain in the landscape for years afterward (Kafle et al., 2017; Simonovic et al., 2022).

Changes in river morphology and channel shifting resulting from sediment variability are recognized causes of flood risk (Blench, 1969; Criss & Shock, 2001; Lane et al., 2007; Neuhold et al., 2009; Pinter et al., 2008; Slater et al., 2015; Stover & Montgomery, 2001; Sofia & Nikolopoulos, 2020a). Several researchers have highlighted how the morphometric characteristics of watersheds provide useful insights into their hydrologic response to rainfall (Borga et al., 2008) since their morphometric characteristics are a crucial influence on flash flood intensity. In HMA, however, these control mechanisms are difficult to model at a large scale.

Accurate evaluation of the socioeconomic impacts of natural disasters is paramount to mitigate the sufferings of the affected people and rehabilitation (Cavallo & Noy, 2010; Meyer et al., 2013; Noy, 2015, 2016a). To date, available studies (Diehl et al., 2021; Mohanty & Simonovic, 2022; Pangali Sharma et al., 2019; Pervin et al., 2020; Piacentini et al., 2020; Yang & Tsai, 2000) have primarily concentrated on vulnerability mapping and risk analysis, employing case studies and descriptive event-based methodologies at a local level. Scaling up the analysis over the entire HMA region is indeed a difficult task, as it requires collecting data from several countries and multiple sources, and this poses challenges due scarcity of ground observations covering consistent timeframes homogeneously (Dollan et al., 2024; Miles et al., 2021). Especially in the context of the impact of floods using socioeconomic data, the analysis involves examining the number of fatalities, people injured, and otherwise affected, as well as the financial damage that natural disasters cause, and this information is

generally collected at the local scale based on reported events. Significant disasters are documented in global databases like The International Disaster Database (EMDAT, www.emdat.be) or, as an example for HMA and this study, the Nepal Disaster Risk Reduction Portal (http://drrportal.gov.np/). However, these databases typically operate at a global or national level resolution, potentially overlooking minor disasters. For example, EMDAT only considers events with at least one of the following criteria: 1) 10 fatalities; 2) 100 affected people; 3) a declaration of state of emergency; 4) a call for international assistance. Additionally, those databases utilized to support insurance may prioritize countries with existing or potential insurance coverage (World Bank, 2012).

The integration of geomorphic properties, population data, and rainfall characteristics for assessing socioeconomic flood impact is only recently being explored comprehensively on a large scale (e.g., Janizadeh et al., 2024). For HMA. this is primarily due to the inherent challenges associated with conducting on-site surveys in rugged and often inaccessible terrain. However, leveraging remote sensing data has emerged as a valuable approach for delving deeper into these dynamics and effectively quantifying flood impacts. Modern global datasets, featuring improved resolution and coverage, further enhance the utility of remote sensing in this regard (Diehl et al., 2021; Jongejan & Maaskant, 2015; Mosavi et al., 2018; Bentivoglio et al., 2022; Mazzoleni et al., 2022; Hawker et al., 2018; Kirschbaum et al., 2020; Mohanty and Simonovic, 2022; Pangali Sharma et al., 2019; Sanyal and Lu, 2004; Yang and Tsai, 2000; Zheng et al., 2018).

Furthermore, machine learning (ML) techniques have emerged as increasingly popular tools in advanced prediction systems over the past two decades. They offer more cost-effective solutions with performance that can be aggregated, surpassing the complexity and time demands associated with simulating the complex development of flood processes. Recent research (Bentivoglio et al., 2022; Deroliya et al., 2022; Mosavi et al., 2018) has showcased encouraging advancements by integrating machine learning (ML) techniques with global datasets. This contemporary approach to mapping flood vulnerability notably streamlines the computational processes associated with data-intensive simulations, enhancing flood risk management strategies. However, ML systems rely on existing data for learning. Insufficient or incomplete data coverage can hinder effective learning, leading to suboptimal performance when deployed in real-world scenarios. Therefore, ensuring robust data enrichment, encompassing both quantity and quality, is imperative.

In this study, we introduce a streamlined methodology for preliminary flood vulnerability assessment on a large scale, leveraging available global datasets. Specifically, we introduce a flood-risk assessment model designed to quantify spatially distributed socioeconomic susceptibility in flood-prone regions. We utilize this model to augment disaster understanding by integrating remotely sensed data, including climate variables and high-resolution terrain information. Finally, we apply this model in the High Mountain Asia (HMA) regions to analyze changes in socioeconomic flood impacts spanning from 1980 to 2020.

## 2 Materials and Methods

### 2.1 Study Area

HMA, otherwise known as the Hindu Kush-Himalayan region, comprises Nepal, Pakistan, Bangladesh, Bhutan, India, Afghanistan, Kazakhstan, Kyrgyzstan, Tajikistan, Uzbekistan, Mongolia, China, and part of many other countries in Asia. HMA is home to some of the world's highest mountain systems, including the Himalayas and the Hindu Kush. This rugged terrain has a highly variable climate ranging from tropical to subpolar, essentially controlled by altitude. Around 1.5 billion people (https://nsidc.org/data/highmountainasia) dwelling in the region are at risk of natural disasters (such as heavy rainfall, flooding (pluvial/ fluvial/ flash), earthquakes, avalanches, and landslides) due to the topographic characteristics, changing climate patterns, and high population density. Some of the world's largest rivers and deltas, such as the Indus and the Ganges, are located in this region. In the summertime (June to September), monsoon rains bring a vast amount of water (Kayastha & Kayastha, 2019) to the rivers and valleys in the southern part of HMA (Northern India, Nepal, Bangladesh, and Pakistan). Kirschbaum et al. (2020) have projected that the greatest increase in very high intensities of precipitation (>20 mm/day) will occur during the monsoon season, with the enormous amount of rain causing all types of devastating floods (Talchabhadel et al., 2018). Referring to the data reported, for example, in EMDAT, among all other hydroclimatic disasters in HMA from 1980 to 2020, floods affected the most people (53% among all other hydroclimatic disasters) and caused the highest total damage (56% among all other hydroclimatic disasters). Bangladesh, Nepal, Pakistan, and parts of India were hotspots with the highest casualties (source: EMDAT).

This study considers approximately 6,000 watersheds across HMA as the main target area (Figure 1): the watersheds were selected to be consistent with the HMA domain and all the datasets produced throughout the different phases of the NASA-funded HiMAT project (https://himat.org/). The analysis initially centred on training and testing a machine-learning model specifically for Nepal. To achieve this, we collected fine-resolution topographic data along with district-scale socioeconomic information on population characteristics and documented flood impacts for this region. Subsequently, leveraging the insights gained from this initial phase, we extended the application of the trained model to predict socioeconomic impacts across all watersheds in HMA.

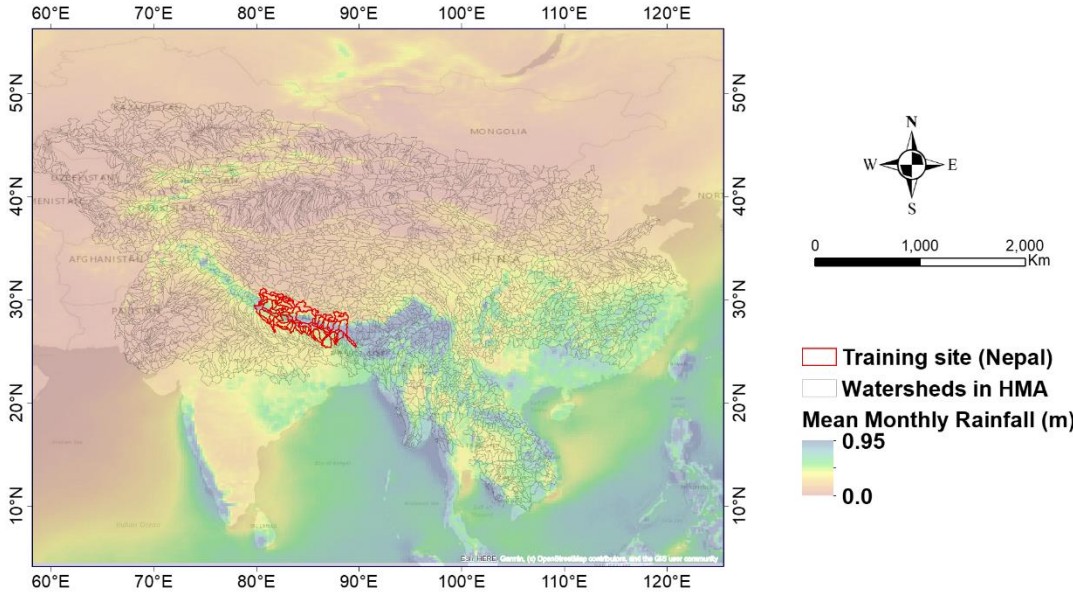

Figure 1: Study area- watersheds across High Mountain Asia (HMA), with highlighted the training domain (Nepal) and the overall rainfall variability across the region. The watershed displayed in black represents the 6000 watersheds that were used in the study. The watersheds were selected to be consistent with the HMA domain and all the datasets produced throughout the different phases of the NASA-funded HiMAT project (https://himat.org/)

## 2.2 Methods

Figure 2 illustrates the conceptual framework guiding this study. We employed machine learning (ML) analysis, utilizing climatic and geomorphologic variables, to forecast the socioeconomic impact of extreme fluvial and pluvial flood events spanning from 1980 to 2020 across High Mountain Asia (HMA). To capture the link between flooding and climatic and geomorphologic processes, the model considers as predictors a climatic index derived from ERA5 rainfall, and a geomorphological index, the Flood Geomorphic Potential -FGP- that characterizes the flood-proneness of the landscape, together with population data. A notable advantage of the proposed approach lies in its reliance on automatic techniques leveraging globally available datasets, thereby facilitating its applicability across diverse geographical regions to forecast socioeconomic flood impacts. The framework also benefits from leveraging geomorphologically driven information to have an improved characterization of the different aspects of the underlying physical processes shaping the landscape and possibly impacting flood characteristics. By incorporating such domain knowledge into the ML model, the framework can better generalize across different regions and conditions, improving robustness and reliability for risk mapping in diverse environments and facilitating informed decision-making for flood management and mitigation strategies.

To represent exposure and socioeconomic impacts, we introduced, respectively, a variable for population and "Lifeyears Index" (LYI) (Noy, 2014, 2016a, 2016b), a unit of measurement used to describe a disaster's impact in terms of

the total years of life lost (see section 2.3.1 for details). To predict the LYI, we applied XGBoosting (eXtreme Gradient Boosting) (Chen et al., 2018; Chen & Guestrin, 2016). The predictor and response variables of the ML framework are described in the subsections below.

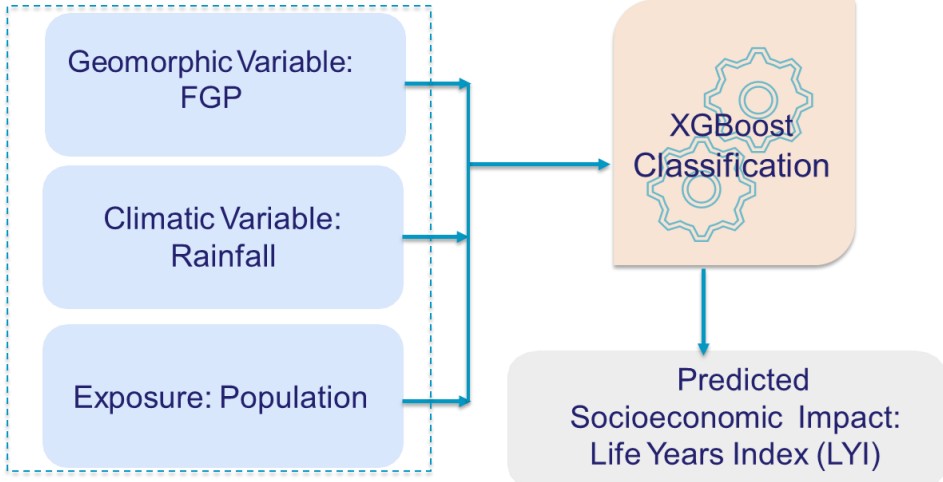

**Figure 2: Conceptual framework. Considered predictors are Flood Geomorphic Potential (FGP), Rainfall, and Population. The predicted value is the socioeconomic impact, characterized as the Lifeyears Index (LYI) (Noy, 2016a; Noy, 2015). Readers should refer to the following sections for an explanation of the predictors and predicted values.**

The analysis follows a multistep approach, beginning with data at both watershed and district scales. Initially, the focus was on the district scale, as socioeconomic data for Nepal, selected as the primary training ground, were readily available at this level through the Nepal Disaster Risk Reduction Portal (http://drrportal.gov.np/). For this region, furthermore, there is a comprehensive coverage of high-resolution (8-meter) Digital Elevation Models (DEMs) from prior High Mountain Asia (HMA) work (High Mountain Asia 8-meter DEMs Derived from Along-track Optical Imagery, 10.5067/0MCWJJH5ABYO). Subsequently, all the information is aggregated at the watershed scale, as phenomena such as fluvial and pluvial flooding occur at this level, necessitating a dataset tailored to this scale.

To transfer the demographic information from the district to the watershed scale, we performed a weighted spatial join between the watersheds and districts. For each watershed, we attributed the statistical characteristics of the intersecting districts, with weights based on the overlapping areas. The aggregation from district to watershed is done by a weighted average, considering the extent of the district area within the watershed as a weight. Generally, the districts in Nepal are smaller in extent compared to the various watersheds.

## 2.3 Datasets

### 2.3.1. Socioeconomic Flood Impacts

The research focused on predicting the socioeconomic impact of floods. Measured economic loss and tangible damages were analyzed by considering the Lifeyears Index (LYI) (Noy, 2014, 2016a, 2016b). This index is presented by

Noy (2016) as "Lifeyears lost" and it is a variant of the WHO Disability Adjusted Life Years (DALYs) lost due to diseases
and injuries (WHO, 2014). We calculated LYI for Nepal by using damage statistics and demographic information collected
from different data portals in Nepal.

The Index is described by Equation 1 and the parameters used in the equation are described in Table 1:

$$LYI = M(Aexp - Amed) + e * T * N + (1 - c)Y/PCGDP \qquad (1)$$

**Table 1: Parameters used to calculate LYI**

| Variable | Description | References |
|---|---|---|
| M | Mortality (number of deaths due to disaster | Nepal Disaster Risk Reduction Portal (http://drrportal.gov.np/) |
| Aexp | Average life expectancy at birth (by year) | WHO (https://data.who.int/countries/524) |
| Amed | Median age (by year) | WHO (https://data.who.int/countries/524) |
| e | Welfare reduction weight associated with being exposed to a disaster | set to e = 0.054 according to Noy, (2016a), based on Mathers et al., 2013 |
| T | Time taken by the affected person to get back to normal | Noy, (2016a) |
| N | Number of affected people | Nepal Disaster Risk Reduction Portal (http://drrportal.gov.np/) |
| c | Percent of time not used in work-related activities (.75) | Noy, (2016a) |
| Y | Financial damage (value of destroyed/damaged infrastructure) | Nepal Disaster Risk Reduction Portal (http://drrportal.gov.np/) |
| PCGDP | Income per capita (by year) | The World Bank (https://data.worldbank.org/country/Nepal) |


In this study, we classified Lifeyears Index (LYI) values into three distinct categories: Low for cases where
log(LYI) < 2; Medium for values falling between 2 and 3; and High for log(LYI) > 3. This classification scheme indicates
that a watershed or district is deemed to be at high risk if the average LYI exceeds 1000 years, while medium risk spans LYI
values ranging from 100 to 1000 years, and Low risk encompasses LYI values less than 100 years. For instance, if the
calculated LYI is 100 years, it implies that the estimated impact of the given disaster equates to a potential loss of 100 years
of life per 100,000 people.
The cumulative LYI for Nepal (Figure 3) can provide an idea of how the cumulated flood impact has been
increasing in a country with time. It also highlights how the index itself captures major disasters, such as those occurring in
1981 (ICIMOD, 2011; Kiran S et al., 2008), 1993 (Nepal - Floods and Landslides, 1993), in 1996 (Nepal - Floods Situation
Report No. 1, 26 July 1996), and in the monsoon seasons in 2003 and 2014 (Nepal Annual Report, 2003.; Nepal: Landslides
and Floods - Aug 2014). The most changes can be noticed in the LYI for the years 1981, 1993, and 2014, the cumulative
step change for these years from the previous year are subsequently 9999, 82865, and 976238 years.

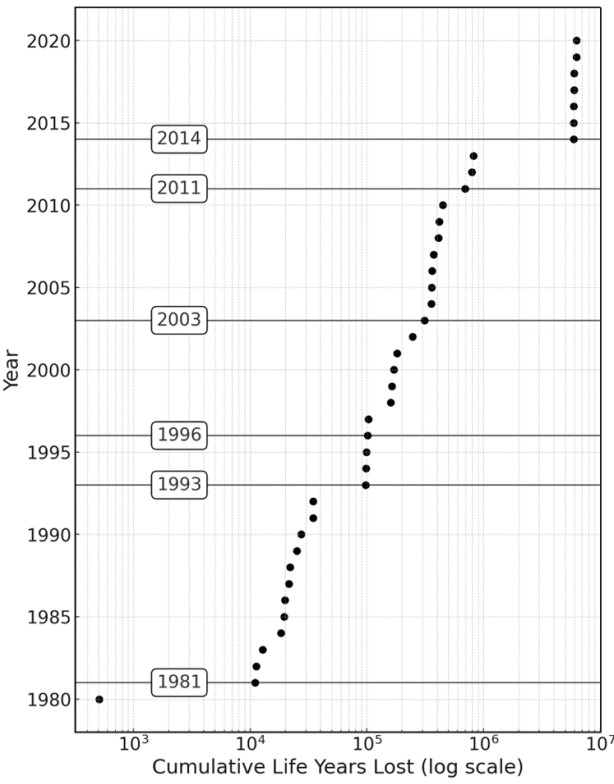


**Figure 3: Cumulative lifeyears lost over the years in Nepal. Highlighted years represent jumps in the cumulative value, mostly**
**related to well-known disasters): 1981 (ICIMOD, 2011; Kiran S et al., 2008), 1993 (Nepal - Floods and Landslides, 1993), 1996**
**(Nepal - Floods Situation Report No. 1, 26 July 1996), and in the monsoon seasons in 2003 and 2014 (Nepal Annual Report, 2003.;**
**Nepal: Landslides and Floods - Aug 2014).**
**2.3.2. Floodplain Mapping**
The identification of areas with the potential to be inundated is fundamental to preserving and protecting human
lives and property while safely supporting economic activities. Hence, we applied a large-scale floodplain delineation
algorithm to identify such areas at the basin scale across the HMA. Many researchers (e.g., Dingle et al., 2020; Lindersson et
al., 2021; Piacentini et al., 2020; Sofia, 2020b) have used DEM-derived geomorphic indices as a high-resolution flood
mapping tool. We opted for considering a variation of the Samela et al. (2017), which is a modified Geomorphic Flood Index
(GFI) by Sofia et al., 2017 & Sofia et al., 2015, thereby described as Flood Geomorphic Potential (FGP).

FGP = ln (h$_r$/ H)                                                           (2)

The index is calculated as the logarithm function of the bankfull elevations, H (estimated using a hydraulic scaling

function, or HSF (w=αA$^β$), based on bankfull width (w) and contributing area (A)) in the element of the river network closest
to the point under examination and the elevation difference between these two points, h$_r$ (Figure 4, Equation 2). The index
was improved over a main aspect: the automatic identification of the HSF directly from terrain data, applying the technique
of Sofia et al., 2017 & Sofia et al., 2015 to retrieve the bankfull location automatically through the landscape. This has the
advantage of allowing for full automation of the mapping starting purely from terrain data.

For this analysis, we trained the model considering FGP derived from the unique 8-meter Digital Elevation Models

(DEMs) for Nepal that are available at the NASA National Snow and Ice Data Center Distributed Active Archive Center
(NSIDC DAAC) (Shean, 2017). While Nepal is entirely covered by the 8m DEM, extending the model to the whole HMA
region is complicated by the gaps in the input satellite strip resulting from limited coverage, clouds, or failed stereo
correlation. For this reason, we also considered the 30m DEM by Copernicus (European Space Agency, Sinergise.
Copernicus Global Digital Elevation Model, 2021), a digital surface model (DSM) that represents the surface of the Earth,
including buildings, infrastructure, and vegetation. Importantly, this DSM is derived from World DEM, an edited DSM in
which the flattening of water bodies and the consistent flow of rivers have been included. Shore and coastlines, special
features such as airports, and implausible terrain structures have also been edited.

We identified flood-prone areas by grouping them into six classes by their FGP index. For each watershed, we then

considered the areas covered by the classes with FGP greater than 4, which, when compared to published data, proved to
correspond realistically with areas subject to floods of about 100-year depth. Figure 4b compares the Flood Geomorphic
Potential (FGP) automatic classes derived for select rivers in Nepal, with baseline inundation scenarios evaluated using
standard inundation depths associated with critical flood events and their return periods provided in the work of Delalay et
al. (2018). This visual comparison serves to highlight the efficacy of flood inundation mapping facilitated by the FGP. The
HAND (Height Above Nearest Drainage) in Delalay et al. (2018) is a widely used approach for estimating flood inundation
extents and water depths. It operates on the principle of deriving relative elevations from a DEM, similar to our approach,
which also relies on DEM-based analysis. While having assumptions may introduce some limitations in accurately capturing
complex flood dynamics, HAND remains a useful and practical method for large-scale flood assessment due to its
computational efficiency and compatibility with readily available topographic data. Given these similarities, we find it
reasonable to include HAND as a comparative reference in our study while acknowledging its limitations.

It's worth noting that the DEM-derived geomorphic index has been previously published and applied in various

contexts (Samela et al., 2017). While testing the quality of the DEM-derived geomorphic index lies beyond the scope of this
work, its effectiveness for flood mapping has been well-established in previous studies (e.g. Manfreda & Samela, 2019),
which have demonstrated the utility of the methodology, particularly in ungauged conditions, for preliminary identification
of flooded areas in regions where conducting expensive and time-consuming hydrologic-hydraulic simulations may not be
feasible.

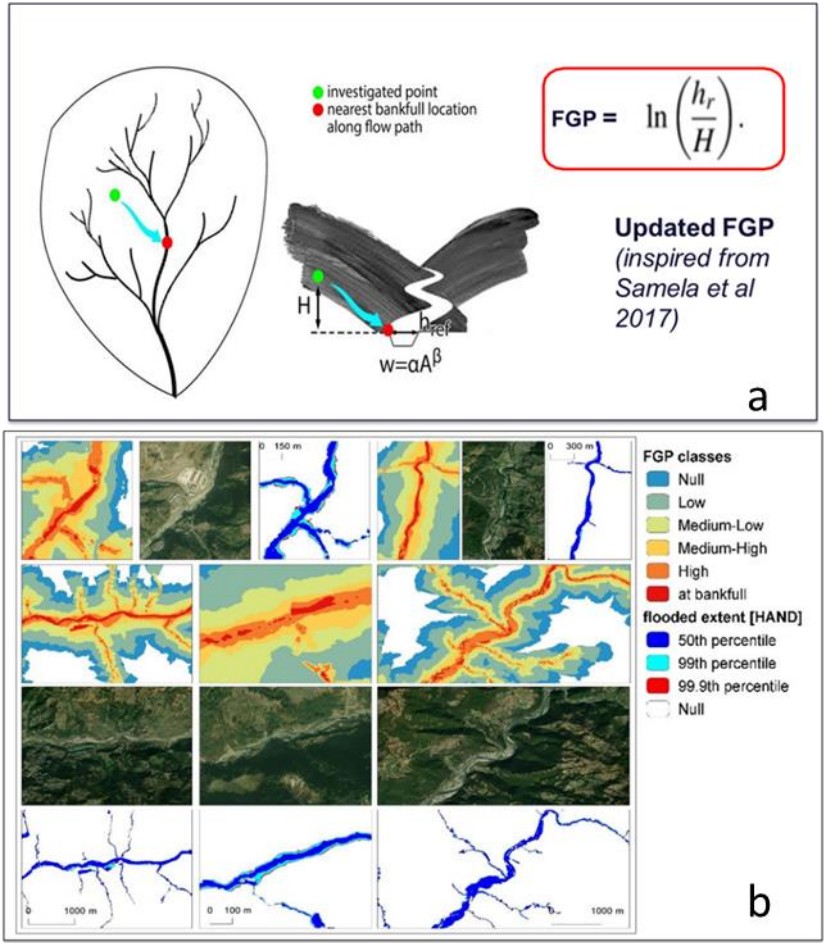

**Figure 4: a. Flood Geomorphic Potential (FGP) (modified from Samela et al. 2017); b. FGP automatic classes compared to baseline inundation depth (HAND) scenarios and Orthophotos of selected areas (aerial imagery © Google Earth 2015).**

### 2.3.3. Rainfall Characteristic

The climatology in HMA is highly variable (Dollan et al. 2024). Summer monsoons drive precipitation in the
Ganges-Brahmaputra basins and the Tibetan Plateau (Bookhagen and Burbank, 2010; Shamsudduha and Panda, 2019);
synoptic storms dominate winter precipitation, impacting areas in the northwestern Karakorum mountains (Winiger et al.,
2005; Barlow et al., 2005). Overall, as well, variations in elevation gradients contribute to diverse microclimates,
exemplified by Nepal's swift transition from high mountains to lowlands (Kansakar et al., 2004; Karki et al., 2016). Winter
precipitation in the area is primarily influenced by the westerly weather system, with western disturbances originating in the
Mid-Atlantic or Mediterranean Sea and traversing through northwest India to western Nepal after passing over Afghanistan

and Pakistan (Kansakar et al., 2004; Hamal et al., 2020). In Nepal, which was used as the training site for the model, regional climate variations exist, mostly driven by changes in elevation, with an overall homogeneity in trends (aside from a few hotspots) and regional statistics of precipitation, in line with the variability of HMA, as highlighted by the recent study by (Khanal et al., 2023).

For this work, for the main rainfall driver of the model, we focused on daily climate concentration. As climate concentration values are mostly related to the temporal variability of the rainfall, not to the total amount or the average yearly and seasonal statistics, using this index allows us to capture various climates globally (Monjo and Martin-Vide, 2016). The variability of climate concentration, furthermore, has been proven to be highly linked to pluvial/fluvial flooding impacts in various regions of the world, including for example Italy (both in mountainous landscapes and floodplains (Sofia et al., 2019), the US (Saki et al., 2023) [over a variety of physiographic regions], or China (Du et al., 2023). Different authors have adopted different methods to determine the temporal concentration of precipitation, and the Concentration Index (CI) (Equation 2) is one of the most used parameters (Caloiero et al., 2019; Martin-Vide, 2004; Monjo, 2016; Sangüesa et al., 2018; Serrano-Notivoli et al., 2018).

$$CI = \frac{S}{S+A} \tag{2}$$

This index was proposed by Martin-Vide (2004) originally to explore the contribution of the days with major rainfall to the total amount within a certain time range. The benefit of this index is that it can describe the temporal variability of rainfall at daily, annual, and seasonal scales using a single metric, as well as spatial variability at pixel or watershed scale. In the present study, we computed CI (Martin-Vide, 2004) using the ERA5 hourly rainfall data from 1980 to 2019. The source of rainfall data was selected as various works for HMA highlighted its effectiveness in capturing extreme events quite accurately compared to other products (Maggioni & Massari, 2018; Maina et al., 2023; Dollan et al., 2024). The CI was calculated considering a 5-year window. The choice of this length was made to have sufficient data to calculate the index, as well as to be able to capture variability over the 49 years of this analysis.

We identified storm events from this dataset primarily based on the criterion of rainfall of more than 0.5 mm, and we separated events when rainfall was below this threshold for more than 12 hours. Furthermore, we calculated CI using the cumulative amount of rainfall and the cumulative frequency of the event duration (Figure 5) for the selected events. The method (similar to Cortesi et al., 2012 and Monjo & Martin-Vide, 2016) eventually aggregates the amount of precipitation that falls during each event into increasing categories and determines the relative contribution (as a percentage) of the progressively accumulated precipitation, as a function of the accumulated percentage of the durations of the events. The concentration index is then calculated as the ratio of the area between the line of equality (y=x) and the fitted curve (S), and the total area under the line of equality (A+S) (Figure 5, equation 2). The index is defined by the relationship between the accumulated percentage of time, and the accumulated rainfall.

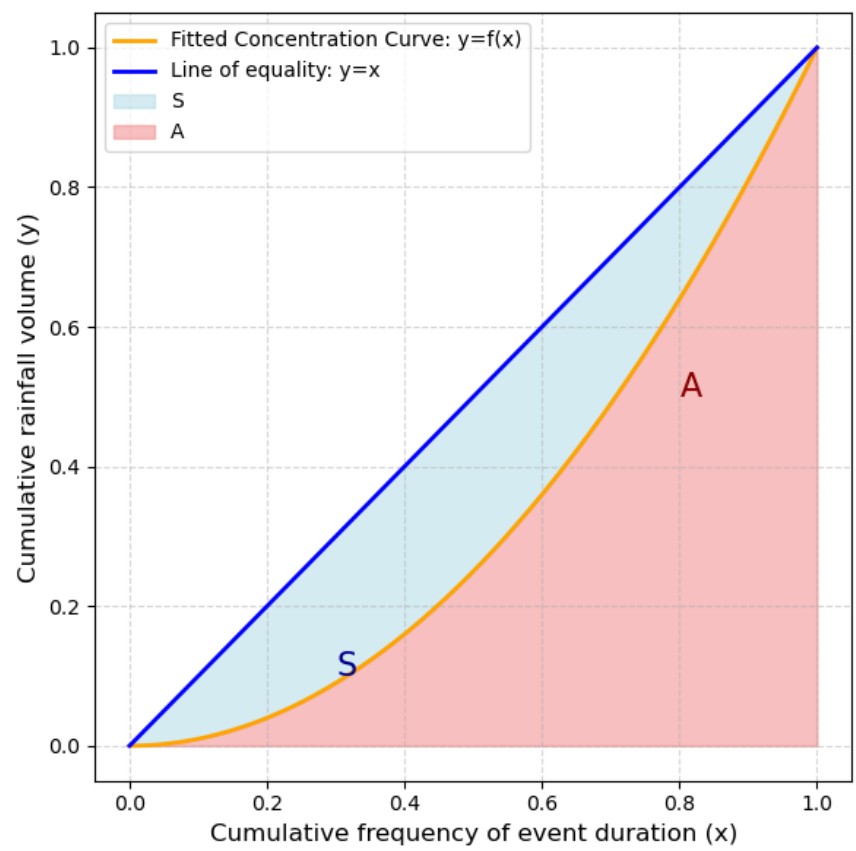

**Figure 5: Example of the line of equality and empirical curve for the rainfall concentration calculation. The concentration index is equal to the area between the line of equality and the fitted curve (S) divided by the total area below the line of equality (S+A)**

### 2.3.4. Exposure (Population)

As all the parameters of the LYI are not always readily available at the watershed scale (as highlighted by most published literature, which considered LYI at the country scale), we added population counts as one of the predictors to train the model. For Nepal, we selected the data from the country's national census (https://censusnepal.cbs.gov.np/Home/Index/EN) and aggregated it at the watershed scale by using the previously mentioned weighted join. To extend the model to the whole HMA, we computed the population for each watershed across the region from the Gridded Population of the World (GPW), v4 | SEDAC, 2024). This dataset provides spatially explicit estimates of population density for the years 2000, 2005, 2010, 2015, and 2020, based on counts consistent with national censuses and population registers, as raster data to facilitate data integration. We used a simple linear regression to retrieve data for the missing years.

### 2.4. Machine Learning Model

XGBoosting is primarily used to solve classification problems. To generate the results, the XGBoost algorithm uses an ensemble of boosted trees. An ensemble is a collection of predictors that together can give a final prediction while

reducing errors significantly. In this case, the predictors were climatic variables, geomorphologic variables, and exposure. Boosted algorithms are those in which each successive model attempts to correct the errors of its predecessor (similar to adaptive learning). The basic XGBoost algorithm can be understood as an ensemble of boosted trees. The idea behind such an ensemble is that multiple trees are built in sequence, each tree built on the previous one's prediction. And each successive tree built considers the errors of the previous trees. This means that when we take an average of all the trees at the end, we get a final tree that is better than any individual tree within the model. We applied the XGBoosting model to the geomorphologic, climatic, and exposure variables to predict classes of LYI in different basins in Nepal and HMA.

### 2.4.1 Validation of the System at the HMA Scale

We conducted thorough testing and validation of our model for Nepal, comparing the predicted value of LYI to the calculated Lifeyears Index (LYI) data from tabular values specific to the region. We trained the model and validated it only using the data for Nepal, at the district scale and then at the watershed scale. Overall, we opted for a 90-10 approach, for which 90% of the Nepal data were used for training and 10% for validation. Upon extending the model's applicability to the entire High Mountain Asia (HMA) region, we rigorously assessed the quality of our results by comparing the predicted social impact with that reported in established flood databases covering the region. We performed a hyperparameter tuning using weighted accuracy (1-3-9 weighting scheme) for subsequently (low, med, and high classes), prioritizing category "high". Initially, when XGBoost was trained, it achieved a 63% test accuracy, but its confusion matrix revealed that it struggled to correctly classify the most destructive category (category 3). Since this category was of primary interest, the model was refined using weighted accuracy, emphasizing its importance. A 5-fold cross-validation with 1000 iterations was conducted, and for each cross-validation, oversampling was applied to balance the dataset.

To verify our findings, we compared the predictions at the HMA level with flood events reported in the Dartmouth Flood Observatory's (DFO) Global Active Archive of Large Flood Events, 1985–Present. This comprehensive database compiles information on major floods sourced from diverse channels such as news reports, governmental records, ground observations, and remote sensing data. Notably, the DFO dataset encompasses various flood types, including lowland floods and mountainous river floods characterized as fluvial and pluvial floods. The dataset provides point locations, representing the centroids of affected areas during floods. While acknowledging that flood centroids may oversimplify the complexities driving flood events, we utilized this dataset to showcase our model's capability to target high-risk locations historically impacted by floods within the specified timeframe. Identifying high-risk areas with recorded flood occurrences centered around these locations underscores the robustness of the model beyond the confines of its training and validation site in Nepal.

Meteorological and climatological severity reported in the DFO database does not directly reflect the social impact of floods, and the events listed often span multiple watersheds. To address these limitations, we compared our model's predictions to the DFO data using a proxy for social severity—specifically, the reported number of people affected, including "Deaths" and "Displaced." Instead of relying on meteorological classifications, we grouped the DFO events by

social severity classes defined as $10^n$, where n corresponds to the severity level indicated in the DFO database. We then
evaluated the marginal probability that events with varying DFO severity occurred in watersheds with different predicted
LYI (Local Yield Impact) levels. Additionally, we computed the conditional probability—that is, the likelihood of a DFO-
classified event occurring within watersheds predicted by our model to have a certain LYI classification. This conditional
probability helps assess how well our system identifies high-impact regions across different time frames. For example, if
only 10% of watersheds are classified as high impact by our model, but most of the DFO's most severe events (e.g., those
with >1000 people affected) occurred within these watersheds, this would indicate that our model effectively captures
regions of elevated social risk. A more detailed discussion of model performance and validation is provided in Section 3.3.
Results Analysis

## 3.1. Variability of the Predictors

The topographical characteristics of an area can influence the local climate and population distribution. Figure 6
shows an example of how climate concentration and population vary in Nepal, as compared to watersheds that have areas of
high FGP to a greater or lesser extent. The figure reports the average for the time frame 1980–2020 for CI and population,
while the FGP is a static value for the time frame (since it is based on a unique DEM dataset), and it represents the overall
geomorphic characteristics of Nepal.
From this analysis, we can see how the variability of CI is complex. If expectedly, the variability of the index is
related to atmospheric characteristics (Sangüesa et al., 2018), the index also varies due to geographical factors influencing
climate (Tuladhar et al., 2020). In their study based on Nepal, Karki et al. (2017) highlighted the difference in the spatial
pattern of high-intensity storm events from that of annual and monsoon events. The rapid rate at which physical processes
(e.g., convection) take place regulates the high temporal concentration of precipitation in the regions where the sea surface
and ground are highly affected by warmer temperatures (Monjo & Martin-Vide, 2016). On the other hand, the low temporal
concentration of rainfall is characterized as a normal pattern caused by cyclical weather events (Monjo & Martin-Vide,
2016). Watersheds with lesser floodplain extents (that is, less areas with high FGP) are related to higher and steeper
mountains, with complex orography. Research has shown that low areas in Nepal are susceptible to receiving high-intensity
storm events even though they have fewer wet days (Karki et al., 2017). The authors of the same study also observed that the
low-intensity events (annual and monsoonal precipitation) were mostly predominant over Nepal's western middle mountains
and central high mountains. In another study, however, Subba et al. (2019) stated that the frequency of extreme events had
decreased significantly over the past two decades in the eastern part of Nepal. For our case, areas having the larger physical
potential to flood (high FGP), appear to be areas showing the largest variation in CI, with values ranging from low (0.2) as
well as very high (0.75), indicating a potential compound effect of highly torrential rains (CI=0.7) in locations where much
of the landscape is potentially floodable (FGP high) and most population reside.  Readers should consider that higher FGP

values do not imply locations having wider channels, but rather they indicate how the landscape is potentially more flood prone as highlighted by (Samela et al., 2017; Manfreda & Samela, 2019).

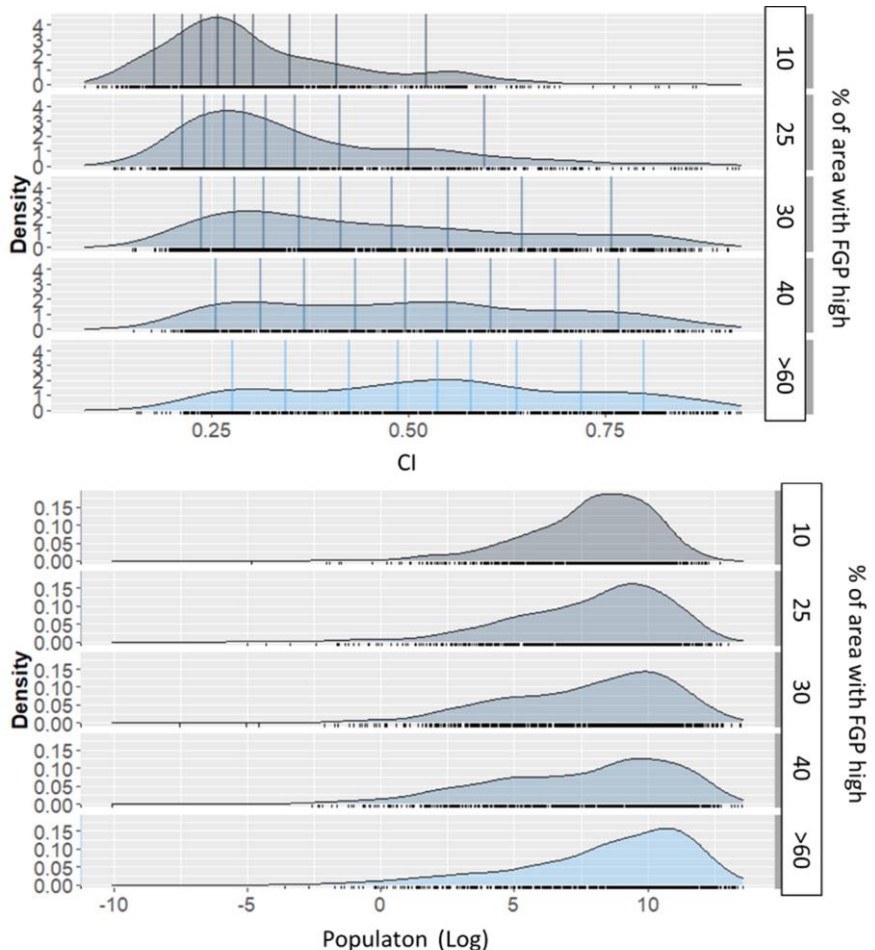

**Figure 6: Average variability of the CI (top) and population (bottom) compared to FGP from 1980-2020**

Much of the population of Nepal tends to be concentrated in areas with higher FGP, as is typical for mountainous areas, where population and economic activities are mostly located in the river valleys. Globally, the floodplains of rivers are preferred living spaces for the population and provide favourable locations for economic development. These areas are commonly exposed to floods, however, an increasing population, together with the changes in storminess, means that the risks from flooding are expected to be higher. On average, the population increased significantly in watersheds that transitioned from low to medium (LtoM), medium to high (MtoH), or low to high (LtoH) flood risk categories (Figure 7: example variability from 1985 to 2020). This suggests that growing population density in certain watersheds may be contributing to increasing flood susceptibility. The CI (climate concentration index) slightly decreased over this period for

some watersheds. However, watersheds experiencing population growth were more likely to influence the transition to a higher flood risk category. Although CI has not significantly increased, the interaction between land-use change, urban expansion, and demographic shifts may be playing a role in driving these transitions. Transitioning watersheds have a higher average FGP compared to the overall average FGP and tend to have a larger average watershed area compared to all watersheds. This indicates that larger watersheds are more prone to experiencing shifts in FGP and flood risk categories, possibly due to their ability to accumulate and distribute larger volumes of runoff and sediment. This supports the idea that intrinsic watershed characteristics (such as geomorphology and size) play a role in flood susceptibility alongside external factors like population growth and rainfall concentration index (CI). Area successfully predicted as at high risk (high LYI) in the most recent years, are areas showing high social vulnerability in terms of favorable Social Conditions (lack of communication, access to electricity and infrastructures, lower education, small children under 5); high percentage of migrating community and high risk of poverty and poor infrastructures (Aksha et al., 2019).

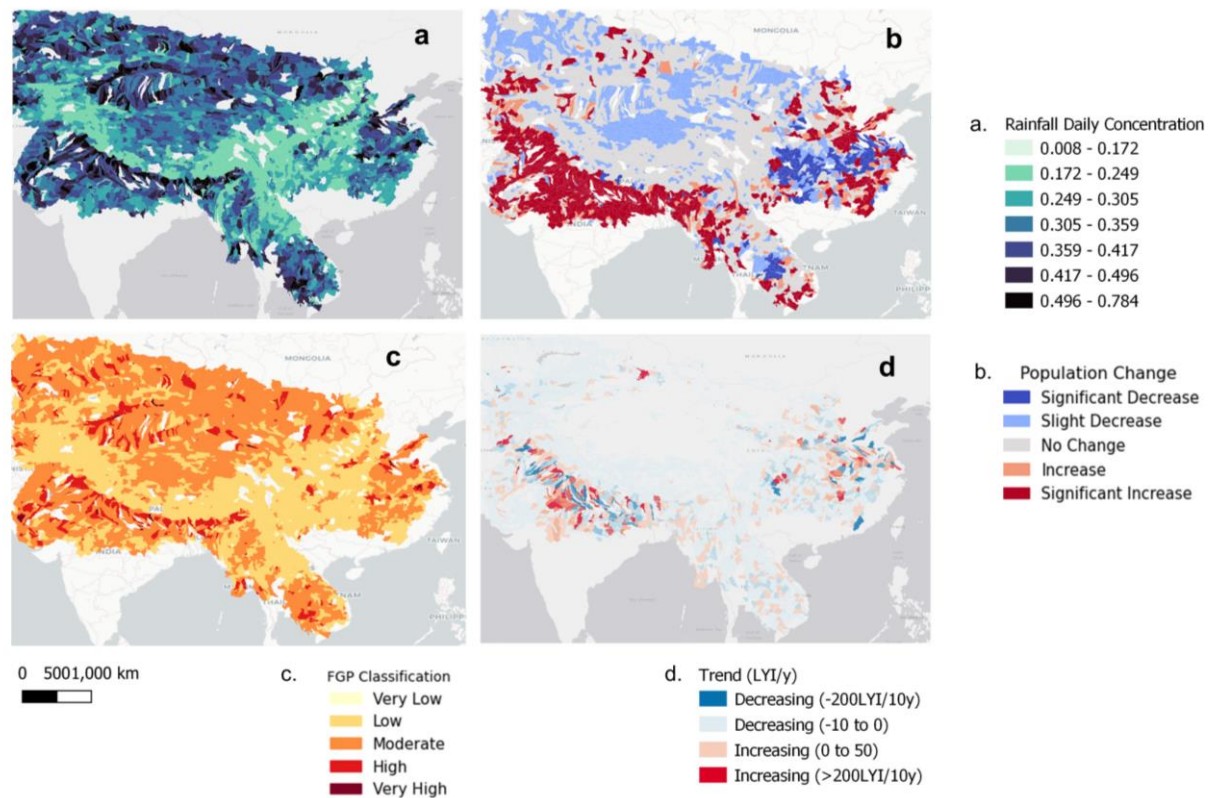

**Figure 7: Average variability of the rainfall CI (a), population change (b), compared to FGP (c) and LYI Trend (d) from 1980-2020**

## 3.2. Variable Importance and Model Performance

In this section, we present a variable importance comparison (Figure 8) based on the Feature Importance Score (F-score) in XGBoost. XGBoost provides an F-score based on how frequently a feature is used in splitting the data across all decision trees. This is the number of times a feature appears in a split across all trees in the model. A higher value indicates that the feature was used more frequently in decision-making, suggesting it has a stronger influence on model predictions. The F-score indicated that population (Pop) was the most important variable, which was consistent with our expectation in the sense that the socioeconomic impact depends largely on the exposure. The climate variable (CI) happened to be the next important variable, showing the significance of the region's climate on the socioeconomic impact of flood occurrences.

The precision, recall, and F1 score are metrics used to evaluate the performance of a classification model. Precision is the fraction of true positives among the predicted positives. Recall is the fraction of true positives among the actual positives. The F1 score is the harmonic mean of precision and recall.

The evaluation metrics reveal in Table 2 that the model performs best in the High class, with the highest precision, recall, and F1 score. The final tuned models achieved weighted accuracies between 52% and 58%, but significantly improved recall (71%), precision (73%), and F1-score (72%) for category "high". This means that out of 34 actual instances of the highest category, 24 were correctly predicted, and out of 33 predicted cases, 24 were accurate, confirming that the model effectively focused on the most critical category. This suggests that while the overall accuracy slightly decreased due to the re-weighting, the model's performance in identifying the most critical cases significantly improved. The Medium class also demonstrates relatively high performance across these metrics. However, the Low class exhibits the lowest performance, suggesting that the model may face challenges in accurately distinguishing between the Low and Medium classes or may demonstrate a bias toward predicting the Medium and High classes. These findings provide valuable insights into the strengths and limitations of the classification model and can guide future efforts to improve its performance. Overall, considering that the model aims to target substantial risk areas, a higher rate of predicting impacts is acceptable, compared to an underestimation of the risk.

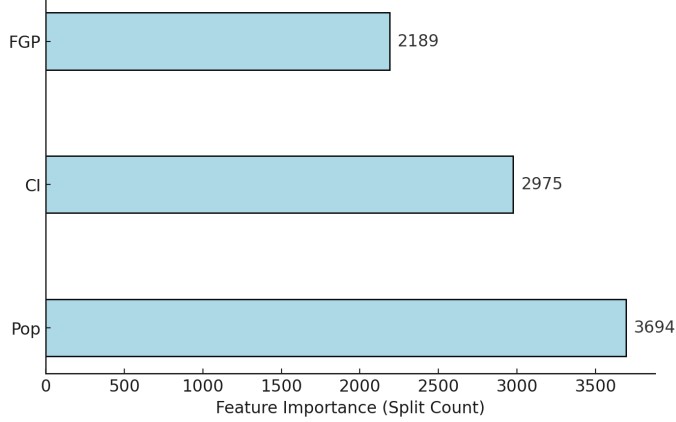

**Figure 8: Feature importance score (F-score)**

**Table 2: Performance metrics of the model on the test dataset**

|  | precision | recall | f1-score |
|---|---|---|---|
| **Low** | 0.54 | 0.57 | 0.56 |
| **Medium** | 0.64 | 0.63 | 0.64 |
| **High** | 0.73 | 0.71 | 0.72 |


### 3.3. Predicted versus Observed Flood Impact in Nepal

Comparing predicted Lifeyears Index (LYI) flood impacts with observed data showed good correspondence
between high-risk areas identified by the ML method and historical flood locations in Nepal. This suggests that the proposed
approach effectively delineates flood risk on a national scale. Figure 9 illustrates this comparison, showcasing observed
(empirically evaluated) and ML-predicted LYI values at both watershed (upper row) and district (lower row) levels.
The 'observed' LYI values were empirically calculated from observational data (Table 1) and categorized into three
groups: 'low', 'medium', or 'high', with basins/districts labelled as 'high' for LYI values exceeding 1000 years, 'medium'
between 100 and 1000 years, and 'low' below 10 years. The 'predicted' values represent the outputs from the machine
learning model.
In Nepal, we achieved an overall training accuracy of 97% and a test accuracy of 63%. Notably, training the model
at the watershed level yielded higher accuracy compared to the district level. This is attributed to watersheds being
hydrologic units that integrate geomorphological and climatic properties, thus providing a more accurate representation of
flood dynamics compared to administrative district boundaries.
At the watershed level, nearly all year ranges exhibited a 100% match with observed impacts. In instances where
the model's accuracy fell below 100% (e.g., 1985–90 and 1990–95), the LYI values in the affected watersheds were low,
indicating that the predictors considered were more indicative of major flooding events. The superior accuracy achieved at
the watershed level underscores the value of implementing the model at this scale when scaling up the system.

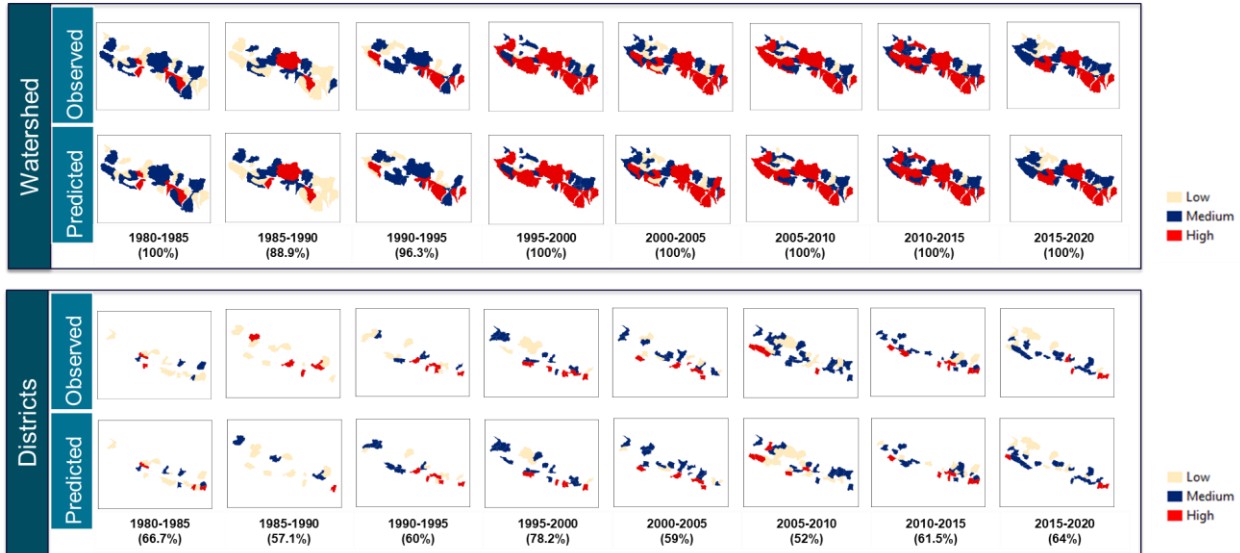


Figure 9: Comparison of prediction with actual socioeconomic impact for watersheds and districts in Nepal. Basin/districts are marked as "high" for LYI over 1000 years. Medium is between 100 and 1000, and low is less than 10. Numbers in parentheses represent accuracy.

### 3.4. Prediction of Socioeconomic Impact of Heavy Rainfall over HMA

We applied the trained model for the watersheds in HMA to five-year intervals from 1980 to 2020. As an example, Figure 10(c, d) shows the predicted basin-averaged LYIs (Low-Med-High) for the watersheds in HMA for two different timelines. The yellow circles highlight the changes in flood impact over the decades. One must consider that most HMA have low population density (blue color in Fig. 10 b), and as expected, the proposed model predicts low flood socioeconomic impacts for these regions. Hotspots of high impacts (Red colors in Figures 10c and d) are present, where population exposure is higher.

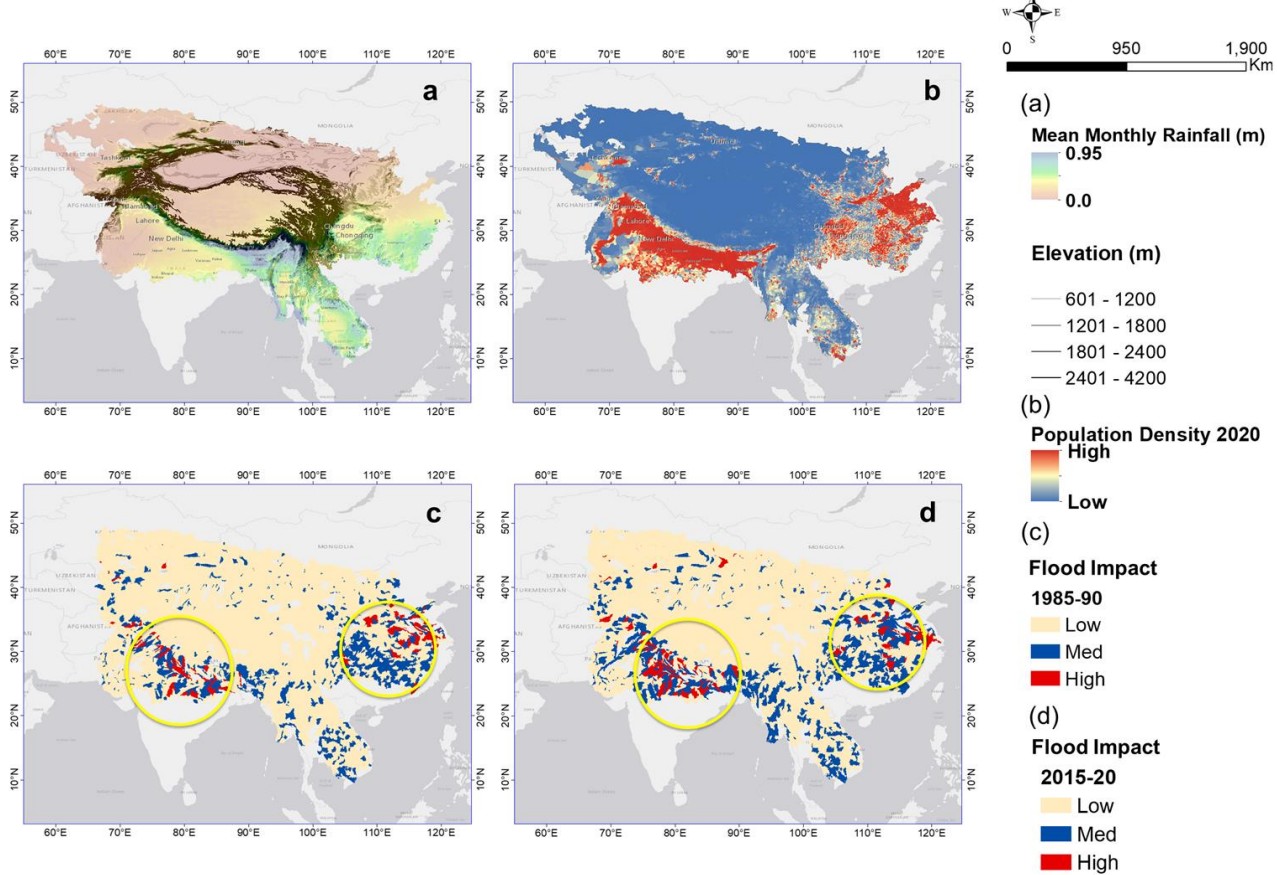

**Figure 10: a) Mean Monthly Rainfall overlayed on Elevation (contours); b) Population density 2020; c,d) Example of predicted basin-averaged flood impact for HMA (left, 1985–90; right, 2015–20). Yellow circles denote the changes in flood impact between the two timelines.**

Summarizing the results presented in Table 3, we can say that, for the years shown, we predicted almost 57% of watersheds (marginal) having LYIs between 1 and 100 years (Low), 35.9% for LYIs between 100 and 1000 years (Med), and only 6% for LYIs greater than 1000 years (High). For the entire period, most of the time we predicted LYIs of 1 to 100 years, for which we captured events of DFO severity around 2 ($10^2$ Deaths + displaced) (conditional = 28.6%). This suggests that most "Low" class DFO events did happen in the watersheds within the lowest predicted LYI range. Readers must consider that "Low" in this case means the flood impact can range from 1 to 100 years lost, and a DFO value of 2 means total deaths and displaced is on the order of $10^2$ people. The events with a DFO value of 4 happened mostly in watersheds with predicted LYIs ranging between 1 and 100 years and between 100 and 1000 years. The events with DFO 6 and 8 happened mostly in ranges greater than 1000 years and between 100 and 1000 years.

**Table 3: LYI compared to DFO flood damage.**

| DFO | LYI | NO | Prop | Marginal Probability | Conditional Probability |
|-----|-----|----|----|----|----|
| 2 | 1–100yr | 54 | 16.6 | 58.2 | 28.6 |
| 2 | 100–1000yr | 26 | 8.0 | 35.7 | 22.4 |
| 2 | >1000year | 5 | 1.5 | 6.2 | 25.0 |
| 4 | 1–100yr | 92 | 28.3 | 58.2 | 48.7 |
| 4 | 100–1000yr | 45 | 13.8 | 35.7 | 38.8 |
| 4 | >1000year | 5 | 1.5 | 6.2 | 25.0 |
| 6 | 1–100yr | 42 | 12.9 | 58.2 | 22.2 |
| 6 | 100–1000yr | 44 | 13.5 | 35.7 | 37.9 |
| 6 | >1000year | 8 | 2.5 | 6.2 | 40.0 |
| 8 | 1–100yr | 1 | 0.3 | 58.2 | 0.5 |
| 8 | 100–1000yr | 1 | 0.3 | 35.7 | 0.9 |
| 8 | >1000year | 2 | 0.6 | 6.2 | 10.0 |


We further investigated how our predicted LYI behaved when it was related to the total population (Table 4),
evaluating, as suggested by (Noy, 2014), the LYI per capita (that is, the number of lifeyears lost per 100k people). As Table
4 shows, we correctly predicted over the years almost 64% of watersheds (marginal) have LYI/100k people less than 1 year
($10^0$), 24.3% at 10year/100k people ($10^1$), 11% at 100year/100k people, and 0.6% at 1000years/100k people. We noticed
that LYI/100k people reached, at most, 6000 for Nepal (at the country scale), and the study by Noy. 2016a also reported
similar values for Nepal in 1987. (Noy, 2016a) reported actual LYI data in the range of LYI > 1000/100k people in South
Asia and stated that the higher number of damages in East and South Asia is likely due to wide-scale flooding. This gave
assurance of the consistency of our prediction with the actual data available. When looking at LYI/100k people, we found
that, for the whole timeframe, most of the floods that registered in the DFO with low severity (DFO = $10^2$
Deaths+displaced) happened in watersheds for which the predicted LYIs were between 1 and 100 years (conditional =
29.8%). This confirmed once again that, in most cases, "low"-risk events did happen in the watersheds having the lowest
predicted range (similar to the findings presented in Table 3). As before, while the probability of a watershed's being labeled
as high risk (LYI>1000year/1000k people) by our system was only 6%, the probability of these watersheds having
experienced events recorded by the DFO as having a great impact (DFO severity > 6, meaning over 1 million people) rose to
40% and 10%.



**Table 4: LYI/100k compared to DFO flood damage.**

| DFO | LYI | NO | Prop | Marginal Probability | Conditional Probability |
|-----|-----|-----|------|----------------------|-------------------------|
| 0 | 0 | 13 | 3.8 | 65.0 | 5.9 |
| 0 | 1 | 1 | 0.3 | 23.5 | 1.3 |
| 0 | 2 | 1 | 0.3 | 10.9 | 2.7 |
| 2 | 0 | 62 | 18.2 | 65.0 | 28.1 |
| 2 | 1 | 13 | 3.8 | 23.5 | 16.3 |
| 2 | 2 | 9 | 2.6 | 10.9 | 24.3 |
| 2 | 3 | 1 | 0.3 | 0.6 | 50.0 |
| 4 | 0 | 97 | 28.5 | 65.0 | 43.9 |
| 4 | 1 | 34 | 10.0 | 23.5 | 42.5 |
| 4 | 2 | 10 | 2.9 | 10.9 | 27.0 |
| 4 | 3 | 1 | 0.3 | 0.6 | 50.0 |
| 6 | 0 | 47 | 13.8 | 65.0 | 21.3 |
| 6 | 1 | 32 | 9.4 | 23.5 | 40.0 |
| 6 | 2 | 15 | 4.4 | 10.9 | 40.5 |
| 8 | 0 | 2 | 0.6 | 65.0 | 0.9 |
| 8 | 2 | 2 | 0.6 | 10.9 | 5.4 |


Figure 11 shows the LYI per 100k people (LYI/100k) evaluated for different time frames for all the locations
reported in the DFO database to compare the DFO severity with our predictions. Overall, the DFO and predicted results were
quite consistent instead with some minor variability in some scattered areas. When we compared the changes over time, we
noticed an increase in vulnerability. As the plot makes evident, the largest changes took place in 1990–95 and 2010–15; the
two concentrated areas were Nepal and China. As Figure 3 shows, two big jumps occurred during these timelines for Nepal
because of extreme storm-induced flood events. In Figure 3, we have discussed the predominant events that occurred in these
timelines.  Regarding China, as of June 2010, more than 29 million people had been affected by flooding, with up to 2.37
million evacuated and 195,000 homes destroyed (China: Floods Information Bulletin N° 1 GLIDE N°, 2010).

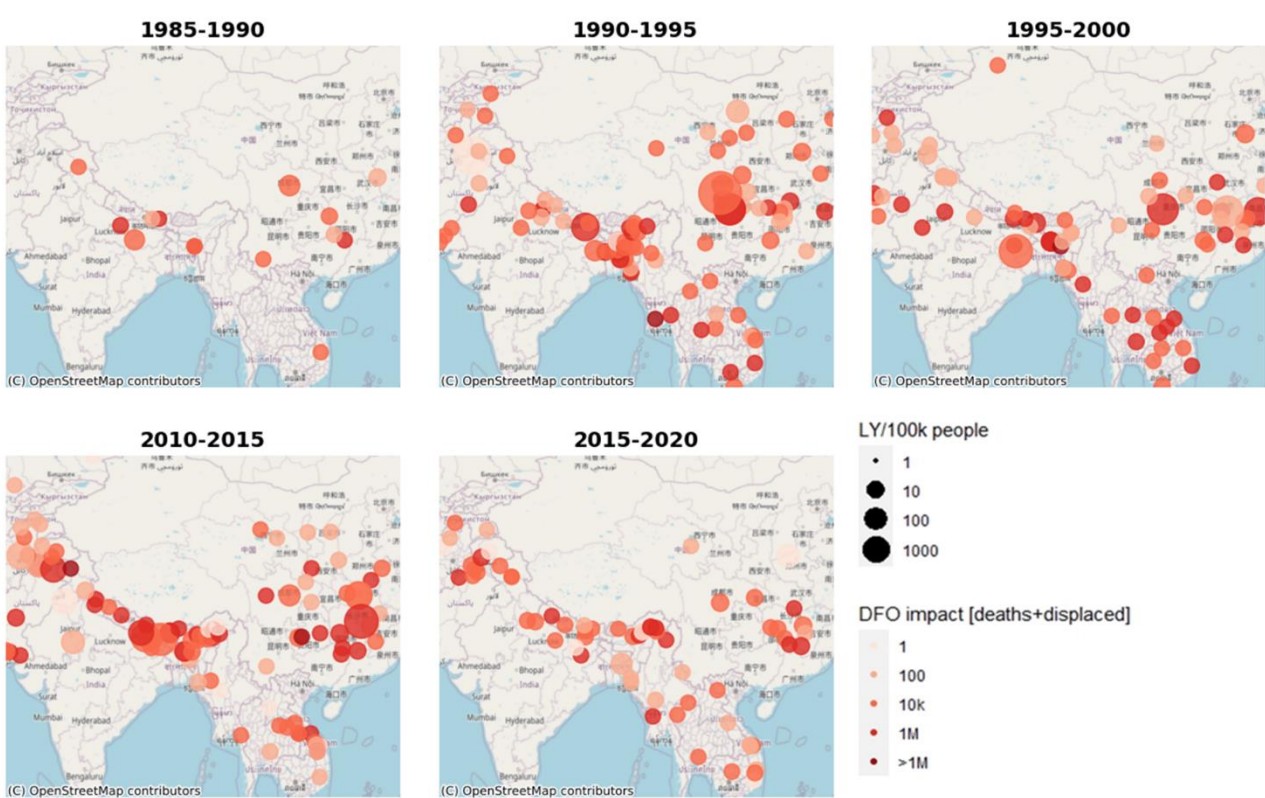


**Figure 11: Comparison of DFO and LYI/ 100K people for all the timelines**
**3.6. Change in Socioeconomic Impact over Time**

Figure 12a presents our maps of the watersheds where flood impacts increased over time. Furthermore, Figure 12b
shows our evaluation of the percentage changes in the number of watersheds between timelines, focusing on three different
changes: low to medium (LtoM); medium to high (MtoH); and low to high (LtoH). Some watersheds have not changed, and
some have decreased impact. For the sake of highlighting potential increases in flood impacts, we focused on those locations
where risk increased over time, from low to medium, or medium to high. The largest changes were from LtoM for all the
timelines, which represented a notable change in vulnerability. Several watersheds showed higher flood impacts (from low to
medium, medium to high, and low to high) in recent years as compared to 1985–90. Again, we observed the largest changes
for 1990–95 and 2010–15, which was consistent with Figure 12. The exposure changed significantly, along with the intensity
of the events; hence, the risk of flooding was heightened in these areas.

Impact changes from Low to High were next, according to the number of watersheds changed for all the timelines.
It was obvious that more changes would happen overall, but the comparison of the 1990–95 and 1995–2000 timelines
demonstrated that heightened flood impact occurred in a considerable number of watersheds within a brief period. For many
watersheds, the risk was heightened by a population boom during the overall period.

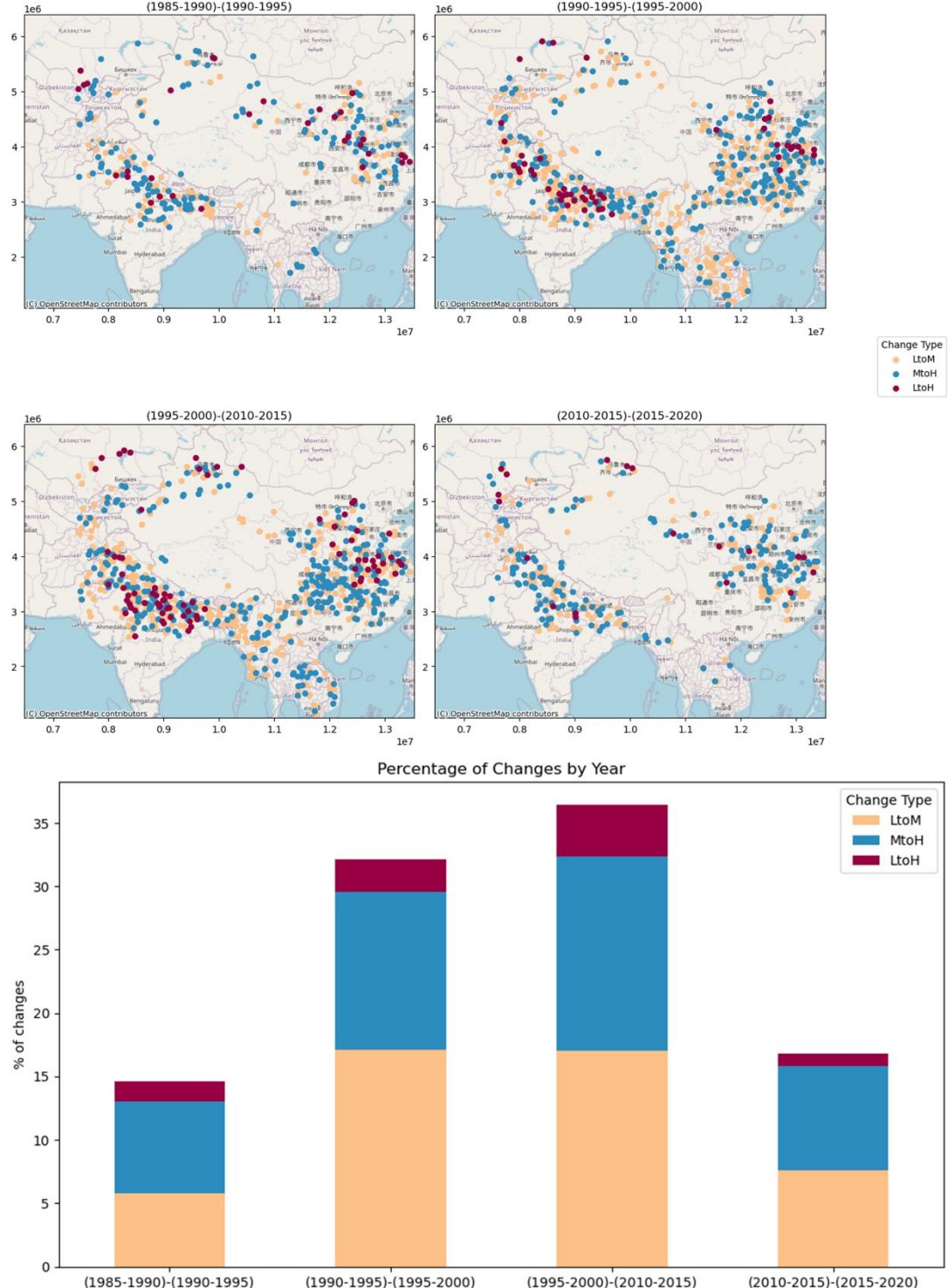


**Figure 12: Flood impact change in HMA over time**

### 3.7. Model constraints and limits

While this study demonstrates the promise of accurate flood impact prediction, the use of static Flood Geomorphic Potential (FGP) maps presents limitations. Flooding alters channel morphology and downstream topography, impacting future flood dynamics (Khanam et al., 2024). Therefore, dynamic flood topographies are essential for robust hazard assessment. Although high-resolution data post-extreme events can enhance prediction accuracy, the availability of such data is constrained by acquisition frequency. Hence, efforts to improve data availability post-disaster are crucial for enhancing the reliability of predictive models. Researchers could also derive FGPs from enhanced high-resolution terrain data, such as that derived from LiDAR sources, if available. In such cases, however, it is advisable to retrain the model and reassess the significance of this parameter in the updated model, as terrain resolution and survey techniques might determine a variability of the data, especially when dealing with hydrologic parameters (Sofia, 2020a).

The climate index considered in this study might vary depending on the input dataset (Reanalysis VS measurements), as well as on the timescale of the analysis. When comparing results to this study, researchers should make careful consideration of the length of the time window used for this evaluation (5 years). If daily data are considered over shorter time windows (e.g., 1 year), the index itself might result in higher values, capturing only short-term variability due to specific isolated storms. Seasonal analyses, on the other hand, would capture more of the concentration due to monsoon periods, or dry vs. wet months. The proposed multi-year analysis is in line with literature studies on climate change and on the effect of flooding (Sofia et al., 2019; Saki et al., 2023; Du et al., 2023).

Population data for this work relies on standard available datasets. When considering the method to predict future changes, outside the time range covered by the proposed model, headcounts alone cannot offer a full picture. It is also crucial to consider additional elements that could determine population shifts over time.

### 4. Conclusions

High Mountain Asia (HMA) presents a multifaceted landscape characterized by rugged terrain, diverse climates, rich vegetation, and substantial population exposure to natural disasters. Given its susceptibility to natural disasters, effective management is imperative for the region's long-term sustainability. Addressing the considerable threat posed by flooding demands a comprehensive strategy involving disaster risk reduction, sustainable land use practices, and climate change mitigation.

In this study, we introduced a simplified approach to identify vulnerability hotspots within the HMA region, focusing on intense rainfall events. To map the socioeconomic flood vulnerability, we employed a remotely sensed data-driven model integrating geomorphological and climate variability factors. This adaptable framework can be tailored to various regions, provided that similar terrain and climate datasets are available, accommodating adjustments to flood drivers such as climate and geomorphology, as well as population dynamics. The resulting predictions offer valuable insights into vulnerabilities across HMA watersheds, facilitating proactive flood management planning.

The novelty of our study lies in the efficiency and versatility of the proposed predictive model. Requiring only a small number of variables, our model accurately forecasts the socioeconomic impact of pluvial and fluvial flooding events. In densely populated, possibly ungaged regions with rapidly changing climates, such a model serves as a valuable decision-

making tool for stakeholders. The efficacy of the framework, as demonstrated in Nepal, underscores its potential applicability across regions with similar climatic and morphological characteristics. Our goal is to provide a reasonable assessment of vulnerability through life years lost, rather than to definitively classify flood-prone areas by societal factors. Despite certain limitations, our findings offer valuable insights into regional flood risk and its key drivers.

With advancing technology, we can now predict the drivers of impending extreme events, enabling proactive measures to mitigate their impact. Stakeholders could leverage our model to forecast vulnerability to future flood events with precision, enhancing hazard assessment, decision-making, planning, and mitigation efforts.

**Competing interests:** The contact author has declared that none of the authors has any competing interests.

**Acknowledgment:** This work was supported by the NASA High Mountain Asia program (grant #80NSSC20K1300).

**Data Availability:** The FGP products are available at NASA National Snow and Ice Data Center (NSIDC) (https://nsidc.org/data/hma2_fgp/versions/1)

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
