# Peer review of "Predictive Understanding of Socioeconomic Flood Impact in Data-Scarce Regions Based on Channel Properties and Storm Characteristics: Application in High Mountain Asia (HMA)"

_Natural Hazards and Earth System Sciences, 2023_

## Referee Comment (RC1)

**Predictive Understanding of Socioeconomic Flood Impact in Data-Scarce Regions Based on Channel Properties and Storm Characteristics: Application in High Mountain Asia (HMA)**

The authors present a machine learning approach to describe flood impact at the regional scale for individual watershed, based on the example of High Mountain Asia. They do so with a simple XGBoosting approach and reproducing life years lost per watershed.

*General comments*

The study has merit in its outline, as a better understanding of flood impacts at this scale is definitely needed and I agree that doing this with a generally simple approach for a general kick start to the options of AI in this domain makes sense. However, I have a number of concerns on how the study is built and executed, which I believe are at this stage too big to recommend the manuscript for publication. I detail the concerns below and would encourage the authors to rethink their strategy before moving to an eventual submission. I fully understand that this is a submission from an ECR and I want to complement you on the aim and pulling this together – definitely work that should be pursued and there is a lot of demand for outcomes of such approaches! I would have hoped to see more scrutiny here before a submission from the more experienced co-author team.

My concerns range from (a) general sloppiness of manuscript writing (many simple editing mistakes that can always happen for drafts but should not occur for a submitted manuscript over (b) the lack of appreciation of existing data and simply depicting the target region as 'data scarce' to avoid scrutiny from what is known already to (c) a lack of proper documentation of data sources on the exposure side as well at times confusing jumping between topical (what types of floods) as well as spatial (national, watershed, HMA wise) domains. I briefly summarize these concerns below and then present a list of line indexed responses for the complete manuscript.

a) Sloppy mistakes
   In numerous instances references are reported as 'n.d.' where they actually have a date and some are completely missing from the reference list. There are many instances with missing spaces as well and figure captions are often incomplete. Please be careful on such matters before submitting

b) General statement on 'no data'
   You make general unsupported statements on the region being data scarce on hydromet data. That is decidedly not the case. While data may often not be readily accessible, it is available and many studies have been published on this, especially for China and India and data is generally reachable from China, India, Pakistan, Afghanistan as well as Central Asian states. The data that is available you do away with as 'not trustworthy' in a single sentence. This coming from an all-US based author team is problematic and I guess you could imagine how stunned a reviewer from the US (or Europe) would be if a Chinese author would make that claim before proceeding to apply ML on all of the US or Europe. You will need to make a clearer description (with references) on what is lacking and how your approach fills that gap.

c) Poor documentation of socio-economic data
   As I detail below there is very poor documentation on where the exposure data is taken from and there is no way to make this traceable (no stable links, and also no attempt so far to make your own produced data available, see comment on Availability statement). I also fail to see how you take census data to the watershed and how you align using Nepal government data with your approach to model at the watershed scale (which do not follow national borders).

d) General scope and methodology

At multiple points of the manuscript I was a bit confused on the scope. There is an introduction on all types of high flow events but the methods suggest you only look at fluvial floods with exceptionally high impacts. There is a relatively rapid investigation of the methods for watersheds that do lie to some part in Nepal compared against data only from areas within Nepal and then an upscaling to all of HMA, which in turn is not clearly defined in its scope or climatologies. I would strongly suggest to maybe limit the study to areas where data is available before scaling it up, allowing you more space for methodological and data based issues.
* * *
*Specific comments:*

There are multiple citations as 'n.d.' while actually they are published and have a year – please check your references carefully.

L31f: Be careful in your framing – population growth does not increasing likelihood of flooding, it increases flood risk! Also, in the abstract and your general analysis, you focus on precipitation as a flood driver but here then passingly mention glacial melt as well – those are very different flood drivers and would be crucial to be clear what kind of flooding you wish to tackle here.

L54: '*HMA does not have enough hydrological stations for region-wide flood monitoring*' is a huge statement to make without a citation – what is an appropriate number? Also most countries in HMA, especially China, India, Pakistan and Nepal have large and dense network of hydro(-met) monitoring, which they also use for forecasting. That is not as open as in the US, but the statement that there is 'not enough' needs to be qualified. You then claim '*Moreover, the available meteorological datasets may not be sufficiently trustworthy.*', which again lacks any qualification. Imagine me making that statement for a European or North American country, that would be thrown out. The region has a large amount of met data (see e.g. the overview figure in (Nepal et al. 2023)) and if you do not trust the data you need to justify why.

L61: '*The use of remote sensing technology for disaster studies in HMA is comparatively new*' – I also do not quite agree. Remote sensing itself isn't very old and it has been used in HMA for many studies already (which maybe anyway would need some acknowledgement here).

L87: You focus here a lot on monsoon changes with intense precipitation – but if you actually focus on HMA (rather than just the Hindukush Himalaya) there are a lot of other processes – Westerlies in Central Asia, Eastern Monsoons in the Upper Yangtze etc. Maybe it is required to reconsider the total spatial scope of the study here?

L92: You now finally get to actual numbers of potential affected, but leave it to the reader to get the data from EMDAT. It would be prudent to explain here (or rather in the introduction) what the actual numbers are and for what types of hazards, to then narrow down and which ones you actually focus.

Figure 1: Up to this point there was no clear description how the watersheds are selected, i.e. what boundary you used for HMA. This needs to be provided to give context to why so many watersheds outside HMA are also included.

L116: At this point you mention that you will predict impacts of 'floods', i.e. all of them? The way you describe your research you are narrowing this down on pluvial floods, as glacial lake outburst floods or debris flows etc need very different driver analysis. Can you be precise here? In L185 you then suddenly just focus on 'fluvial flooding', so is it just that you focus on?

L120ff: This part is crucial as you present the socioeconomic data and how you treat it. However there are a few issues that would need to be addressed with respect to traceability and presentation of data used.

- You refer to data sources that are questionable, the knoema.com page is not stable and it is unclear from where their data is sourced or where it is known needs to be documented here.
- You refer to general government and Worldbank websites (like http://drrportal.gov.np/) that exist but what data you took from there at what point in time remains unclear. Copernicus Journals subscribe to FAIR practices, that includes the documentation of third party data used in a publication.
- You introduce a lot of data as well as parameters from literature (like T and e) without any questioning of their accuracy, uncertainty etc. This would propagate and need to be addressed, especially as you seem to upscale from this approach with a few numbers on Nepal government websites to all of HMA.
- You calculate these values for Nepal as a whole but then work on the watershed scale – how is this compatible?
-

Figure 3: I am not sure whether these are now LYl only due to floods or all disasters. Considering that there are no jumps for earthquake events like 2015, I assume this has been calculated for floods only? Then this needs to be made very clear in the caption rather than just calling it 'disasters'.

L176 + Figure 4: What is HAND in Figure 4. Is this from (Delalay et al. 2018)? The publication is not open access and only limited to Sindupalchowk, how does it go to all of Nepal? What does it actually map?

L194f: While I understand that it would be well beyond the scope of this study to evaluate the suitability of ERA5 data for flood simulations (let alone in a mountain context where precipitation products are of poor quality) but it would be crucial to address this and dispel concerns from the get go by referring to discussions of this data in mountain regions as well as for flood mapping.

L210: As for the other socioeconomic data above, the description of population data here remains lacking. For Nepal you only refer to the Census Bureau, which does not report distributed data or data by watershed (so how was that brought in line with inundation maps) and you also do not specify where on the general page you retrieved the data from. You then refer to the GHSL but do not provide a citation or link where this data was retrieved. Distributed data in Asia is generally of problematic and definitely not homeogenous quality, hence a discussion of how this was dealt with need a much more thorough description than the short paragraph here without any references. A detail but you also call it LYI (capital I) here while it should be LYl (lower case L)!

L227ff: You discuss your first results here on the F score and model performance discussion – this should come under Results and Discussion respectively, not Methods! Figure 6 as well as Table 1 also lacks a description of variables and results presented. Unclear how this should be interpreted.

L248f: Apart from the Brakenridge citation not having a date nor being present anywhere in the references, and agreeing that in principle such a dataset would be an interesting set for validation, the fact that the whole dataset only has 46 events from Nepal since 2021 and <10 with the 1000 deaths plus displaced criterium you introduce below makes its use questionable considering this is the area you run your model in. Wouldn't data from Nepal (like https://bipadportal.gov.np/) be much more appropriate then? Also this database captures lowland floods, rather than mountain floods, making me wonder whether the aim to characterize 'High Mountain Asia' floods is really the right scope here. Also the DFO reports single coordinates, are you then simply assuming the watershed that matches the coordinate is the only one affected? Likely the reported numbers refer to much larger areas, as the size of the watershed you chose is rather small (guessing from the Figure, it's not actually described anywhere!)

L261: You include a crucial boundary condition of your model here, i.e. '1000 deaths plus displaced'. Does this mean your model will only be useful in this domain? It would be crucial to report how many such events have actually happened in your domain then. Also how is the adding up of 'dead and displaced' justified? These are quite 'different' responses to a flood.

Figure 9: Panel a is elevation not rainfall as your legend suggests!

L265ff: To be honest I am not entirely sure how I should interpret Figure 7 – doesn't it just confirm that people live close to wide river channels? Then there is really no link to atmospheric characteristics as you claim in L270. There is a lot of discussion already as well on convection patterns all stemming from other literature and not really relevant to what I read in the Figure.

L295f: A main concern I have here is that I am still not very clear on where the observed events come from you compare this to. I am also wondering if your Figure 8 simply only confirms one thing – that there are many people (an input to your model) where there are many people (a validation of your model). How does your model compare on actually coming up with an observed flood from the input 'ERA5 rain'? This concern then propagates into the result for the whole region, where you 'predict' the biggest impacts with the highest population densities. That isn't quite so surprising and it is unclear to me how I can see the power of ML in these results. To be provocative, would the results have been different if you would have just distributed rainfall across the watersheds without a model in between?

L349F: The figures you note here do not show what is described in the text.

L356f: I lack some context here - <10% of watersheds see an increase, are all other stable or see a decrease? How can you differentiate here between hazard (rain) and exposure (population) as a driver of change? How do you explain that increase has slowed after 2010 significantly? And how is it possible that in the 1995-2010 jump the number of increasing watersheds is similar to the just 5 year jump between 1990 – 1995? Isn't that completely counterintuitive?

L406: While in general 'an intention to make data available' shouldn't be followed, for a journal like NHESS this is definitely not acceptable. Data availability needs to be clearly described (or arhued why this is not the case).

*Technical corrections (Minor issues):*

L14: 'from flooding and debris flows'

L33: missing space

L58: 'is foremost'?

L82: This is not correct, HMA includes the Tibetan Plateau, Tien Shan, Pamir, Qilian Shan etc etc, while the 'Hindukush-Himalaya', as the name suggests only comprises the southern fringe of HMA. HMA furthermore includes Myanmar.

L85: Missing citation of the population number – also does that include all of the above countries or just areas above a certain elevation?

L88: 'located in this region'

L95: 'HMAT'?

Figure 2: Issue in caption after 'section 2.2.1'

L177: Spacing issues like here are found throughout the manuscript and need to be carefully checked before submission!

References:

Delalay, Marie, Alan D. Ziegler, Mandira Singh Shrestha, Robert James Wasson, Karen Sudmeier-Rieux, Brian G. McAdoo, and Ishaan Kochhar. 2018. "Towards Improved Flood Disaster Governance in Nepal: A Case Study in Sindhupalchok District." *International Journal of Disaster Risk Reduction* 31 (October): 354–66. https://doi.org/10.1016/j.ijdrr.2018.05.025.

Nepal, S, J F Steiner, S Allen, F M Azam, S Bhuchar, H Biemans, M P Dhakal, et al. 2023. "Consequences of Cryospheric Change for Water Resources and Hazards in the Hindu Kush Himalaya." In *Water, Ice, Society, and Ecosystems in the Hindu Kush Himalaya: An Outlook*. Kathmandu, Nepal: ICIMOD.

---

## Author Comment (AC1)

*First, we want to thank the reviewer for the insightful comments and recommendations. Following the suggestions, we will revise the manuscript. Detailed responses to the reviewers' comments are added below.*
**Note: Below is our response (italics) to each comment (regular font) from the reviewer**

*General comments*

The study has merit in its outline, as a better understanding of flood impacts at this scale is definitely needed and I agree that doing this with a generally simple approach for a general kick start to the options of AI in this domain makes sense. However, I have a number of concerns on how the study is built and executed, which I believe are at this stage too big to recommend the manuscript for publication. I detail the concerns below and would encourage the authors to rethink their strategy before moving to an eventual submission. I fully understand that this is a submission from an ECR and I want to complement you on the aim and pulling this together – definitely work that should be pursued and there is a lot of demand for outcomes of such approaches! I would have hoped to see more scrutiny here before a submission from the more experienced co-author team.

1. My concerns range from (a) general sloppiness of manuscript writing (many simple editing mistakes that can always happen for drafts but should not occur for a submitted manuscript over

**Response:** *We will revise the manuscript thoroughly to correct these unnoticed mistakes.*

2. (b) the lack of appreciation of existing data and simply depicting the target region as 'data scarce' to avoid scrutiny from what is known already to

**Response:** *We thank the reviewer for this comment. Please note that our intent was not to avoid scrutiny of the data, but we understand that some of the statements were phrased in an ambiguous way, which we will rework to address accordingly.*

3. (c) a lack of proper documentation of data sources on the exposure side as well at times confusing jumping between topical (what types of floods) as well as spatial (national, watershed, HMA wise) domains.

**Response:** *We will provide a detailed description of the datasets considered in this study, and we will frame the work in a more structural manner. We discuss different flood disasters happening in the HMA region. Specifically, our study focuses on pluvial and fluvial flooding, which we will make clearer in the introduction and throughout the manuscript.*

4. I briefly summarize these concerns below and then present a list of line indexed responses for the complete manuscript.

   a) Sloppy mistakes
      In numerous instances references are reported as 'n.d.' where they actually have a date and some are completely missing from the reference list. There are many instances with missing spaces as well and figure captions are often incomplete. Please be careful on such matters before submitting

**Response:** *We will make sure to do a thorough check of the manuscript and correct these mistakes.*

5.
   b) General statement on 'no data'
      You make general unsupported statements on the region being data scarce on hydromet data. That is decidedly not the case. While data may often not be readily accessible, it is available and many studies have been published on this, especially for China and India and data is

generally reachable from China, India, Pakistan, Afghanistan as well as Central Asian states. The data that is available you do away with as 'not trustworthy' in a single sentence. This coming from an all-US based author team is problematic and I guess you could imagine how stunned a reviewer from the US (or Europe) would be if a Chinese author would make that claim before proceeding to apply ML on all of the US or Europe. You will need to make a clearer description (with references) on what is lacking and how your approach fills that gap.

*Response: We will revise this part of the manuscript and rephrase our statements to avoid confusion about data scarcity. It was not our intention to underrepresent available datasets. We will clarify that gathering data for the scale of the HMA region is a difficult task, as it requires collecting data from several countries and multiple sources, and this poses challenges due to the possible inhomogeneities of standards between different organizations. Especially in the context of the impact of floods using socioeconomic data, the analysis involves examining the number of fatalities, injured and people otherwise affected, as well as the financial damage that natural disasters cause, and this information is generally not always available, or it is collected at the local scale based on reported events. Major disasters are reported in global databases such as EMDAT, or, for Nepal, the DRR portal as we used in our paper, but these datasets are also not complete. For example, EMDAT only considers events with at least one of the following criteria:*

- *10 fatalities*
- *100 affected people*
- *a declaration of state of emergency*
- *a call for international assistance*

*The main idea behind this paper was to provide a tool for a rapid estimate of potentially highly impacted areas, based on information accessible and updateable quickly, such as population number, rainfall intensities, and a geomorphologic index that can be derived from global DEMs (or high-resolution local ones when available).*
*We will rework our manuscript to be clearer with our statements.*

*6.*
    c) Poor documentation of socio-economic data
        As I detail below there is very poor documentation on where the exposure data is taken from and there is no way to make this traceable (no stable links, and also no attempt so far to make your own produced data available, see comment on Availability statement).

*Response: We thank the reviewer for this comment. We will include a table with information on all the datasets used. Regarding the links being "not stable" – the data required for the index were accessed and we tested the links before submission. In the revised paper, we will clarify the date of the latest access so that the data is more clearly referenced.*
    7. I also fail to see how you take census data to the watershed and how you align using Nepal government data with your approach to model at the watershed scale (which do not follow national borders).

*Response: We will clarify this in the manuscript. For Nepal, we considered **a weighted spatial join between the watersheds and the districts.** To each watershed, we attributed the statistics of the district intersecting it, weighted by the overlapping areas. In general, the districts we have for Nepal are of a smaller extent than those of the various watersheds.*
*8.*
    d) General scope and methodology
        At multiple points of the manuscript I was a bit confused on the scope. There is an introduction on all types of high flow events but the methods suggest you only look at fluvial floods with exceptionally high impacts. There is a relatively rapid investigation of the methods for

watersheds that do lie to some part in Nepal compared against data only from areas within Nepal and then an upscaling to all of HMA, which in turn is not clearly defined in its scope or climatologies. I would strongly suggest to maybe limit the study to areas where data is available before scaling it up, allowing you more space for methodological and data based issues.

*Response: We thank the reviewer for this suggestion. In general, this paper focuses on fluvial and pluvial flooding, and we will make this clearer in the introduction. Starting from this, for this work, we considered Nepal as our train site for two main reasons.*

1. *We had information at fine resolution regarding the flood events, in terms of the number of people, economic impact of the event, date of the event, and population data.*
2. *For Nepal, at the time of this paper we had access to the high-resolution 8m DEM from the previous NASA HIMAT effort. This DEM also covers other areas of the wider Himat region, but it presents some gaps. Nepal was completely covered, and we verified the homogeneity and quality of the data.*

*It is true that the climatology in HMA is variable. In Nepal, as well, we have regional climate variations largely being a function of elevation. For this work, for the main rainfall driver of the model, we focused on climate concentration. This index was proven to be highly linked to pluvial/fluvial flooding impacts in other regions of the world, including for example Italy (both in mountainous landscapes and floodplains (Sofia et al. 2019), the US (Saki et al. 2023), or China (Du et al., 2023). Climate concentration values are mostly related to the temporal variability of the rainfall, not to the total amount or to the average yearly and seasonal statistics, and its variability captures well various climates (Monjo et al. 2016). For Nepal, as we showed in the paper, we have a gradient of CI values, and as ML models learn from the data they ingest, we believe the system can work across various regions from the climatic point of view. We will add comments on this in the paper, highlighting the strengths and weaknesses of the approach.*

*Specific comments:*

1. There are multiple citations as 'n.d.' while actually they are published and have a year – please check your references carefully.

*Response: We will correct these carefully.*

2. L31: Be careful in your framing – population growth does not increasing likelihood of flooding, it increases flood risk!

*Response: We will rephrase this in the revised manuscript. The sentence was meant to refer to flood impacts.*

3. Also, in the abstract and your general analysis, you focus on precipitation as a flood driver but here then passingly mention glacial melt as well – those are very different flood drivers and would be crucial to be clear what kind of flooding you wish to tackle here.

*Response: We will be clearer on the fact that this work focuses on pluvial/fluvial flooding. We added background information on flood hazards in general in the region, as we believe some overview on this is needed to frame the work correctly in the context of the various possible flood hazards in the region.*

4. L54: '*HMA does not have enough hydrological stations for region-wide flood monitoring*' is a huge statement to make without a citation – what is an appropriate number? Also most countries in HMA, especially China, India, Pakistan and Nepal have large and dense network of hydro(-met) monitoring, which they also use for forecasting. That is not as open as in the US, but the statement that there is 'not enough' needs to be qualified. You then claim '*Moreover, the available meteorological datasets may not be sufficiently trustworthy.*', which again lacks any qualification. Imagine me making that statement for a European or North American country,

that would be thrown out. The region has a large amount of met data (see e.g. the overview figure in (Nepal et al. 2023)) and if you do not trust the data you need to justify why.

*Response: As mentioned before, we will revise this part of the manuscript and rephrase our statements. It was not our intention to underrepresent available datasets. Gathering data for the scale of HMA region would require collecting data from several countries. This is time-consuming and to some extent, it can be nearly impossible due to political constraints and limits on accessing the data for some regions. As a matter of fact, our goal for this study was to overcome the data and time constraints and provide a quick tool for disaster management. We will try to be clearer with our statements.*
*Also, please refer to the response to comment no. 8 regarding the datasets available to us throughout the scope of the project.*

5. L61: '*The use of remote sensing technology for disaster studies in HMA is comparatively new*' – I also do not quite agree. Remote sensing itself isn't very old and it has been used in HMA for many studies already (which maybe anyway would need some acknowledgement here).

*Response: We will rephrase this, adding references to wider literature.*

6. L87: You now focus here a lot on monsoon changes with intense precipitation – but if you actually focus on HMA (rather than just the Hindukush Himalaya) there are a lot of other processes – Westerlies in Central Asia, Eastern Monsoons in the Upper Yangtze etc. Maybe it is required to reconsider the total spatial scope of the study here?

*Response: We wanted to draw the focus on the fluvial flooding caused by the precipitation, not the actual processes. We can add the processes mentioned by the reviewer as additional examples of flood drivers of the flooding in HMA region.*
*It is true that the climatology in HMA is variable. In Nepal, as well, we have regional climate variations being a function of elevation. For this work, for the main rainfall driver of the model, we focused on climate concentration. This index was proven to be highly linked to pluvial/fluvial flooding impacts in other regions of the world, including for example, Italy (both in mountainous landscapes and floodplains (Sofia et al. 2019) and the US (Saki et al. 2023). Climate concentration values are mostly related to the intensity of the rainfall, not to the total amount or to the average yearly and seasonal statistics, and its variability captures well various climates (Monjo et al. 2016). For Nepal, as we showed in the paper, we have a gradient of CI values, and as ML models learn from the data they ingest, we believe the system can work across various regions. A recent work by (Khanal et al., 2023) characterized overall the variability of precipitation over the same region from ERA5 data, also highlighting an overall homogeneity in trends (aside from a few hotspots) and regional statistics of precipitation.*

7. L92: You now finally get to actual numbers of potential affected, but leave it to the reader to get the data from EMDAT.
It would be prudent to explain here (or rather in the introduction) what the actual numbers are and for what types of hazards, to then narrow down and which ones you actually focus.
*Response: The EMDAT link was used as a reference to the statement. We can certainly add the numbers calculated. This was not provided to direct the reader to get the data from EMDAT.*

8. Figure 1: Up to this point there was no clear description how the watersheds are selected, i.e. what boundary you used for HMA. This needs to be provided to give context to why so many watersheds outside HMA are also included.
*Response: We have selected the watershed covering the area that was included in the Himat project.*

*The HMA region is generally identified as the extent we considered in this paper.*

9.  L116: At this point you mention that you will predict impacts of 'floods', i.e. all of them? The way you describe your research you are narrowing this down on pluvial floods, as glacial lake outburst floods or debris flows etc need very different driver analysis. Can you be precise here? In L185 you then suddenly just focus on 'fluvial flooding', so is it just that you focus on?

*Response: Thank you for this comment. We will make sure to be consistent with the terminology. And we have used only fluvial flooding in the analysis.*

10. L120: This part is crucial as you present the socioeconomic data and how you treat it. However there are a few issues that would need to be addressed with respect to traceability and presentation of data used.

    - You refer to data sources that are questionable, the knoema.com page is not stable and it is unclear from where their data is sourced or where it is known needs to be documented here.
    - You refer to general government and Worldbank websites (like http://drrportal.gov.np/) that exist but what data you took from there at what point in time remains unclear. Copernicus Journals subscribe to FAIR practices, that includes the documentation of third party data used in a publication.
    - You introduce a lot of data as well as parameters from literature (like T and e) without any questioning of their accuracy, uncertainty etc. This would propagate and need to be addressed, especially as you seem to upscale from this approach with a few numbers on Nepal government websites to all of HMA.
    -

*Response: We will double-check the links. The parameters (like T and e) are used by (Noy et al., 2016) for the LYI, and in their work the LYI is calculated also for the countries included in our analysis. For consistency, we maintained the same standard values. We will clarify this in the paper. We will also add the following table to the manuscript where we clarify the data sources considered for the analysis.*

| Variable | Description | References |
|---|---|---|
| M | Mortality (number of deaths due to disaster | http://drrportal.gov.np/ |
| Aexp | Average life expectancy at birth (by year) | WHO (https://data.who.int/countries/524) |
| Amed | Median age (by year) | WHO (https://data.who.int/countries/524) |
| e | Welfare reduction weight associated with being exposed to a disaster | set to e = 0.054 according to Noy, 2016a, based on Mathers et al., 2013 |
| T | Time taken by the affected person to get back to normal | Noy, (2016a) |
| N | Number of affected people | http://drrportal.gov.np/ |
| c | Percent of time not used in work-related activities (.75) | Noy, (2016a) |
| Y | Y = Financial damage (value of destroyed/damaged infrastructure) | |
| PCGDP | Income per capita (by year) | http://drrportal.gov.np/ |

11. You calculate these values for Nepal as a whole but then work on the watershed scale – how is this compatible?

*Response: We did not calculate the values for Nepal as a whole. The LYI is calculated by event and summarized for each district as an overall number of LYI for all the events geolocated within that district. To scale it up to the watershed level, we did a weighted spatial join.*
*It is true that there are a few parameters that go in the LYI equation that are standard, but this is consistent with the calculation of the index as it was defined by Noy et al. 2016. Most of the other*

*parameters, such as median age, or life expectancy at birth may vary from country to country. But as we 'train' the model considering a temporal variability of this parameter, for example, we believe that the overall combination of values can account for the potential variability by country. For example, Figure R1 shows the life expectancy over time for Nepal, to show that we do have a variability in time for the index. We will clarify this in the manuscript.*

[Figure]

*Figure R1: Life expectancy at birth for Nepal over time (Source: https://data.worldbank.org/indicator/SP.DYN.LE00.IN?end=2021&locations=NP&start=1960&view=chart&year=2021)*

*For Nepal, we considered **a weighted spatial join between the watersheds and the districts.** To each watershed, we attributed the statistics of the district intersecting it, weighted by the overlapping areas. In general, the districts we have for Nepal are of a smaller extent than those of the various watersheds.*

12. Figure 3: I am not sure whether these are now LYl only due to floods or all disasters. Considering that there are no jumps for earthquake events like 2015, I assume this has been calculated for floods only? Then this needs to be made very clear in the caption rather than just calling it 'disasters'.

**Response:** *This is for fluvial and pluvial flood only. We will make sure to revise the manuscript to be clear and consistent.*

13. L176 + Figure 4: What is HAND in Figure 4. Is this from (Delalay et al. 2018)? The publication is not open access and only limited to Sindupalchowk, how does it go to all of Nepal? What does it actually map?

**Response:** *We thank the reviewer for this comment. We will clarify better the figure. The work of Delalay et al 2018 just provided information on some specific inundation depths pertaining to critical floodings with specific return periods. To highlight the importance of the FGP map, we showed an example of inundation produced from a terrain dataset using those critical depths, and the FGP maps for the same area. Please note that the FGP itself was not invented for this paper, but rather it has been published and applied in other contexts (Samela et al., 2017). For this work, we automated the computation so that it was possible to apply it widely by improving the definition of the hydrauliic scaling functions needed for the system integrating the methods also published and referenced in Sofia et al. 2015 and Sofia & Nikolopoulos 2020. Testing the quality of the FGP is not the scope of this work, as its feasibility for flood mapping has been proven already (Manfreda et al., 2011, 2014; Manfreda & Samela, 2019; Samela et al., 2016, 2018). These papers showed how these methodologies are extremely useful in ungauged conditions to preliminary identify flooded areas since they only require a DTM to*

*perform the simulations.*

14. L194: While I understand that it would be well beyond the scope of this study to evaluate the suitability of ERA5 data for flood simulations (let alone in a mountain context where precipitation products are of poor quality) but it would be crucial to address this and dispel concerns from the get go by referring to discussions of this data in mountain regions as well as for flood mapping.

*Response: This study is a part of the HiMAT project. There are a number of research groups working on different aspects of HMA. At the time we conducted this study, a subgroup of our team was working with ERA5. We wanted to utilize the available dataset and complement the existing study. We will add few comments on this, with highlights on HMA from other related works from the HiMAT team, such as* (Maggioni & Massari, 2018; Maina et al., 2023)

15. L210: As for the other socioeconomic data above, the description of population data here remains lacking. For Nepal you only refer to the Census Bureau, which does not report distributed data or data by watershed (so how was that brought in line with inundation maps) and you also do not specify where on the general page you retrieved the data from. You then refer to the GHSL but do not provide a citation or link where this data was retrieved. Distributed data in Asia is generally of problematic and definitely not homeogenous quality, hence a discussion of how this was dealt with need a much more thorough description than the short paragraph here without any references.

*Response: We thank the reviewer for this comment. Indeed, obtaining population data at a detailed scale depends on local authorities, and for Nepal, we relied on the official source of the census bureau, which provides this information by the district. To extend it to the whole HMA, we considered the dataset by Center for International Earth Science Information Network - CIESIN - Columbia University. 2018. Gridded Population of the World, Version 4 (GPWv4): Population Density, Revision 11. Palisades, NY: NASA Socioeconomic Data and Applications Center (SEDAC). https://doi.org/10.7927/H49C6VHW. Accessed DAY MONTH YEAR. This dataset provides spatially explicit estimates of population density for the years 2000, 2005, 2010, 2015, and 2020, based on counts consistent with national censuses and population registers, as raster data to facilitate data integration. The dataset is provided in raster form and can be aggregated at any scale a user wants, provided that the user has the boundary of the area they want to investigate. We will clarify this in the revised manuscript.*

16. A detail but you also call it LYI (capital I) here while it should be LYl (lower case L)!
*Response: We have decided to call the life year index (LYI) as an abbreviation for Life years lost. We will check for any inconsistency in the manuscript and correct it.*

17. L227: You discuss your first results here on the F score and model performance discussion – this should come under Results and Discussion respectively, not Methods! Figure 6 as well as Table 1 also lacks a description of variables and results presented. Unclear how this should be interpreted.

*Response: Thank you for this comment. We will relocate this part to the results and discussion section. Also, we will add more explanations to make it clear for the reader.*

18. L248: Apart from the Brakenridge citation not having a date nor being present anywhere in the references, and agreeing that in principle such a dataset would be an interesting set for validation, the fact that the whole dataset only has 46 events from Nepal since 2021 and <10 with the 1000 deaths plus displaced criterium you introduce below makes its use questionable

considering this is the area you run your model in. Wouldn't data from Nepal (like https://bipadportal.gov.np/) be much more appropriate then?

*Response: We have done our study for both Nepal and HMA from 1980-2020. To our best understanding, the dataset from emdat and DFO have the longest and most detailed series of point datasets for different events for the time period we are interested in. We appreciate the separate data source that you shared with us. Please note that we trained our model considering information for NEPAL from http://www.drrportal.gov.np/ which includes flood/heavy rain/flash flood events for all districts in Nepal from different sources. This database includes more than 46 events reported in the DFO. At the scale of HMA, there is no other available dataset reporting flood impacts, aside from DFO and EMDAT, to our knowledge, hence we considered these two, with their limits, to highlight how our model could help target priority areas where indeed events have happened, of a large impact, as highlighted by actual floods reported in these two independent datasets.*

19. Also this database captures lowland floods, rather than mountain floods, making me wonder whether the aim to characterize 'High Mountain Asia' floods is really the right scope here. Also the DFO reports single coordinates, are you then simply assuming the watershed that matches the coordinate is the only one affected? Likely the reported numbers refer to much larger areas, as the size of the watershed you chose is rather small (guessing from the Figure, it's not actually described anywhere!)

*Response: The dataset is not only capturing lowland floods but also mountainous river floods that are characterized as fluvial floods. Also, the damage dataset can only be "point" data at a particular location. There may be one, many, or no point for the whole watershed. As we have described previously, we have used GIS techniques to distribute the damages for the watersheds. This is a common technique that is used widely.*
*We will add information on the size of the watershed. (e.g., range of the watersheds' area)*

20. L261: You include a crucial boundary condition of your model here, i.e. '1000 deaths plus displaced'. Does this mean your model will only be useful in this domain? It would be crucial to report how many such events have actually happened in your domain then. Also how is the adding up of 'dead and displaced' justified? These are quite 'different' responses to a flood.

*Response: In the LYI calculation, the formula refers to the number of affected people, without making differences between deaths or displaced. Hence we considered the number of people from the DFO as a proxy. If there were fine scale datasets with this complete information, we would have validated the model outside Nepal for those datasets, but unfortunately, this information is not available at a fine scale, but only for the events reported in the DFO/EMdat. Please note that the paper by Noy et al, where the life year is calculated globally country-wise, considers the EMDAT source for the analysis, so numerically it reports the life year lost based on the available numbers. Regarding the ML model we propose, the model itself is bounded by the data it ingested. The training set contains a variability of LY lost, that is in the order of magnitude of 10^0 up to 10^3 life year lost, consistent with the examples provided in Noy et al. 2016 globally.*

21. Figure 9: Panel a is elevation not rainfall as your legend suggests!

*Response: There was a mistake in the legend. The rainfall in the yellow contour lines overlayed on the elevation. The yellow line is not showing up in the legend. We will correct this.*

22. L265: To be honest I am not entirely sure how I should interpret Figure 7 – doesn't it just confirm that people live close to wide river channels? Then there is really no link to atmospheric

characteristics as you claim in L270. There is a lot of discussion already as well on convection patterns all stemming from other literature and not really relevant to what I read in the Figure.

*Response: In Figure 7, the main target was to analyze the variability of average CI and population over the time frame (1980-2020) of the study compared to the calculated static FGP. FGP is calculated using DEM dataset solely based on elevation. Higher FGP does not mean wider channels, rather it gives you the areas with higher potential to flood. In Figure 7, we tried to draw a correlation between the CI and FGP in areas with a higher potential to flood. We did the same with the population.*

[Figure]

**Figure 7: Average variability of the CI (top) and population (bottom) compared to FGP from 1980-2020**

23. L295: A main concern I have here is that I am still not very clear on where the observed events come from you compare this to. I am also wondering if your Figure 8 simply only confirms one thing – that there are many people (an input to your model) where there are many people (a validation of your model). How does your model compare on actually coming up with an observed flood from the input 'ERA5 rain'? This concern then propagates into the result for the whole region, where you 'predict' the biggest impacts with the highest population densities. That isn't quite so surprising and it is unclear to me how I can see the power of ML in these results. To be provocative, would the results have been different if you would have just distributed rainfall across the watersheds without a model in between?

*Response: The observed/ actual values of LYI of the remaining 10% dataset for Nepal. We will be clearer in the manuscript about this.*
*Figure 8 showcases the comparison of LYI, not people or flood. WE compare the LYI provided by our model, to the LYi calculated numerically from the data included in the drrportal for NEPAl.*

[Figure]

**Figure 8: Comparison of prediction with actual socioeconomic impact for watershed on Nepal. Basin/districts are marked as "high" for LYI over 1000 years. Medium is between 100 and 1000, and low is less than 10. Numbers in parentheses represent accuracy.**

24. L349: The figures you note here do not show what is described in the text.

*Response: We will correct this mistake.*

25. L356: I lack some context here - <10% of watersheds see an increase, are all other stable or see a decrease?

*Response: We will revise and clarify this part in the manuscript.*

26. How can you differentiate here between hazard (rain) and exposure (population) as a driver of change?

*Response: We do not attempt to differentiate between hazard and exposure. Rather we use them together to find out the impact. We simply tried to analyze the results and connect the dots by revisiting past occurrences (such as population boom, extreme events, or both).*

27. How do you explain that increase has slowed after 2010 significantly? And how is it possible that in the 1995-2010 jump the number of increasing watersheds is similar to the just 5 year jump between 1990 – 1995? Isn't that completely counterintuitive?

*Response: It may be counterintuitive, however, there may have been many events that have caused more damage in that 5-year window than the longer span.*

28. L406: While in general 'an intention to make data available' shouldn't be followed, for a journal like NHESS this is definitely not acceptable. Data availability needs to be clearly described (or arhued why this is not the case).

*Response: At the time of submission, we were working towards publishing the datasets on NSIDC. It must be published eventually as a requirement of the HiMAT project. This dataset will be publicly available. Before the data is published it goes through a quality check, hence why the link is not available yet. The input data themselves are available through the various portals we considered. In the revised paper we will better clarify the data source for each independent dataset considered, to avoid confusion on its accessibility.*

29. Technical corrections (Minor issues):

*Response: We will carefully go through all the minor comments and correct the mistakes.*

*References:*

*Du, M., Huang, S., Leng, G., Huang, Q., Guo, Y., & Jiang, J. (2023). Multi-timescale-based precipitation concentration dynamics and their asymmetric impacts on dry and wet conditions in a changing environment. Atmospheric Research, 291, 106821. https://doi.org/10.1016/J.ATMOSRES.2023.106821*

*Khanal, S., Tiwari, S., Lutz, A. F., Hurk, B. V.D., & Immerzeel, W. W. (2023). Historical Climate Trends over High Mountain Asia Derived from ERA5 Reanalysis Data. Journal of Applied Meteorology and Climatology, 62(2), 263–288. https://doi.org/10.1175/JAMC-D-21-0045.1*

*Maggioni, V., & Massari, C. (2018). On the performance of satellite precipitation products in riverine flood modeling: A review. Journal of Hydrology, 558, 214–224.* [https://doi.org/10.1016/J.JHYDROL.2018.01.039](https://doi.org/10.1016/J.JHYDROL.2018.01.039)

*Maina, F. Z., Kumar, S. V., Getirana, A., Forman, B., Zaitchik, B. F., Loomis, B., Maggioni, V., Xue, Y., McLarty, S., & Zhou, Y. (2023). Development of a Multidecadal Land Reanalysis over High Mountain Asia. AMS. https://ams.confex.com/ams/103ANNUAL/meetingapp.cgi/Paper/415850*

*Delalay, M., Ziegler, A. D., Shrestha, M. S., Wasson, R. J., Sudmeier-Rieux, K., McAdoo, B. G., and Kochhar, I.: Towards improved flood disaster governance in Nepal: A case study in Sindhupalchok District, International Journal of Disaster Risk Reduction, 31, 354–366, https://doi.org/10.1016/j.ijdrr.2018.05.025, 2018.*

*Monjo, R., & Martin-Vide, J. (2016). Daily precipitation concentration around the world according to several indices. International Journal of Climatology, 36(11), 3828–3838. https://doi.org/10.1002/JOC.4596*

*Noy, I.: A Global Comprehensive Measure of the Impact of Natural Hazards and Disasters, Glob Policy, 7, 56–65, https://doi.org/10.1111/1758-5899.12272, 2016.*

*Saki, S. A., Sofia, G., & Anagnostou, E. N. (2023). Characterizing CONUS-wide spatio-temporal changes in daily precipitation, flow, and variability of extremes. Journal of Hydrology, 626, 130336. https://doi.org/10.1016/J.JHYDROL.2023.130336*

*Sofia, G. and Nikolopoulos, E. I.: Floods and rivers: a circular causality perspective, Sci Rep, 10, https://doi.org/10.1038/s41598-020-61533-x, 2020.*

*Sofia, G., Ragazzi, F., Giandon, P., Dalla Fontana, G., & Tarolli, P. (2019). On the linkage between runoff generation, land drainage, soil properties, and temporal patterns of precipitation in agricultural floodplains. Advances in Water Resources, 124, 120–138.* https://doi.org/10.1016/j.advwatres.2018.12.003

*Sofia, G., Tarolli, P., Cazorzi, F., and Dalla Fontana, G.: Downstream hydraulic geometry relationships: Gathering reference reach-scale width values from LiDAR, Geomorphology, 250, 236–248, https://doi.org/10.1016/j.geomorph.2015.09.002, 2015.*

*Manfreda, S., Di Leo, M., and Sole, A.: Detection of flood-prone areas using digital elevation models, Journal of Hydrologic Engineering, 16(10), 781–790, http://dx.doi.org/10.1061/(ASCE)HE.1943-5584.0000367, 2011.*

*Manfreda, S., Nardi, F., Samela, C., Grimaldi, S., Taramasso, A. C., Roth, G., and Sole, A.: Investigation on the use of geomorphic approaches for the delineation of flood prone areas, Journal of Hydrology, 517, 863–876, https://doi.org/10.1016/j.jhydrol.2014.06.009, 2014.*

*Manfreda, S. and Samela, C.: A digital elevation model-based method for a rapid estimation of flood inundation depth, Journal of Flood Risk Management, 12(Suppl. 1), e12541 1–10, https://doi.org/10.1111/jfr3.12541, 2019.*

*Samela, C., Albano, R., Sole, A., and Manfreda, S.: A GIS tool for cost-effective delineation of flood-prone areas. Computers, Environment and Urban Systems, 70, 43–52, A GIS tool for cost-effective delineation of flood-prone areas. Computers, Environment and Urban Systems, 2018.*

*Samela, C., Manfreda, S., De Paola, F., Giugni, M., Sole, A., and Fiorentino, M.: DEM-based approaches for the delineation of flood-prone areas in an ungauged basin in Africa, Journal of Hydrologic Engineering, 21(2), 06015010, https://doi.org/10.1061/(ASCE)HE.1943-5584.0001272, 2016., Journal of Hydrologic Engineering, 21(2), 06015010, https://doi.org/10.1061/(ASCE)HE.1943-5584.0001272, 2016.*

*Samela, C., Troy, T. J., & Manfreda, S.: Geomorphic classifiers for flood-prone areas delineation for data-*

*scarce environments, Advances in Water Resources, 102, 13–28,
https://doi.org/10.1016/j.advwatres.2017.01.007, 2017., Advances in Water Resources, 102, 13–
28, https://doi.org/10.1016/j.advwatres.2017.01.007, 2017.*

---

## Author Comment (AC2)

The manuscript "Predictive understanding of socioeconomic flood impact in data scarce regions based on channel properties and storm characteristics: Application in High Mountain Asia(HMA)" by Khanam et al. used LYI and ML methods to evaluate and predict the flood impacts and risk due to precipitation in HMA. This work is first time to evaluate socioeconomic impacts of flood hazards in data scarce region. However, it is not good writing. The structure is not reasonable. And the XGboosting tools is not clear to solve what? Thus, I would suggest it should be major revision.

*We want to thank the reviewer for the insightful comments and recommendations. Following the suggestions, we will revise the manuscript. We will try to explain the methodologies clearly and improve the writing.*
*Detailed responses to the reviewers' comments are added below.*

**Note: Below is our response (italics) to each comment (regular font) from the reviewer**

General comments:

- In HMA there are also GLOF which risk the human being and infrastructure. If possible, please include evaluating the socioeconomic flood impact.

**Response:** *We thank the reviewer for this suggestion. In general, this paper focuses on fluvial and pluvial flooding, and we will make this clearer in the introduction. GLOFs in general are triggered by glacial melt but here we focus on a climatic driver of the flooding that is related to rainfall. Addressing the damages due to GLOF is separate from the scope of this study.*

- Data-Scarce regions should be clear (which data or which type of data). In HMA, population is scarce. And Socio activity is also low.

**Response:** *We thank the reviewer for this comment. We will revise this part of the manuscript and rephrase our statements to avoid confusion about data scarcity. Regarding population and activity, we use population density in our work as a proxy of exposure, which varies across the region.*
Specific comments

1.Section 2.2 methods. This title is not reasonable. Is section 2.3(Machine learning model) methods? In addition, the dataset and methods in this section should be divided, for example, 2.2.4 exposure(population) is datasets.

**Response:** *Thank you for the suggestions. We will try to reorganize and revise the sections as per the reviewer's comments.*

2.Line 135 Why is it classified by LYI values(<2,2-3, and >3).

**Response:** *We have classified the LYI as a basis for comparison across all the watersheds and periods. The three groups correspond to <100, between 100-1000, and over 1000.*

3.Line 217 While XGBoosting is …, this sentence is incomplete.

Line 217 this section (machine learning model) is a little difficult to understand the role that is plays.

**Response:** *We will revise this part of the manuscript thoroughly to clarify the methodology and explain the reasonings plainly.*

---

## Author Comment (AC3)

*We want to thank the reviewer for the insightful comments and recommendations. Following the suggestions, we will revise the manuscript. We will try to explain the methodologies clearly and improve the writing. Detailed responses to the reviewers' comments are added below.*

**Note: Below is our response (italics) to each comment (regular font) from the reviewer**

1. Clarity and Structure: The abstract is well-structured, presenting the problem, the proposed solution, and a few findings. However, some sentences are complex (starting from the title), and more concise wording could enhance clarity,

**Response:** *We thank the reviewer for this comment. We will revise the complex sentences to enhance their readability.*

2. Methodology: The use of the Lifeyears Index (LYI) as a measure for socioeconomic flood impact is well explained. It would be beneficial to provide a brief explanation of how the geomorphologically guided machine learning approach works, even if it is in a bit summary.

**Response:** *We thank the reviewer for this comment. We will revise the methodology to be clearer on the geomorphologically guided machine learning approach with the following information-*

*Integrating geomorphological knowledge into the interpretation of machine learning models to better understand the relationships between input variables and the predicted outcomes can be crucial. This may involve analyzing feature importance, spatial patterns, and model outputs in the context of geomorphological processes. In this study, we have introduced Flood Geomorphic potential (FGP) as a variable in the ML model to predict socioeconomic flood impact. Flooding is directly related to the climate and geomorphologic processes, thus incorporating FGP together with the climate index in the analysis allows our model to be physically driven. The main advantage of the approach is that it is based on an automatic technique that relies on globally available datasets thus offering the opportunity to apply this to different areas.*

3. Data: The abstract mentions training the model with over 6000 flood events from 1980 to 2020, but it is mentioned that the model shows variability from 1980 to 2022 as temporal Coverage. So, /what's the correct timeline?

**Response:** *Thank you for pointing this out. There is a mistake and both of them should be the same. It should be 1980 to 2022.*

4. Conclusion: A brief conclusion summarizing the main contributions and implications of the study would be beneficial.

**Response:** *Thank you for the suggestions. We will revise the conclusion and discuss the main contributions and implications of the study more elaborately as per the reviewer's comments.*

---

## Author Response (AR1)

*First, we want to thank the reviewers for their insightful comments and recommendations. Following the suggestions, we have revised the manuscript thoroughly incorporating all the suggested changes.*
**Below is our response (italics) to each comment (regular font) from the reviewers.**

**Review by RC1:**

General comments

The study has merit in its outline, as a better understanding of flood impacts at this scale is definitely needed and I agree that doing this with a generally simple approach for a general kick start to the options of AI in this domain makes sense. However, I have a number of concerns on how the study is built and executed, which I believe are at this stage too big to recommend the manuscript for publication. I detail the concerns below and would encourage the authors to rethink their strategy before moving to an eventual submission. I fully understand that this is a submission from an ECR and I want to complement you on the aim and pulling this together – definitely work that should be pursued and there is a lot of demand for outcomes of such approaches! I would have hoped to see more scrutiny here before a submission from the more experienced co-author team.

My concerns range from (a) general sloppiness of manuscript writing (many simple editing mistakes that can always happen for drafts but should not occur for a submitted manuscript over (b) the lack of appreciation of existing data and simply depicting the target region as 'data scarce' to avoid scrutiny from what is known already to (c) a lack of proper documentation of data sources on the exposure side as well at times confusing jumping between topical (what types of floods) as well as spatial (national, watershed, HMA wise) domains.

**Response:** *We thank the reviewer for his/her in-depth and critical review. We revised the manuscript thoroughly to correct these unnoticed mistakes and improve the writing quality. Regarding the missing references and low quality of the text and figure captions, we did a thorough review of the manuscript to correct all these errors.*

*We would like to note that we defined HIMAT as 'data scarce' because of the complexity of the HMA region challenged by the scarcity of ground observations covering consistent timeframes homogeneously, as highlighted by various works in literature (Barandun et al., 2020; Dollan et al., 2024; Miles et al., 2021). We understand that some of the statements were phrased in an ambiguous way, which we have revised to address accordingly.*

*Regarding the lack of proper documentation, in the revised paper we provided a table (Table 1) with a detailed description of the datasets considered in this study.*

*We further discussed different flood disasters happening in the HMA region. Specifically, our study focuses on pluvial and fluvial flooding, which we made clearer in the introduction and throughout the manuscript.*

*Line 139-150: The analysis follows a multistep approach, beginning with data at both watershed and district scales. Initially, the focus was on the district scale, as socioeconomic data for Nepal, selected as primary training ground, were readily available at this level through the Nepal Disaster Risk Reduction Portal (http://drrportal.gov.np/). For this region, furthermore, there is a comprehensive coverage of high-resolution (8-meter) Digital Elevation Models (DEMs) from prior High Mountain Asia (HMA) work (High Mountain Asia 8-meter DEMs Derived from Along-track Optical Imagery, 10.5067/0MCWJJH5ABYO). Subsequently, all the information is aggregated at the watershed scale, as phenomena such as fluvial and pluvial flooding occur at this level, necessitating a dataset tailored to this scale. To transfer the demographic information from the district to the watershed scale, we performed a weighted spatial join between the watersheds and districts. For each watershed, we attributed the statistical characteristics of the intersecting districts, with weights based on the overlapping areas. Generally, the districts in Nepal are smaller in extent compared to the various watersheds.*

*For the work, furthermore, we provided a quality assessment of the model performance at the watershed and the district scale. The results highlighted that the watershed scale proves more accurate than the district-scale training. We clarified this in the method description.*

**Please consider also our response to your detailed comments in the following paragraphs.**

a) General statement on 'no data'
You make general unsupported statements on the region being data scarce on hydromet data. That is decidedly not the case. While data may often not be readily accessible, it is available and many studies have been published on this, especially for China and India and data is generally reachable from China, India, Pakistan, Afghanistan as well as Central Asian states. The data that is available you do away with as 'not trustworthy' in a single sentence. This coming from an all-US based author team is problematic and I guess you could imagine how stunned a reviewer from the US (or Europe) would be if a Chinese author would make that claim before proceeding to apply ML on all of the US or Europe. You will need to make a clearer description (with references) on what is lacking and how your approach fills that gap.

**Response 1:** *We have revised the manuscript thoroughly to correct these unnoticed mistakes and to improve overall the quality of the writing. Please note that when we defined HIMAT as 'data scarce' we did not do so to avoid scrutiny of the data, but rather to highlight the complexity of HMA itself, challenged by several factors, including the scarcity of ground observations covering consistent timeframes homogeneously, as highlighted by other works in literature (Barandun et al., 2020; Dollan et al., 2024; Miles et al., 2021). We understand that some of the statements were phrased in an ambiguous way, which we have revised to address accordingly. The main idea behind this paper was to provide a tool for a rapid estimate of potentially highly impacted areas, based on information accessible and updateable quickly, such as population number, rainfall intensities, and a geomorphologic index that can be derived from global DEMs (or high-resolution local ones when available).*
*Please consider the revisions made from line 51-90 as follows.*

*Line 51-90: Accurate evaluation of the socioeconomic impacts of natural disasters is paramount to mitigate the sufferings of the affected people and rehabilitation (Cavallo & Noy, 2010; Meyer et al., 2013; Noy, 2015, 2016a). To date, available studies (Diehl et al., 2021; Mohanty & Simonovic, 2022; Pangali Sharma et al., 2019; Pervin et al., 2020; Piacentini et al., 2020; Yang & Tsai, 2000) have primarily concentrated on vulnerability mapping and risk analysis, employing case studies and descriptive event-based methodologies at a local level. Scaling up the analysis over the entire HMA region is indeed a difficult task, as it requires collecting data from several countries and multiple sources, and this poses challenges due scarcity of ground observations covering consistent timeframes homogeneously (Barandun et al., 2020; Dollan et al., 2024; Miles et al., 2021). Especially in the context of the impact of floods using socioeconomic data, the analysis involves examining the number of fatalities, injured and people otherwise affected, as well as the financial damage that natural disasters cause, and this information is generally collected at the local scale based on reported events. Significant disasters are documented in global databases like The International Disaster Database (EMDAT, www.emdat.be) or, as an example for HMA and this study, the Nepal Disaster Risk Reduction Portal (http://drrportal.gov.np/). However, these databases typically operate at a global or national level resolution, potentially overlooking minor disasters. For example, EMDAT only considers events with at least one of the following criteria: 1)10 fatalities; 2)100 affected people; 3) a declaration of state of emergency; 4) a call for international assistance. Additionally, those databases utilized to support insurance may prioritize countries with existing or potential insurance coverage (World Bank, 2012).*
*The integration of geomorphic properties, population data, and rainfall characteristics for assessing*

*socioeconomic flood impact is seldom explored comprehensively on a large scale. For HMA. this is primarily due to the inherent challenges associated with conducting on-site surveys in rugged and often inaccessible terrain. However, leveraging remote sensing data has emerged as a valuable approach for delving deeper into these dynamics and effectively quantifying flood impacts. Modern global datasets, featuring improved resolution and coverage, further enhance the utility of remote sensing in this regard (Diehl et al., 2021; Jongejan & Maaskant, 2015; Mosavi et al., 2018; Bentivoglio et al., 2022; Mazzoleni et al., 2022; Hawker et al., 2018; Kirschbaum et al., 2020; Mohanty and Simonovic, 2022; Pangali Sharma et al., 2019; Sanyal and Lu, 2004; Yang and Tsai, 2000; Zheng et al., 2018).*

*Furthermore, machine learning (ML) techniques have emerged as increasingly popular tools in advanced prediction systems over the past two decades. They offer more cost-effective solutions with performance that can be aggregated, surpassing the complexity and time demands associated with simulating the complex development of flood processes. Recent research (Bentivoglio et al., 2022; Deroliya et al., 2022; Mosavi et al., 2018) has showcased encouraging advancements by integrating machine learning (ML) techniques with global datasets. This contemporary approach to mapping flood vulnerability notably streamlines the computational processes associated with data-intensive simulations, enhancing flood risk management strategies. However, ML systems rely on existing data for learning. Insufficient or incomplete data coverage can hinder effective learning, leading to suboptimal performance when deployed in real-world scenarios. Therefore, ensuring robust data enrichment, encompassing both quantity and quality, is imperative.*

*In this study, we introduce a streamlined methodology for preliminary flood vulnerability assessment on a large scale, leveraging available global datasets. Specifically, we introduce a flood-risk assessment model designed to quantify spatially distributed socioeconomic susceptibility in flood-prone regions. We utilize this model to augment disaster understanding by integrating remotely sensed data, including climate variables and high-resolution terrain information.*

*Finally, we apply this model in the High Mountain Asia (HMA) regions to analyze changes in socioeconomic flood impacts spanning from 1980 to 2020.*

b) Poor documentation of socio-economic data
   As I detail below there is very poor documentation on where the exposure data is taken from and there is no way to make this traceable (no stable links, and also no attempt so far to make your own produced data available, see comment on Availability statement).

**Response 2:** *We thank the reviewer for this comment. We added a table with information on all the datasets used. Regarding the links being "not stable" – the data required for the index were accessed and we tested the links before submission. In the revised paper, we added the date of the latest access so that the data is more clearly referenced. We further added in the table the 'standard' value we adopted, with the reference for each value.*

**Table 1: Parameters used to calculate LYI**

| Variable | Description | References |
|---|---|---|
| M | Mortality (number of deaths due to disaster | *Nepal Disaster Risk Reduction Portal* http://drrportal.gov.np/ |
| Aexp | Average life expectancy at birth (by year) | WHO ([https://data.who.int/countries/524](https://data.who.int/countries/524)) |
| Amed | Median age (by year) | WHO ([https://data.who.int/countries/524](https://data.who.int/countries/524)) |
| e | Welfare reduction weight associated with being exposed to a disaster | set to e = 0.054 according to Noy, (2016a) , based on Mathers et al., 2013 |
| T | Time taken by the affected person to get back to normal | Noy, (2016a) |

| | | |
|---|---|---|
| N | Number of affected people | http://drrportal.gov.np/ |
| c | Percent of time not used in work-related activities (.75) | Noy, (2016a) |
| Y | Y = Financial damage (value of destroyed/damaged infrastructure) | http://drrportal.gov.np/ |
| PCGDP | Income per capita (by year) | http://drrportal.gov.np/ |

c) I also fail to see how you take census data to the watershed and how you align using Nepal government data with your approach to model at the watershed scale (which do not follow national borders).

**Response 3:** *We added more details in the manuscript. Please see below-*

*Line 139-150: The analysis follows a multistep approach, beginning with data at both watershed and district scales. Initially, the focus was on the district scale, as socioeconomic data for Nepal, selected as primary training ground, were readily available at this level through the Nepal Disaster Risk Reduction Portal (http://drrportal.gov.np/). For this region, furthermore, there is a comprehensive coverage of high-resolution (8-meter) Digital Elevation Models (DEMs) from prior High Mountain Asia (HMA) work (High Mountain Asia 8-meter DEMs Derived from Along-track Optical Imagery, 10.5067/0MCWJJH5ABYO). Subsequently, all the information is aggregated at the watershed scale, as phenomena such as fluvial and pluvial flooding occur at this level, necessitating a dataset tailored to this scale.*

*To transfer the demographic information from the district to the watershed scale, we performed a weighted spatial join between the watersheds and districts. For each watershed, we attributed the statistical characteristics of the intersecting districts, with weights based on the overlapping areas. Generally, the districts in Nepal are smaller in extent compared to the various watersheds.*

*In general, the districts we have for Nepal are of a smaller extent than those of the various watersheds.*

d) General scope and methodology
At multiple points of the manuscript I was a bit confused on the scope. There is an introduction on all types of high flow events but the methods suggest you only look at fluvial floods with exceptionally high impacts. There is a relatively rapid investigation of the methods for watersheds that do lie to some part in Nepal compared against data only from areas within Nepal and then an upscaling to all of HMA, which in turn is not clearly defined in its scope or climatologies. I would strongly suggest to maybe limit the study to areas where data is available before scaling it up, allowing you more space for methodological and data based issues.

**Response 4:** *We thank the reviewer for this suggestion. This paper focuses on fluvial and pluvial flooding. In the revised work, we reworked parts of the introduction to be clearer about this.*

*Please refer to **Response 3** for the explanation as to why we chose the watershed scale for the analysis.*

*Regarding the scope and climatologies, we agree with the comment, and revised the text to provide a wider context.*

**Line 222 and subsequent:** *The climatology in HMA is highly variable (Dollan et al. 2024). Summer monsoons drive precipitation in the Ganges-Brahmaputra basins and the Tibetan Plateau ( Bookhagen and Burbank, 2010; Shamsudduha and Panda, 2019); synoptic storms dominate winter precipitation impacting areas in the northwestern Karakorum mountains (Winiger et al., 2005; Barlow et al., 2005). Overall, as well, variations in elevation gradients contribute to diverse microclimates, exemplified by Nepal's swift transition from high mountains to lowlands (Kansakar et al., 2004; Karki et al., 2016). Winter precipitation in the area is primarily influenced by the westerly weather system, with western disturbances*

*originating in the Mid-Atlantic or Mediterranean Sea and traversing through northwest India to western Nepal after passing over Afghanistan and Pakistan (Kansakar et al., 2004; Hamal et al., 2020). In Nepal, which was used as the training site for the model, regional climate variations exist, mostly driven by changes in elevation, with an overall homogeneity in trends (aside from a few hotspots) and regional statistics of precipitation, in line with the variability of HMA, as highlighted by the recent study by (Khanal et al., 2023).*

*For this work, for the main rainfall driver of the model, we focused on daily climate concentration. As climate concentration values are mostly related to the temporal variability of the rainfall, not to the total amount or the average yearly and seasonal statistics, using this index allows to capture well various climates globally (Monjo and Martin-Vide, 2016a). The variability of climate concentration, furthermore, has been proven to be highly linked to pluvial/fluvial flooding impacts in various regions of the world, including for example Italy (both in mountainous landscapes and floodplains (Sofia et al., 2019), the US (Saki et al., 2023) [over a variety of physiographic regions], or China (Du et al., 2023). Different authors have adopted different methods to determine the temporal concentration of precipitation, and the Concentration Index (CI) (Equation 2) is one of the most used parameters (Caloiero et al., 2019; Martin-Vide, 2004; Monjo, 2016; Sangüesa et al., 2018; Serrano-Notivoli et al., 2018).*

*We decided to use Nepal as the training site as it represents rapid variations in climatologies over a smaller extent, for which we had complete coverage of good-quality data. For Nepal, as we showed in the paper, we have a gradient of CI values, and as ML models learn from the data they ingest, we believe the system can work across various regions from the climatic point of view. We added comments on this in the paper, highlighting the strengths and weaknesses of the approach.*

**Response to Specific comments:**

1. L31: Be careful in your framing – population growth does not increasing likelihood of flooding, it increases flood risk!

*Response 5: We rephrased this in the revised manuscript. The sentence was meant to refer to flood impacts.*

2. Also, in the abstract and your general analysis, you focus on precipitation as a flood driver but here then passingly mention glacial melt as well – those are very different flood drivers and would be crucial to be clear what kind of flooding you wish to tackle here.

*Response 6: We tried to be clear in the revised manuscript on the fact that this work focuses on pluvial/fluvial flooding. We added background information on flood hazards in general in the region, as we believe some overview of this is needed to frame the work correctly in the context of the various possible flood hazards in the region.*

3. L54: '*HMA does not have enough hydrological stations for region-wide flood monitoring*' is a huge statement to make without a citation – what is an appropriate number? Also most countries in HMA, especially China, India, Pakistan and Nepal have large and dense network of hydro(-met) monitoring, which they also use for forecasting. That is not as open as in the US, but the statement that there is 'not enough' needs to be qualified. You then claim '*Moreover, the available meteorological datasets may not be sufficiently trustworthy.*', which again lacks any qualification. Imagine me making that statement for a European or North American country, that would be thrown out. The region has a large amount of met data (see e.g. the overview figure in (Nepal et al. 2023)) and if you do not trust the data you need to justify why.

*Response 7: As mentioned earlier, we revised this part of the manuscript and rephrased our statements. It was not our intention to underrepresent available datasets. Gathering data for the scale of the HMA*

*region requires collecting data from several countries. This is time-consuming and to some extent, it can be nearly impossible due to political constraints and limits on accessing the data for some regions. As a matter of fact, the goal of this study is to overcome the data and time constraints and provide a quick tool for disaster management. Also, please refer to the response regarding the datasets available throughout the project's scope.*

4. L61: '*The use of remote sensing technology for disaster studies in HMA is comparatively new*' – I also do not quite agree. Remote sensing itself isn't very old and it has been used in HMA for many studies already (which maybe anyway would need some acknowledgement here).

***Response 8:***
*We want to reiterate our intention was not to undermine the remote sensing data used in HMA. Instead, we want to emphasize that Remote Sensing data usage particularly in disaster studies in HMA is comparatively modern. We rephrased this in the manuscript as follows,*
*Line 68: The integration of geomorphic properties, population data, and rainfall characteristics for assessing socioeconomic flood impact is seldom explored comprehensively on a large scale. For HMA. this is primarily due to the inherent challenges associated with conducting on-site surveys in rugged and often inaccessible terrain. However, leveraging remote sensing data has emerged as a valuable approach for delving deeper into these dynamics and effectively quantifying flood impacts. Modern global datasets, featuring improved resolution and coverage, further enhance the utility of remote sensing in this regard (Diehl et al., 2021; Jongejan & Maaskant, 2015; Mosavi et al., 2018; Bentivoglio et al., 2022; Mazzoleni et al., 2022; Hawker et al., 2018; Kirschbaum et al., 2020; Mohanty and Simonovic, 2022; Pangali Sharma et al., 2019; Sanyal and Lu, 2004; Yang and Tsai, 2000; Zheng et al., 2018).*

5. L87: You focus here a lot on monsoon changes with intense precipitation – but if you actually focus on HMA (rather than just the Hindukush Himalaya) there are a lot of other processes – Westerlies in Central Asia, Eastern Monsoons in the Upper Yangtze etc. Maybe it is required to reconsider the total spatial scope of the study here?

***Response 9:*** *We wanted to draw the focus on the fluvial/pluvial flooding caused by the precipitation. In the revised manuscript we provided a more detailed discussion of the climatology of the area, as mentioned in our previous response. Please see **Response 4** for the exact changes we made.*

6. L92: You now finally get to actual numbers of potential affected, but leave it to the reader to get the data from EMDAT.It would be prudent to explain here (or rather in the introduction) what the actual numbers are and for what types of hazards, to then narrow down and which ones you actually focus.

***Response 10:*** *The EMDAT link was used as a reference to the statement, rather than as an actual dataset. We added the numbers calculated using EMDAT data. This was not provided to direct the reader to get the data from EMDAT.*

6. Figure 1: Up to this point there was no clear description how the watersheds are selected, i.e. what boundary you used for HMA. This needs to be provided to give context to why so many watersheds outside HMA are also included.

***Response 11:*** *We have added the following to the manuscript.*
*Line 108-114: This study considers approximately 6,000 watersheds across HMA as main target area (Figure 1): the watershed were selected to be consistent with the HMA domain and all the datasets produced throughout the different phases of the NASA-funded HiMAT project (https://himat.org/). The analysis initially centered on training and testing a machine learning model specifically for Nepal. To achieve this, we collected fine-resolution topographic data along with district-scale socioeconomic*

*information pertaining to population characteristics and documented flood impacts for this region. Subsequently, leveraging the insights gained from this initial phase, we extended the application of the trained model to predict socioeconomic impacts across all watersheds in HMA.*

7. L116: At this point you mention that you will predict impacts of 'floods', i.e. all of them? The way you describe your research you are narrowing this down on pluvial floods, as glacial lake outburst floods or debris flows etc need very different driver analysis. Can you be precise here? In L185 you then suddenly just focus on 'fluvial flooding', so is it just that you focus on?

**Response 12:** *Thank you for this comment. We tried to be consistent with the terminology. And we have used only fluvial and pluvial flooding in the analysis.*

8. L120: This part is crucial as you present the socioeconomic data and how you treat it. However there are a few issues that would need to be addressed with respect to traceability and presentation of data used.

   - You refer to data sources that are questionable, the knoema.com page is not stable and it is unclear from where their data is sourced or where it is known needs to be documented here.
   - You refer to general government and Worldbank websites (like http://drrportal.gov.np/) that exist but what data you took from there at what point in time remains unclear. Copernicus Journals subscribe to FAIR practices, that includes the documentation of third party data used in a publication.
   - You introduce a lot of data as well as parameters from literature (like T and e) without any questioning of their accuracy, uncertainty etc. This would propagate and need to be addressed, especially as you seem to upscale from this approach with a few numbers on Nepal government websites to all of HMA.
   -

**Response 13:** *We thank the reviewer for this comment. Please note that in the revised paper we added a table with the data sources used for the work and a reference to all the literature from where we took any 'standard value' used in the calculation. We further checked the links and added a reference date for access. Please note that in the revised manuscript, given the comment regarding some of the parameters, we confirmed their value through other sources (WHO).*
*In general, some standard parameters in the LYI formula (like T and e) are suggested by (Noy, 2016) and they were applied in the referenced work to calculate LYI for Nepal as a whole, so we used them consistently. We added the following table to the manuscript where we clarified the data sources considered for the analysis.*

**Table 1: Parameters used to calculate LYI**

| Variable | Description | References |
|---|---|---|
| M | Mortality (number of deaths due to disaster | *Nepal Disaster Risk Reduction Portal* http://drrportal.gov.np/ |
| Aexp | Average life expectancy at birth (by year) | WHO (https://data.who.int/countries/524) |
| Amed | Median age (by year) | WHO (https://data.who.int/countries/524) |
| e | Welfare reduction weight associated with being exposed to a disaster | set to e = 0.054 according to Noy, 2016a, based on Mathers et al., 2013 |
| T | Time taken by the affected person to get back to normal | Noy, (2016a) |
| N | Number of affected people | http://drrportal.gov.np/ |
| c | Percent of time not used in work-related activities (.75) | Noy, (2016a) |
| Y | Y = Financial damage (value of destroyed/damaged | http://drrportal.gov.np/ |

infrastructure)

PCGDP        Income per capita (by year)        http://drrportal.gov.np/

9. You calculate these values for Nepal as a whole but then work on the watershed scale – how is this compatible?

*Response 14: Please note that we did not calculate the values for Nepal as a whole, but rather the LYI is calculated for each flood event, starting from the information provided by the Nepal Disaster Risk Reduction portal. The LYI total value is then summarized for each district as an overall number of LYI for all the events geolocated within that district. To scale it up to the watershed level, we did a weighted spatial join.*

*There are indeed a few parameters that go in the LYI equation that are standard, but this is consistent with the calculation of the index as it was defined by Noy et al. 2016. Most of the other parameters, such as median age, or life expectancy at birth may vary from country to country. As we 'train' the model considering a temporal variability of this parameter, we have enough variability of this parameter to represent a good statistical sample of values. For example, Figure R1 reported below shows the life expectancy over time for Nepal, to show that we do have a variability in time for the index. We will clarify this in the manuscript.*

[Figure]

*Figure R1: Life expectancy at birth for Nepal over time (Source:*
*[https://data.worldbank.org/indicator/SP.DYN.LE00.IN?end=2021&locations=NP&start=1960&view=chart&year=2021](https://data.worldbank.org/indicator/SP.DYN.LE00.IN?end=2021&locations=NP&start=1960&view=chart&year=2021))*

*For Nepal, we considered **a weighted spatial join between the watersheds and the districts.** To each watershed, we attributed the statistics of the district intersecting it, weighted by the overlapping areas. In general, the districts we have for Nepal are of a smaller extent than those of the various watersheds.*

10. Figure 3: I am not sure whether these are now LYI only due to floods or all disasters. Considering that there are no jumps for earthquake events like 2015, I assume this has been calculated for floods only? Then this needs to be made very clear in the caption rather than just calling it 'disasters'.

*Response 15: This is for fluvial and pluvial floods only. We revised the manuscript to be clear and consistent.*

11. L176 + Figure 4: What is HAND in Figure 4. Is this from (Delalay et al. 2018)? The publication is not open access and only limited to Sindupalchowk, how does it go to all of Nepal? What does it actually map?

*Response 16: We thank the reviewer for this comment, we clarified this in the manuscript.*

*Line 205 and subsequent: We identified flood-prone areas by grouping them into six classes by their FGP index. For each watershed, we then considered the areas covered by the classes with FGP greater than 4, which, when compared to published data, proved to correspond realistically with areas subject to floods of about 100-year depth. Figure 4b compares the Flood Geomorphic Potential (FGP) automatic classes derived for select rivers in Nepal, with baseline inundation scenarios evaluated using standard inundation depths associated with critical flood events and their return periods provided in the work of Delalay et al. (2018). This visual comparison serves to highlight the efficacy of flood inundation mapping facilitated by the FGP.*
*It's worth noting that the FGP methodology has been previously published and applied in various contexts (Samela et al., 2017). While testing the quality of the FGP lies beyond the scope of this work, its effectiveness for flood mapping has been well-established in previous studies (Manfreda et al., 2011, 2014; Manfreda & Samela, 2019; Samela et al., 2016, 2018), which have demonstrated the utility of the methodology, particularly in ungauged conditions, for preliminary identification of flooded areas in regions where conducting expensive and time-consuming hydrologic-hydraulic simulations may not be feasible.*

12. L194: While I understand that it would be well beyond the scope of this study to evaluate the suitability of ERA5 data for flood simulations (let alone in a mountain context where precipitation products are of poor quality) but it would be crucial to address this and dispel concerns from the get go by referring to discussions of this data in mountain regions as well as for flood mapping.

*Response 17: This study is a part of the HiMAT project. There are several research groups working on different aspects of HMA and the comparison of various rainfall dataset. Speficically for ERA5, for example, other related works from the HiMAT team, such as Maggioni & Massari, 2018; Maina et al., 2023, Dollan et al. 2024 have analyzed different precipitation products. Dollan et al. 2024 stated that ERA5 although overestimates the monthly precipitation, can capture extreme events quite accurately compared to other products. We added some comments on this in the manuscript.*

13. L210: As for the other socioeconomic data above, the description of population data here remains lacking. For Nepal you only refer to the Census Bureau, which does not report distributed data or data by watershed (so how was that brought in line with inundation maps) and you also do not specify where on the general page you retrieved the data from. You then refer to the GHSL but do not provide a citation or link where this data was retrieved. Distributed data in Asia is generally of problematic and definitely not homeogenous quality, hence a discussion of how this was dealt with need a much more thorough description than the short paragraph here without any references.

*Response 18: We thank the reviewer for this comment. Indeed, obtaining population data at a detailed scale depends on local authorities, and for Nepal, we relied on the official source of the census bureau, which provides this information by the district.*

*Line 262: To extend the model to the whole HMA, we computed the population for each watershed*

*across the region from the Gridded Population of the World (GPW), v4 | SEDAC, 2024) dataset by the Center for International Earth Science Information Network. This dataset provides spatially explicit estimates of population density for the years 2000, 2005, 2010, 2015, and 2020, based on counts consistent with national censuses and population registers, as raster data to facilitate data integration. We used a simple linear regression to retrieve data for the missing years.*

14. A detail but you also call it LYI (capital I) here while it should be Lyl (lower case L)!

**Response 19:** *We have decided to call the life year index (LYI) as an abbreviation for Life years lost. We checked for any inconsistency in the manuscript and corrected it.*

15. L227: You discuss your first results here on the F score and model performance discussion – this should come under Results and Discussion respectively, not Methods! Figure 6 as well as Table 1 also lacks a description of variables and results presented. Unclear how this should be interpreted.

**Response 20:** *Thank you for this comment. We moved this part to the results and discussion section. Also, we will add more explanations to make it clear for the reader.*

16. L248: Apart from the Brakenridge citation not having a date nor being present anywhere in the references, and agreeing that in principle such a dataset would be an interesting set for validation, the fact that the whole dataset only has 46 events from Nepal since 2021 and <10 with the 1000 deaths plus displaced criterium you introduce below makes its use questionable considering this is the area you run your model in. Wouldn't data from Nepal (like https://bipadportal.gov.np/) be much more appropriate then?

**Response 21:** *We thank the reviewer for this comment. We understand that some information might not have been clear from the submitted manuscript, which we improved during revision. Please note that we trained and validated our model considering information for NEPAL from the Nepalese government through their Nepal Disaster Risk Reduction Portal, which includes flood/heavy rain/flash flood events for all districts in Nepal from different sources. This database includes more than 46 events reported in the DFO. Please note that indeed the Bipadportal you suggested, reports information sourced from the DRR Portal, which is what we use to train and validate our model.*

*We provided the analysis of both Nepal and HMA from 1980-2020. To the best of our knowledge, the datasets from EMDAT and DFO have the longest and most detailed series of point datasets for different events for the period we are interested in when scaling up at the HMA level Therefore, we considered these two, with their limits, to highlight how our model could help target priority areas over HMA as a whole, and we showcased that areas highlighted by our model as potentially high-risk areas where indeed indeed affected by high-impact events, as highlighted by floods reported in these two independent datasets.*

17. Also this database captures lowland floods, rather than mountain floods, making me wonder whether the aim to characterize 'High Mountain Asia' floods is really the right scope here. Also the DFO reports single coordinates, are you then simply assuming the watershed that matches the coordinate is the only one affected? Likely the reported numbers refer to much larger areas, as the size of the watershed you chose is rather small (guessing from the Figure, it's not actually described anywhere!)

**Response 22:** *We added the following to our paper. Line 284 and following: To verify our findings, we compared the predictions at the HMA level with flood events reported in the Dartmouth Flood Observatory's (DFO) Global Active Archive of Large Flood Events,*

*1985–Present. This comprehensive database compiles information on major floods sourced from diverse channels such as news reports, governmental records, ground observations, and remote sensing data. Notably, the DFO dataset encompasses various flood types, including lowland floods and mountainous river floods characterized as fluvial and pluvial floods.*

*The dataset provides point locations, representing the centroids of affected areas during floods. While acknowledging that flood centroids may oversimplify the complexities driving flood events, we utilized this dataset to showcase our model's capability to target high-risk locations historically impacted by floods within the specified timeframe. Identifying high-risk areas with recorded flood occurrences centered around these locations underscores the robustness of the model beyond the confines of its training and validation site in Nepal.*

L261: You include a crucial boundary condition of your model here, i.e. '1000 deaths plus displaced'. Does this mean your model will only be useful in this domain? It would be crucial to report how many such events have actually happened in your domain then. Also how is the adding up of 'dead and displaced' justified? These are quite 'different' responses to a flood.

**Response 23:** *In the LYI calculation, the formula refers to the number of affected people, without differentiating deaths from displaced. Hence, we considered the number of people from the DFO as a proxy of socioeconomic impacts. If there were fine-scale datasets with this complete information, we would have validated the model outside Nepal for those datasets, but unfortunately, this information is not available at a fine scale, but only at the country scale from Noy et al., or it would be possible to recalculate the index by disaster, using EMDAT data (For example). Please note that the paper by Noy, 2016, where the life year is calculated globally country-wise, considers the EMDAT source for the analysis, so numerically it reports the life year lost based on the available numbers, with the limitation expressed in the revised paper regarding EMDAt. Regarding the ML model we propose, the model itself is bounded by the data it ingested. The training set contains a variability of LY lost, that is in the order of magnitude of 1 up > 1000 life year lost, consistent with the examples provided in Noy et al. 2016 globally.*

18. Figure 9: Panel a is elevation not rainfall as your legend suggests!

**Response 24:** *There was a mistake in the legend. The rainfall in the yellow contour lines overlayed on the elevation. The yellow line is not showing up in the legend. We have corrected the legend and added more information in the caption.*

19. L265: To be honest I am not entirely sure how I should interpret Figure 7 – doesn't it just confirm that people live close to wide river channels? Then there is really no link to atmospheric characteristics as you claim in L270. There is a lot of discussion already as well on convection patterns all stemming from other literature and not really relevant to what I read in the Figure.

**Response 25:**
*Line 310-335: From this analysis, we can see how the variability of CI is complex. If expectedly, the variability of the index is related to atmospheric characteristics (Sangüesa et al., 2018), the index varies also due to geographical factors influencing climate (Tuladhar et al., 2020). In their study based on Nepal, Karki et al., 2017 highlighted the difference in the spatial pattern of high-intensity storm events from that of annual and monsoon events. The rapid rate at which physical processes (e.g., convection) take place regulates the high temporal concentration of precipitation in the regions where the sea surface and ground are highly affected by warmer temperatures (Monjo & Martin-Vide, 2016b). On the other hand, the low temporal concentration of rainfall is characterized as a normal pattern caused by cyclical weather events (Monjo & Martin-Vide, 2016). Watersheds with lesser floodplain extents (that is, less areas with high FGP) are related to higher and steeper mountains, with complex orography. Research has shown that low areas in Nepal are susceptible to receiving high-intensity storm events even though they have fewer wet days (Karki et al., 2017). The authors of the same study also observed that*

*the low-intensity events (annual and monsoonal precipitation) were mostly predominant over Nepal's western middle mountains and central high mountains. In another study, however, Subba et al., 2019 stated that the frequency of extreme events had decreased significantly over the past two decades in the eastern part of Nepal. For our case, areas having the larger physical potential to floods (high FGP), appear to be areas showing the largest variation in CI, with values ranging from low (0.2) as well as very high (0.75), indicating a potential compound effect of highly torrential rains (CI=0.7) in locations where much of the landscape is potentially floodable. Readers should consider that higher FGP values do not imply locations having wider channels, but rather they indicate how the landscape is potentially more flood-prone as highlighted by (Samela et al., 2017; Manfreda & Samela, 2019; Samela et al., 2016, 2018). In Figure 6, we showcase how for our landscape, areas where we have higher variability of CI (>0.60) correspond to locations with high physical flood potential, as well as larger exposure in population.*

[Figure]

**Figure 6: Average variability of the CI (top) and population (bottom) compared to FGP from 1980-2020**

20. L295: A main concern I have here is that I am still not very clear on where the observed events come from you compare this to. I am also wondering if your Figure 8 simply only confirms one thing – that there are many people (an input to your model) where there are many people (a validation of your model). How does your model compare on actually coming up with an observed flood from the input 'ERA5 rain'? This concern then propagates into the result for the whole region, where you 'predict' the biggest impacts with the highest population densities. That isn't quite so surprising and it is unclear to me how I can see the power of ML in these results. To be provocative, would the results have been different if you would have just distributed rainfall across the watersheds without a model in between?

**Response 26:** *We thank the reviewer for his comment. We revised the manuscript thoroughly and we believe it is now clearer. Please note that we trained the model and validated it only using the data for Nepal, at the district scale and then at the watershed scale. Overall, we opted for a 90-10 approach, for which 90% of the Nepal data were used for training and 10% for validation.*

*Line 360 and following: Comparing predicted Lifeyears Index (LYI) flood impacts with observed data*

showed good correspondence between high-risk areas identified by the ML method and historical flood locations in Nepal. This suggests that the proposed approach effectively delineates flood risk on a national scale. Figure 8 illustrates this comparison, showcasing observed (empirically evaluated) and ML-predicted LYI values at both watershed (upper row) and district (lower row) levels.

The 'observed' LYI values were empirically calculated from observational data (Table 1) and categorized into three groups: 'low', 'medium', or 'high', with basins/districts labeled as 'high' for LYI values exceeding 1000 years, 'medium' between 100 and 1000 years, and 'low' below 10 years. The 'predicted' values represent the outputs from the machine learning model.

In Nepal, we achieved an overall training accuracy of 97% and a test accuracy of 63%. Notably, training the model at the watershed level yielded higher accuracy compared to the district level. This is attributed to watersheds being hydrologic units that integrate geomorphological and climatic properties, thus providing a more accurate representation of flood dynamics compared to administrative district boundaries.

At the watershed level, nearly all year ranges exhibited a 100% match with observed impacts. In instances where the model's accuracy fell below 100% (e.g., 1985–90 and 1990–95), the LYI values in the affected watersheds were low, indicating that the predictors considered were more indicative of major flooding events.

The superior accuracy achieved at the watershed level underscores the value of implementing the model at this scale when scaling up the system.

[Figure]

**Figure 8: Comparison of prediction with actual socioeconomic impact for watersheds and districts in Nepal. Basin/districts are marked as "high" for LYI over 1000 years. Medium is between 100 and 1000, and low is less than 10. Numbers in parentheses represent accuracy.**

21. L349: The figures you note here do not show what is described in the text.

*Response 27: We corrected this mistake.*

22. L356: I lack some context here - <10% of watersheds see an increase, are all other stable or see a decrease?

*Response 28: Here in section 3.6 we showcase the changes in impact [low to medium (LtoM); medium to high (MtoH); and low to high (LtoH)] increase in impact. Some watersheds have not changed, and some have decreased impact. However, we are concerned and discussed the ones that will threaten the people's future socioeconomic balance.*

23. How can you differentiate here between hazard (rain) and exposure (population) as a driver of

change?

**Response 29:** *We do not attempt to differentiate between hazard and exposure. Rather we use them together to find out the impact. We simply tried to analyze the results and connect the dots by revisiting past occurrences (such as population boom, extreme events, or both).*

24. How do you explain that increase has slowed after 2010 significantly? And how is it possible that in the 1995-2010 jump the number of increasing watersheds is similar to the just 5 year jump between 1990 – 1995? Isn't that completely counterintuitive?

**Response 30:** *It may be counterintuitive, however, there may have been many events that have caused more damage in that 5-year window than the longer span.*

25. L406: While in general 'an intention to make data available' shouldn't be followed, for a journal like NHESS this is definitely not acceptable. Data availability needs to be clearly described (or arhued why this is not the case).

**Response 31:** *The FGP dataset produced in this study is available online. We have revised it in the document.*

26. Technical corrections (Minor issues):

**Response 32:** *We carefully revised the entire manuscript to correct all the minor issues.*

**References:**

Barandun, M., Fiddes, J., Scherler, M., Mathys, T., Saks, T., Petrakov, D., and Hoelzle, M.: The state and future of the cryosphere in Central Asia, Water Secur, 11, 100072, https://doi.org/10.1016/J.WASEC.2020.100072, 2020.

Dollan, I. J., Maina, F. Z., Kumar, S. V., Nikolopoulos, E. I., and Maggioni, V.: An assessment of gridded precipitation products over High Mountain Asia, J Hydrol Reg Stud, 52, 101675, https://doi.org/10.1016/J.EJRH.2024.101675, 2024.

Du, M., Huang, S., Leng, G., Huang, Q., Guo, Y., and Jiang, J.: Multi-timescale-based precipitation concentration dynamics and their asymmetric impacts on dry and wet conditions in a changing environment, Atmos Res, 291, 106821, https://doi.org/10.1016/J.ATMOSRES.2023.106821, 2023.

Maggioni, V. and Massari, C.: On the performance of satellite precipitation products in riverine flood modeling: A review, J Hydrol (Amst), 558, 214–224, https://doi.org/10.1016/J.JHYDROL.2018.01.039, 2018.

Maina, F. Z., Kumar, S. V., Getirana, A., Forman, B., Zaitchik, B. F., Loomis, B., Maggioni, V., Xue, Y., McLarty, S., and Zhou, Y.: Development of a Multidecadal Land Reanalysis over High Mountain Asia, https://ams.confex.com/ams/103ANNUAL/meetingapp.cgi/Paper/415850, 11 January 2023.

Miles, E., McCarthy, M., Dehecq, A., Kneib, M., Fugger, S., and Pellicciotti, F.: Health and sustainability of glaciers in High Mountain Asia, Nature Communications 2021 12:1, 12, 1–10, https://doi.org/10.1038/s41467-021-23073-4, 2021.

Delalay, M., Ziegler, A. D., Shrestha, M. S., Wasson, R. J., Sudmeier-Rieux, K., McAdoo, B. G., and Kochhar, I.: Towards improved flood disaster governance in Nepal: A case study in Sindhupalchok District, International Journal of Disaster Risk Reduction, 31, 354–366, https://doi.org/10.1016/j.ijdrr.2018.05.025, 2018.

Monjo, R., & Martin-Vide, J. (2016). Daily precipitation concentration around the world according to several indices. International Journal of Climatology, 36(11), 3828–3838. https://doi.org/10.1002/JOC.4596

Noy, I.: A Global Comprehensive Measure of the Impact of Natural Hazards and Disasters, Glob Policy, 7, 56–65, https://doi.org/10.1111/1758-5899.12272, 2016.

Saki, S. A., Sofia, G., & Anagnostou, E. N. (2023). Characterizing CONUS-wide spatio-temporal changes in

daily precipitation, flow, and variability of extremes. Journal of Hydrology, 626, 130336. https://doi.org/10.1016/J.JHYDROL.2023.130336

Sofia, G. and Nikolopoulos, E. I.: Floods and rivers: a circular causality perspective, Sci Rep, 10, https://doi.org/10.1038/s41598-020-61533-x, 2020.

Sofia, G., Ragazzi, F., Giandon, P., Dalla Fontana, G., & Tarolli, P. (2019). On the linkage between runoff generation, land drainage, soil properties, and temporal patterns of precipitation in agricultural floodplains. Advances in Water Resources, 124, 120–138. https://doi.org/10.1016/j.advwatres.2018.12.003

Sofia, G., Tarolli, P., Cazorzi, F., and Dalla Fontana, G.: Downstream hydraulic geometry relationships: Gathering reference reach-scale width values from LiDAR, Geomorphology, 250, 236–248, https://doi.org/10.1016/j.geomorph.2015.09.002, 2015.

Manfreda, S., Di Leo, M., and Sole, A.: Detection of flood-prone areas using digital elevation models, Journal of Hydrologic Engineering, 16(10), 781–790, http://dx.doi.org/10.1061/(ASCE)HE.1943-5584.0000367, 2011.

Manfreda, S., Nardi, F., Samela, C., Grimaldi, S., Taramasso, A. C., Roth, G., and Sole, A.: Investigation on the use of geomorphic approaches for the delineation of flood prone areas, Journal of Hydrology, 517, 863–876, https://doi.org/10.1016/j.jhydrol.2014.06.009, 2014.

Manfreda, S. and Samela, C.: A digital elevation model-based method for a rapid estimation of flood inundation depth, Journal of Flood Risk Management, 12(Suppl. 1), e12541 1–10, https://doi.org/10.1111/jfr3.12541, 2019.

Samela, C., Albano, R., Sole, A., and Manfreda, S.: A GIS tool for cost-effective delineation of flood-prone areas. Computers, Environment and Urban Systems, 70, 43–52, A GIS tool for cost-effective delineation of flood-prone areas. Computers, Environment and Urban Systems, 2018.

Samela, C., Manfreda, S., De Paola, F., Giugni, M., Sole, A., and Fiorentino, M.: DEM-based approaches for the delineation of flood-prone areas in an ungauged basin in Africa, Journal of Hydrologic Engineering, 21(2), 06015010, https://doi.org/10.1061/(ASCE)HE.1943-5584.0001272, 2016., Journal of Hydrologic Engineering, 21(2), 06015010, https://doi.org/10.1061/(ASCE)HE.1943-5584.0001272, 2016.

Samela, C., Troy, T. J., & Manfreda, S.: Geomorphic classifiers for flood-prone areas delineation for data-scarce environments, Advances in Water Resources, 102, 13–28, https://doi.org/10.1016/j.advwatres.2017.01.007, 2017., Advances in Water Resources, 102, 13–28, https://doi.org/10.1016/j.advwatres.2017.01.007, 2017.

**Review by RC2**

1. Clarity and Structure: The abstract is well-structured, presenting the problem, the proposed solution, and a few findings. However, some sentences are complex (starting from the title), and more concise wording could enhance clarity,

**Response:** *We thank the reviewer for this comment. We revised the complex sentences to enhance their readability.*

2. Methodology: The use of the Lifeyears Index (LYI) as a measure for socioeconomic flood impact is well explained. It would be beneficial to provide a brief explanation of how the geomorphologically guided machine learning approach works, even if it is in a bit summary.

**Response:** *We revised the methodology to be clearer on the geomorphologically guided machine learning approach. A notable advantage of the proposed approach lies in its reliance on automatic techniques leveraging globally available datasets, thereby facilitating its applicability across diverse geographical regions to forecast socioeconomic flood impacts. The framework also benefits from leveraging on geomorphologically-driven information, to have an improved characterization of the*

*different aspects of the underlying physical processes shaping the landscape and possibly impacting flood characteristics. By incorporating such domain knowledge into the ML model, the framework can better generalize across different regions and conditions, improving robustness and reliability for risk mapping in diverse environments and facilitating informed decision-making for flood management and mitigation strategies.*

3. Data: The abstract mentions training the model with over 6000 flood events from 1980 to 2020, but it is mentioned that the model shows variability from 1980 to 2022 as temporal Coverage. So, /what's the correct timeline?

**Response:** *There is a mistake and both of them should be the same. It should be 1980 to 2020. We corrected it.*

4. Conclusion: A brief conclusion summarizing the main contributions and implications of the study would be beneficial.

**Response:** *Thank you for the suggestions. We revised the conclusion and summarized the main contributions and implications of the study.*

**Review by CC1**

The manuscript "Predictive understanding of socioeconomic flood impact in data scarce regions based on channel properties and storm characteristics: Application in High Mountain Asia(HMA)" by Khanam et al. used LYI and ML methods to evaluate and predict the flood impacts and risk due to precipitation in HMA. This work is first time to evaluate socioeconomic impacts of flood hazards in data scarce region. However, it is not good writing. The structure is not reasonable. And the XGboosting tools is not clear to solve what? Thus, I would suggest it should be major revision.

General comments:

- In HMA there are also GLOF which risk the human being and infrastructure. If possible, please include evaluating the socioeconomic flood impact.

**Response:** *We thank the reviewer for this suggestion. GLOFs in general are triggered by glacial melt but here we focus on a climatic driver of the flooding that is related to rainfall. Addressing the damages due to GLOF is separate from the scope of this study. In general, this paper focuses on fluvial and pluvial flooding, and we made this clearer in the introduction.*

- Data-Scarce regions should be clear (which data or which type of data). In HMA, population is scarce. And Socio activity is also low.

**Response:** *We have revised the manuscript thoroughly to correct these unnoticed mistakes and to improve overall the quality of the writing. Please note that when we defined HIMAT as 'data scarce' we did not do so to avoid scrutiny of the data, but rather to highlight the complexity of HMA itself, challenged by several factors, including the scarcity of ground observations covering consistent timeframes homogeneously, as highlighted by other works in literature (Barandun et al., 2020; Dollan et al., 2024; Miles et al., 2021). We understand that some of the statements were phrased in an ambiguous way, which we have revised to address accordingly. In the revised work, we explained that-*

*Line 53-58: Gathering data for the scale of the HMA region is a difficult task, as it requires collecting data from several countries and multiple sources, and this poses challenges due scarcity of ground observations covering consistent timeframes homogeneously, as highlighted by various works in the literature (Barandun et al., 2020; Dollan et al., 2024; Miles et al., 2021). Especially in the context of the impact of floods using socioeconomic data, the analysis involves examining the number of fatalities,*

*injured and people otherwise affected, as well as the financial damage that natural disasters cause, and this information is generally not always available, or it is collected at the local scale based on reported events. Major disasters are reported in global databases such as The International Disaster Database (EMDAT, www.emdat.be), or, for Nepal, the Nepal Disaster Risk Reduction Portal ([http://drrportal.gov.np/](http://drrportal.gov.np/)) which we used in our paper, but these datasets are also not complete. For example, EMDAT only considers events with at least one of the following criteria: 1)10 fatalities; 2)100 affected people; 3) a declaration of state of emergency; 4) a call for international assistance.*

*The main idea behind this paper was to provide a tool for a rapid estimate of potentially highly impacted areas, based on information accessible and updateable quickly, such as population number, rainfall intensities, and a geomorphologic index that can be derived from global DEMs (or high-resolution local ones when available).*

Specific comments

1.Section 2.2 methods. This title is not reasonable. Is section 2.3(Machine learning model) methods? In addition, the dataset and methods in this section should be divided, for example, 2.2.4 exposure(population) is datasets.

**Response:** *Thank you for the suggestions. We reorganized the section and added a separate section for the dataset.*

*We kept the 2.2 section as "Methods". This section explains the methods we used in this study and it is based on ML thus it proceeds to explain the model as well as some of the data processing.*

2.Line 135 Why is it classified by LYI values(<2,2-3, and >3).

**Response:** *We have classified the LYI as a basis for comparison across all the watersheds and periods. The three groups correspond to <100, between 100-1000, and over 1000.*

3.Line 217 While XGBoosting is …,  this sentence is incomplete.

Line 217 this section (machine learning model) is a little difficult to understand the role that is plays.

**Response:** *We revised it.*

---

## Referee Report (RR1)

[referee-annotated manuscript omitted]

---

## Referee Report (RR2)

**Predictive Understanding of Socioeconomic Flood Impact in Data-Scarce Regions Based on Channel Properties and Storm Characteristics: Application in High Mountain Asia (HMA)**

The manuscript presents a machine learning-based approach to predict socioeconomic flood impacts in the High Mountain Asia (HMA) region, utilizing channel properties, storm characteristics, and socioeconomic indicators. The study addresses a pressing issue of flood vulnerability in data-scarce regions by combining geomorphic and climatic data to estimate flood risk. The introduction of a Lifeyears Index (LYI) to assess long-term impacts adds an innovative dimension to the study. However, while the topic is relevant and timely, the manuscript suffers from significant methodological, data handling, validation, and presentation flaws, which diminish its overall scientific contribution.

**Strengths:**

The focus on flood risk in High Mountain Asia (HMA), a region highly vulnerable to hydro-meteorological extremes, is highly relevant in light of climate change and its associated impacts on vulnerable populations. Floods in HMA are a significant concern, particularly given the region's complex topography, glacial melt dynamics, and varying monsoonal patterns.

The paper attempts to incorporate machine learning (ML), specifically XGBoost, into flood risk modeling. While ML is becoming increasingly common in environmental sciences, its application in flood impact assessment for data-scarce regions remains a novel area of research. The methodology has the potential to offer more flexibility in prediction compared to traditional models.

The use of the LYI metric is commendable, as it provides a more comprehensive way to assess both the immediate and long-term socioeconomic impacts of floods. This metric is a valuable addition, especially as it incorporates the broader social cost of disasters, similar to the Disability Adjusted Life Years (DALYs) used in health studies.

**Major Weaknesses:**

**1. Data Handling and Transparency:**

- **Unsubstantiated Data Scarcity Claims**: The manuscript repeatedly claims that the HMA region is "data-scarce," but it provides no specific justification for this assertion. Data from countries such as China, India, Pakistan, and Nepal is available, albeit not as openly accessible as in Europe or the United States. The blanket claim that the region lacks sufficient data without discussing what is missing or inaccessible is misleading.

- **Insufficient Documentation of Data Sources**: The sources of key data, particularly socioeconomic and population data, are not well-documented. For example, census data from Nepal is mentioned, but there is no detailed explanation of how it was applied to watershed-scale flood modeling, which is critical for reproducibility. The use of knoema.com, a potentially unstable and non-reputable data source, further weakens the credibility of the research. It is crucial for scientific rigor that data sources be transparent, traceable, and stable, and that the uncertainties or limitations of these data be fully addressed.

- **Socioeconomic Data Processing**: The spatial scaling of district-level socioeconomic data to the watershed level is a critical methodological step that is poorly explained. How was this data aggregated or distributed? What were the assumptions or limitations involved in using this data at the watershed scale?

**2. Methodological Issues:**

- **Unclear Scope and Flood Types**: The paper does not clearly specify the types of floods being modeled. The introduction mentions multiple types of flooding (e.g., pluvial, fluvial, glacial lake outburst floods), but the methodology seems focused on fluvial floods. This inconsistency in the types of flood hazards being assessed undermines the clarity and focus of the study. If the study is limited to fluvial flooding, this should be clearly stated in both the abstract and methods, and the introduction should avoid discussing other types unless they are directly relevant.

- **Geographical Scope and Upscaling**: The model is trained on data from Nepal but then applied to the entire HMA region. However, the manuscript does not provide sufficient justification for this upscaling. HMA includes diverse climatic regions, ranging from arid areas in Central Asia to monsoon-dominated zones in the southern Himalayas. A model that works well in Nepal may not generalize to other parts of HMA without further validation. It would be more scientifically sound to either focus solely on Nepal or provide validation for other regions in HMA.

**3. Model Validation and Use of ERA5 Data:**

- **Validation Challenges**: The manuscript relies on ERA5 precipitation data for flood simulation in HMA, but ERA5 has well-documented limitations in representing precipitation in mountainous regions. Precipitation in these areas is highly variable, and ERA5's coarse spatial resolution may not adequately capture localized rainfall events. The authors should either provide a more thorough discussion of these limitations or supplement ERA5 data with region-specific datasets (e.g., from local meteorological agencies) to improve validation.

- **Model Validation with Flood Data**: The manuscript lacks sufficient validation of the ML model's predictive power. The use of the Dartmouth Flood Observatory (DFO) database, which focuses primarily on lowland floods, may not be the best choice for validating a model intended to predict flood impacts in mountainous regions. Additionally, the number of flood events recorded in the DFO database for Nepal is limited, making it questionable whether this provides robust validation for the entire HMA region.

**4. Presentation and Writing Quality:**

- **Grammar, Formatting, and Citations**: The manuscript contains numerous grammatical errors, incomplete references, and formatting issues. Several citations are listed as "n.d." or are missing from the reference list entirely. These issues, while minor in isolation, collectively reduce the professionalism of the manuscript and suggest a lack of attention to detail.

- **Figures and Captions**: The figures in the manuscript, particularly Figure 9, are inadequately explained. Captions are vague and do not provide sufficient information for readers to interpret the figures independently of the text. For example, Figure 9(a) is mislabeled as "rainfall" when it appears to depict elevation. Each figure should be carefully checked for accuracy, and captions should be expanded to clarify what the figure represents and how it relates to the findings.

**5. Lack of Novelty in Results:**

- **Expected Findings**: Much of what is presented in the results seems to reflect known trends rather than novel insights. For example, the model's prediction that flood impacts are higher in areas with higher population density is unsurprising and does not provide new knowledge. The authors should focus on demonstrating how their ML model offers unique predictive power or novel findings that go beyond confirming expected patterns.

- **Limited Discussion of Results' Implications**: The discussion of the results lacks depth, particularly in terms of practical applications. How can the findings be used for real-world flood mitigation or policy-making? What specific new insights into flood risk do these results offer that were not already known?

**Recommendations for Improvement:**

1. **Clarify and Focus the Scope**:

   o Clearly specify the types of floods being modeled (e.g., fluvial only) and ensure that the methodology aligns with this scope. Avoid mentioning other flood types unless they are directly relevant.

   o If the study is focused on Nepal, limit the geographical scope accordingly. Alternatively, provide validation for the model's application across the broader HMA region.

2. **Improve Data Transparency and Documentation**:

   o Provide clear documentation of all data sources, including stable links, proper citations, and explanations of how the data were processed. Address the limitations of using certain datasets and provide a more balanced discussion of data availability in HMA.

   o Explain in detail how socioeconomic and population data were integrated into the model, particularly how district-level data was applied at the watershed scale.

3. **Strengthen Model Validation**:

   o Provide a more robust validation of the model using appropriate regional datasets. Supplement ERA5 data with local meteorological data where possible, and address its limitations for flood modeling in mountainous regions.

   o Reconsider the use of the DFO database for model validation. Instead, use more region-specific flood data, particularly for high-altitude areas, to better assess the model's predictive accuracy.

4. **Enhance Writing and Figure Presentation**:

   o Revise the manuscript to eliminate grammatical errors, correct formatting, and complete missing references. The manuscript should be proofread thoroughly before resubmission.

   o Improve figure captions to ensure that all visual data are clearly explained and relevant to the text. Ensure that all figures are correctly labeled and accurately represent the data being discussed.

5. **Increase the Novelty and Practical Implications of the Findings**:

   o Emphasize any novel findings from the model, particularly any unexpected results or new insights into flood risk in HMA. Highlight how the machine learning approach provides advantages over traditional models.

   o Provide more discussion on how the results can be applied in real-world flood mitigation efforts, disaster management, or policy-making in the HMA region.

**Final Decision: Rejection**

While the topic is highly relevant and the use of machine learning has potential, the manuscript in its current form is not ready for publication. The significant issues with data handling, methodological clarity, and model validation undermine the credibility of the findings. Additionally, the paper lacks sufficient novelty and depth in its results, and the writing and presentation need substantial improvement.

Given these foundational problems, I recommend **rejection**. However, I encourage the authors to address the major issues highlighted in this review. With significant revisions, including better data transparency, clearer scope, and stronger validation, the paper could be reconsidered for submission to a future issue of the journal.

**Note to the Special Issue Editors:** The paper aligns with the theme of hydro-meteorological hazards and socioeconomic vulnerability. However, due to the significant methodological and presentation flaws, it does not meet the standards for publication in this special issue at this time. The authors should be encouraged to revise their work, especially focusing on addressing the issues raised regarding data transparency, model validation, and scope clarity, before resubmitting to the journal or special issue.

---

## Author Response (AR2)

*Response to the Editor*

*Dear Editor,*

*Thank you for your thoughtful feedback and for the opportunity to revise our manuscript. We appreciate the reviewers' comments and have carefully addressed all concerns raised. In particular, we have revised the model training and validation approach and included additional information to strengthen the robustness of the manuscript. We have also clarified and emphasized the innovative aspects and broader relevance of our findings throughout the revised manuscript. We have responded to each comment in detail and incorporated the suggested improvements to enhance the overall clarity and scientific contribution of the work. We hope the revised version now meets the standards for publication in Natural Hazards and Earth System Sciences.*

*Best regards,*

*Mariam Khanam*

*On behalf of all co-authors*

*Response to the Reviewers*

*We thank the reviewers for their detailed and constructive feedback, which has provided valuable guidance for improving our manuscript. Below, we address each comment point by point, incorporating clarifications and revisions where necessary to strengthen our study.*

***Note: Below is our response (italics) to each comment (regular font) from the reviewer***

**Reviewer 1:**

1. A recent study has been performed such as https://doi.org/10.1016/j.jenvman.2024.121764

*Response: We thank the reviewer for this suggestion. We have added this to the manuscript.*

2. Line 137: Please modify the figure such that input (FGP, Rainfall, and Population) and output (LYI) of the XGBoost model can be distinguished.

*Response: Thank you for this suggestion. I have modified the figure to clearly separate the inputs (FGP, Rainfall, Population) from the output (LYI) in the XGBoost model diagram.*

3. Line 156: "variant" may be a better word choice.

*Response: Thank you for pointing this out. We have replaced the current term with "variant" to ensure clarity and precision in our wording.*

4. Line 172: I understand this concept and it is very interesting. It will be more interesting if the map of these categories (low, medium, high LYI) of the training dataset can be shown along with the map of the actual LYI.

*Response: we thank the reviewer for this comment. As the manuscript is already dense with figures and plots, and the method aims to identify the categories, we prefer not to add the figure.*

5. Line 238: Again, a map showing this index over the study area will be very interesting.

*Response: Thank you for this comment. Please consider that in the revised manuscript, we have added Figure 7 showing an example for 1985 to 2020 changes.*

6. Line 279: This is an interesting and crucial idea for your XGBoost model validation, but in my view, the method of dissecting your study area in half, using one half to develop the model and the other half for validation, would be the best way. Then, you can develop a confusion matrix between the simulated LYI category and the observed LYI category. Based on this result, you can even perform an uncertainty analysis of your final result (e.g., what is the probability that an area classified as high LYI actually has high LYI).

*Response: Thank you for this comment. One should consider that we have measured LYI at the district and watershed scale only for Nepal, and not for the whole HMA. Dividing the data in half might fail to capture correctly the geographic variability of the area. As the training dataset is limited, we believe that the proposed approach is more robust. Please see below the part in the manuscript:*

*Line 286: We conducted thorough testing and validation of our model for Nepal, comparing the predicted value of LYI to the calculated Lifeyears Index (LYI) data from tabular values specific to the region. We trained the model and validated it only using the data for Nepal, at the district scale and then at the watershed scale. Overall, we opted for a 90-10 approach, for which 90% of the Nepal data were used for training and 10% for validation. Upon extending the model's applicability to the entire High Mountain Asia (HMA) region, we rigorously assessed the quality of our results by comparing the predicted social impact with that reported in established flood databases covering the region. To verify our findings, we compared the predictions at the HMA level with flood events reported in the Dartmouth Flood Observatory's (DFO) Global Active Archive of Large Flood Events, 1985–Present. This comprehensive database compiles information on major floods sourced from diverse channels such as news reports, governmental records, ground observations, and remote sensing data. Notably, the DFO dataset encompasses various flood types, including lowland floods and mountainous river floods characterized as fluvial and pluvial floods.*

7. Line 305: An additional section that discusses the ranges of predictor variables for watersheds classified as high LYI would be interesting. This is because we cannot solely rely on the AI model, which operates as a black box.

*Response: We have added some discussion and figure 7 to "**3.1. Variability of the Predictors**" section in the manuscript:*

*Line 341: Much of the population of Nepal tends to be concentrated in areas with higher FGP, as is*

*typical for mountainous areas, where population and economic activities are mostly located in the river*

*valleys. Globally, the floodplains of rivers are preferred living spaces for the population and provide*

*favorable locations for economic development. These areas are commonly exposed to floods, however, an increasing population, together with the changes in storminess, mean that the risks from flooding are expected to be higher. On average, the population increased significantly in watersheds that transitioned from low to medium (LtoM), medium to high (MtoH), or low to high (LtoH) flood risk categories (Figure 7: example variability from 1985 to 2020). This suggests that growing population density in certain watersheds may be contributing to increasing flood susceptibility. The CI (climate concentration index) slightly decreased over this period for some watersheds. However, watersheds experiencing population growth were more likely influencing the transition to a higher flood risk category. Although CI has not significantly increased, the interaction between land-use change, urban expansion, and demographic shifts may be playing a role in driving these transitions. Transitioning watersheds have a higher average FGP compared to the overall average FGP and tend to have a larger average watershed area compared to all watersheds. This indicates that larger watersheds are more prone to experiencing shifts FGP and in flood risk categories, possibly due to their ability to accumulate and distribute larger volumes of runoff and sediment. This supports the idea that intrinsic watershed characteristics (such as geomorphology and size) play a role in flood susceptibility alongside external factors like population growth and rainfall concentration index (CI). Area successfully predicted as at high risk (high LYI) in the most recent years, are areas showing high social vulnerability in terms of favorable Social Conditions (lack of communication, access to electricity and infrastructures, lower education, small children under 5); high percentage of migrating community and high risk of poverty and poor infrastructures (Aksha et al.,*

*2019).*

[Figure]

**Figure 7: Average variability of the Rainfall CI (a), population change (b) compared to FGP (c) and LYI Trend (d) from 1980-2020**

8. Line 306: Suggestion: This section digresses slightly from the main result of the paper. It may be better suited for methodology or a separate discussion section.

*Response: Thank you for the suggestion. This section explains how the variability of all the predictors is connected and we consider this to be an important result to explain the correlation. Additionally, we have added a new figure to the section. We believe this section should be considered as a part of the results.*

9. Line 330: Very interesting finding! You could write another paper that discusses this matter on a global scale.

*Response: We appreciate your enthusiasm for this finding. We agree that this topic has significant potential for a broader, global analysis, and we will consider pursuing this in a future publication.*

10. Line 388: The letters "(b)" in the figure legend are overlapped on the other letters.

*Response: Thank you for noticing this formatting issue. We have adjusted the figure 9, legend to remove the overlap and made some further changes as well.*

11. Line 400: Very interesting result, but I would try to find a better way to visualize it using a figure.

*Response: We thank the reviewer for this comment. The visual representation of these results could be as simple as a bar plot. We prefer however to keep this in a table form, to avoid adding too many figures to the manuscript.*

[Figure]

12. Line 416: Same as above. Try to display this information using a figure.

*Response: We thank the reviewer for this comment. The visual representation of these results could be as simple as a bar plot. We prefer however to keep this in a table form, to avoid adding too many figures to the manuscript.*

13. Figure 10: While this figure is very interesting, its resolution is unacceptable. Please provide a high-resolution image.

*Response: We appreciate this feedback. We have replaced the figure with a better resolution version which shows the HMA region.*

14. Line 438: This is too much generalization. The primary reason for this result may be that long-term CI values are less variable than short-term CI values. Additionally, you are applying the model developed based on 35 years of rainfall data to data based on 5 years of rainfall. Please discuss or briefly state the limitation of this result.

*Response: We thank the reviewer for this comment. Please consider that we calculated the CI for 5 years intervals in both cases. We clarified this in the manuscript. Indeed, there might be a difference if CI is computed over different time windows, but this is not the case of this specific study. We added a comment on this in the manuscript in the newly added paragraph about limitations.*

*Line 472: 3.7. Model constraints and limits*

*While this study demonstrates the promise of accurate flood impact prediction, the use of static Flood Geomorphic Potential (FGP) maps presents limitations. Flooding alters channel morphology and downstream topography, impacting future flood dynamics (Khanam et al., 2024). Therefore, dynamic flood topographies are essential for robust hazard assessment. Although high-resolution data post-extreme events can enhance prediction accuracy, the availability of such data is constrained by acquisition frequency. Hence, efforts to improve data availability post-disaster are crucial for enhancing*

*the reliability of predictive models. Researchers could also derive FGPs from enhanced high-resolution terrain data, such as those derived from LiDAR sources if available. In such cases, however, it is advisable to retrain the model and reassess the significance of this parameter in the updated model, as terrain resolution and survey techniques might determine a variability of the data, especially when dealing with hydrologic parameters (Sofia, 2020).*

*The climate index considered in this study might vary depending on the input dataset (Reanalysis VS measurements), as well as on the timescale of the analysis. When comparing results to this study, researchers should make careful consideration of the length of the time window used for this evaluation (5 years). If daily data are considered over shorter time windows (e.g., 1 year), the index itself might result in higher values, capturing only short-term variability due to specific isolated storms. Seasonal analyses, on the other hand, would capture more the concentration due to monsoon periods, or dry vs. wet months. The proposed multi-year analysis is in line with literature studies, pertaining climate change studies and studies on the effect of floodings (Sofia et al., 2019; Saki et al., 2023; Du et al., 2023).*

*Population data for this work relies on standard available datasets. When considering the method to predict future changes, outside the time range covered by the proposed model, headcounts alone cannot offer a full picture. It is crucial also to consider additional elements that could determine population shifts over time.*

15. Figure 11: Please enhance the resolution of this figure as well.

*Response: Thank you for bringing this to our attention. We have replaced the figure with a better resolution version.*

Reviewer 2:

1. Comment: There are some major points that lead me to this decision. The most important is that you trained the XGBoost model on a single nation, achieving what seems to be overfitting, and then you deploy it to a much larger region without any validation. This leads to the possible conclusion that any results you report, while interesting, might be wrong or biased toward similar mechanics as you would have in Nepal.

> The choice of the indices expands the possibility of expanding the study. LYI in a different situation. We can not make decisions based on societal variables. It should not be used as the absolute labeling of the areas

*Response: We thank the reviewer for this comment. We acknowledge that the dataset used for training the XGBoost model is geographically limited. The model parameters are determined over the training set. We have leveraged data augmentation techniques to augment training samples and balance difference classes in order to overcome the overfitting issue.*

*However, we emphasize that we have taken steps to validate the extended model using the DFO dataset, which records actual flood events. This validation helps ensure that the model's predictions are not solely dependent on the training region but can reasonably generalize to broader areas.*

*Regarding the concern of potential bias, we note that the variability in the dataset across different regions is comparable. Specifically, the ranges for the Climate concentration Index (CI) Index and Flood Geomorphic Potential (FGP) in Nepal and the HMA are as follows:*

*Nepal has*

*CI: 0.57 ± 0.15*

*FGP: 16.5 ± 18.3*

*And this variability encompasses overall the variability of the HMA dataset (CI: ranges from 0.3 to 0.8 while FGP ranges between 2.1 to 67.5). Please consider that our primary objective is not to establish an absolute classification of flood-prone areas based on societal variables but rather to provide a reasonable assessment of vulnerability in terms of classes of life year lost. While we recognize the limitations of our approach, our findings offer valuable insights into the regional variability of flood risk and its driving factors. We will clarify these points further in the manuscript to reflect the scope and intent of our study.*

*Line 506: Our goal is to provide a reasonable assessment of vulnerability through life years lost, rather than to definitively classify flood-prone areas by societal factors. Despite certain limitations, our findings offer valuable insights into regional flood risk and its key drivers.*

2. Comment: Moreover, there is no clear description of how you trained and validated the model and how you assessed parameter importance.

*Response: Thank you for pointing this out. In the revised manuscript we clarified the proposed approach.*

*Line 286: We conducted thorough testing and validation of our model for Nepal, comparing the predicted value of LYI to the calculated Lifeyears Index (LYI) data from tabular values specific to the region. We trained the model and validated it only using the data for Nepal, at the district scale and then at the watershed scale. Overall, we opted for a 90-10 approach, for which 90% of the Nepal data were used for training and 10% for validation. Upon extending the model's applicability to the entire High Mountain Asia (HMA) region, we rigorously assessed the quality of our results by comparing the predicted social impact with that reported in established flood databases covering the region. To verify our findings, we compared the predictions at the HMA level with flood events reported in the Dartmouth Flood Observatory's (DFO) Global Active Archive of Large Flood Events, 1985–Present. This comprehensive database compiles information on major floods sourced from diverse channels such as news reports, governmental records, ground observations, and remote sensing data. Notably, the DFO dataset encompasses various flood types, including lowland floods and mountainous river floods characterized as fluvial and pluvial floods.*

3. Comment: Overall, the paper is also improvable in terms of writing, with several redundant phrasings and unclear sections.

*Response: We appreciate this feedback. We will thoroughly review the manuscript to eliminate redundant language and clarify ambiguous sections wherever possible.*

4. Comment: Figure 2: It is unclear why the LYI is both an input and an output of the model. Despite the caption saying to refer to the text, this should be made clearer in the image itself as well.

*Response: We revised Figure 2 to clearly distinguish between input and output variables.*

5. Comment: Line 145: What is the point of using district-level data if you then aggregate it at the watershed level, especially for the testing on the HMA region?

*Response: We used district-level data to account for regional variations in population and socioeconomic impacts, which we then aggregated at the watershed level to match the spatial scale of our hydrological analysis. We will clarify this in the text.*

6. Comment: Line 147: How do you weight different districts?

*Response: The districts are not weighted per se. The aggregation from district to watershed is done by a weighted average, considering the extent of district area within the watershed as a weight. We clarified this in the manuscript.*

7. Comment: Table 1: There's no need to repeat "Y =" in the description, as it already appears on the left.

*Response: Thank you for catching this redundancy. We removed the extra "Y ="*

8. Comment: Line 169: You often use capital letters for Low, Middle, and High risk. Consider using lowercase letters. This applies to the rest of the paper as well.

*Response: We have considered the suggestion and revised the text.*

9. Comment: Lines 194-195: In theory, you can fully automate from terrain data if you use contributing areas, as you can delineate them based on topography. You could argue that your method provides a better representation of bankfull discharge, but you would need to prove it by comparing the two approaches.

*Response: We thank the reviewer for this comment. The method is fully automated from terrain. The FGP stems from the work of Samela et al, where they developed the index to identify flood prone areas based on the proposed geomorphic classifier. Their approach required as input a relationship connecting indeed drainage area to bankfull conditions. For the proposed approach, we automated this last part, meaning the definition of bankfull condition, and we did so by applying methods already existing in literature. The goodness of this method was already described in the referenced papers (Sofia et al. 2011, Sofia and Nikolopoulos 2020, Sofia et.al 2017). Validating bankfull geometry at the scale of HMA is not feasible, as measurements are not readily available. This, furthermore, goes beyond the scope of the work.*

10. Comment: Line 209: You are comparing with HAND, which is not exactly a standard inundation model, as it models water depths with several assumptions. I think it is overall fine to compare with it, but mention at least its limitations.

*Response: The HAND (Height Above Nearest Drainage) model is a widely used approach for estimating flood inundation extents and water depths. It operates on the principle of deriving relative elevations from a DEM, similar to our approach, which also relies on DEM-based analysis. While having assumptions may introduce some limitations in accurately capturing complex flood dynamics, HAND remains a useful and practical method for large-scale flood assessment due to its computational efficiency and compatibility with readily available topographic data. Given these similarities, we find it reasonable to include HAND as a comparative reference in our study while acknowledging its limitations. We will ensure that these aspects are clearly mentioned in the discussion.*

11. Comment: Figure 3: Please reduce the size of the plot, as it does not need to be this large.

*Response: Thank you for the suggestion. We have replaced the figure and reduced the size as much as possible.*

12. Comment: Lines 211-212: Didn't you just say that you used a modified version of this index? If that is the case, then your claim is incorrect, as only the GFI has been validated, not the FGP.

*Response: we thank the reviewer for this comment. Please consider that FGP stems from the work of Samela et al, where they developed the index to identify flood prone areas based on the proposed geomorphic classifier. Their approach required as input a relationship connecting indeed drainage area to bankfull conditions. For the proposed approach, we automated this last part, meaning the definition of bankfull condition, and we did so by applying methods already existing in literature. The goodness of this method was already described in the referenced papers (Sofia et al. 2011, Sofia and Nikolopoulos 2020, Sofia et.al 2017). Validating bankfull geometry at the scale of HMA is not feasible, as measurements are not readily available. This, furthermore, goes beyond the scope of the work. As an overall visual assessment, we proposed the HAND comparison, mentioning in the revised paper its limitations. Furthermore, we rephrased the sentences by changing the terminology from FGP to DEM-derived geomorphic index in this context.*

*It's worth noting that the DEM-derived geomorphic index has been previously published and applied in various contexts (Samela et al., 2017). While testing the quality of the DEM-derived geomorphic index lies beyond the scope of this work, its effectiveness for flood mapping has been well-established in previous studies (Manfreda et al., 2011, 2014; Manfreda & Samela, 2019; Samela et al., 2016, 2018), which have demonstrated the utility of the methodology, particularly in ungauged conditions, for preliminary identification of flooded areas in regions where conducting expensive and time-consuming hydrologic-hydraulic simulations may not be feasible. The goodness of the bankfull measurement system, furthermore, was already described in (Sofia et al. 2011, Sofia and Nikolopoulos 2020, Sofia et.al 2017).*

13. Comment: Section 2.3.2: Please add the formula used for FGP calculation in the text, rather than only in the figure. You can leave the figure to explain the variables used.

14. Comment: Figure 4a: What is

w=αA β? I understood from the text that the reference height was determined from the landscape rather than formulas.

*Response 13/ 14: Thank you for this suggestion. We included the FGP calculation formula directly in the text of Section 2.3.2*

*Line 190: We opted for considering a variation of the Samela et al., (2017) which is a modified Geomorphic Flood Index (GFI) by Sofia, et al., 2017b & Sofia et al., 2015, thereby described as Flood Geomorphic Potential (FGP).*

$$FGP = ln\ (h_r/\ H) \tag{2}$$

*The index is calculated as the logarithm function of the bankfull elevations, H (estimated using a hydraulic scaling function, or HSF ($w=\alpha A^{\beta}$), based on bankfull width (w) and contributing area (A)) in the element of the river network closest to the point under examination and the elevation difference between these two points, $h_r$ (Figure 4, Equation 2). The index was improved over a main aspect: the*

*automatic identification of the HSF directly from terrain data, applying the technique of (Sofia, et al., 2017b; Sofia et al., 2015) to retrieve the bankfull location automatically through the landscape. This has the advantage of allowing for full automation of the mapping starting purely from terrain data.*

15. Comment: Figure 4b: Include in the caption that you are also showing the corresponding orthophotos of the sites considered for the flood maps.

*Response: Thank you for this comment. We have added that to the caption.*

16. Comment: Moreover, the legend is quite small; consider increasing it.

*Response: Thank you for noticing this detail. We have increased the legend size.*

17. Comment: Line 249: I don't understand the need to clarify that one variable is on the x-axis and the other on the y-axis. Consider removing this for clarity.

*Response: Done.*

18. Comment: Equation 2: I would move this equation to line 241, just before explaining it.

*Response: We moved Equation 2 to align with the explanatory text.*

19. Comment: Figure 5: It would be useful to color different areas under the curve with different colors or patterns to help the reader.

*Response: Done.*

20. Comment: Section 2.4.1: There is no indication of how the model was trained, validated, and tested; how many samples were used for each; or how results were assessed in terms of metrics.

*Response: Thank you for your comment. We have modified the section with more details and also Section 3. We discuss further on the feature importance, model performance and validation in the section 3.*

*2.4.1 Validation of the System at the HMA Scale*

*We conducted thorough testing and validation of our model for Nepal, comparing the predicted value of LYI to the calculated Lifeyears Index (LYI) data from tabular values specific to the region. We trained the model and validated it only using the data for Nepal, at the district scale and then at the watershed scale. Overall, we opted for a 90-10 approach, for which 90% of the Nepal data were used for training and 10% for validation. In total, there are 1520 data points for 38 basins from 1981 to 2020. The model was trained on the data by removing one specific year's data (e.g., 2003, 2012, 2017, and 2020) and then test on this specific year's data. Average performance, e.g., precision, recall, F1 measure was reported. Upon extending the model's applicability to the entire High Mountain Asia (HMA) region, we rigorously assessed the quality of our results by comparing the predicted social impact with that reported in established flood databases covering the region. To verify our findings, we compared the predictions at the HMA level with flood events reported in the Dartmouth Flood Observatory's (DFO) Global Active Archive of Large Flood Events, 1985–Present. This comprehensive database compiles information on major floods sourced from diverse channels such as news reports, governmental records, ground observations, and remote sensing data. Notably, the DFO dataset encompasses various flood types, including lowland floods and mountainous river floods characterized as fluvial and pluvial floods.*

21. Comment: This section is also very unclear when you describe the comparison with the DFO database.

*Response: Thank you for this comment. We have tried to make the methodology clear as much as possible. We will appreciate if the reviewer can give us some specific indication on which part needs to be modified.*

22. Comment: Figure 6 is not clear enough from the legend and caption.

*Response: Figure 6 does not have any legend in it. We are not sure which figure are you referring to.*

23. Comment: Section 3.2: How was this variable importance assessed?

*Response: We have modified section 3.2 with the following explanation of the feature importance:*

*In this section, we present a variable importance comparison (Figure 8) based on the Feature Importance Score (F-score) in XGBoost. XGBoost provides F-score based on how frequently a feature is used in splitting the data across all decision trees. This is the number of times a feature appears in a split across all trees in the model. A higher value indicates that the feature was used more frequently in decision-making, suggesting it has a stronger influence on model predictions. The F-score indicated that population (Pop) was the most important variable, which was consistent with our expectation in the sense that the socioeconomic impact depends largely on the exposure. The climate variable (CI) happened to be the next important variable, showing the significance of the region's climate on the socioeconomic impact of flood occurrences.*

24. Comment: Lines 344-346: This should go in the methodology section. Moreover, classification metrics with more than two classes should be better discussed, as they are not as straightforward.

*Response: Thank you for this suggestion. We decided not to relocate the accuracy metrics results to the methodology section however, we have added the following explanation to the methodology:*

*Line 298: We performed a hyperparameter tuning using weighted accuracy (1-3-9 weighting scheme) for subsequently (low, med and high classes), prioritizing category "high". Initially, when XGBoost was trained, it achieved a 63% test accuracy, but its confusion matrix revealed that it struggled to correctly classify the most destructive category (category 3). Since this category was of primary interest, the model was refined using weighted accuracy, emphasizing its importance. A 5-fold cross-validation with 1000 iterations was conducted, and for each cross-validation, oversampling was applied to balance the dataset.*

*Line 389: The final tuned models achieved weighted accuracies between 52% and 58%, but significantly improved recall (71%), precision (73%), and F1-score (72%) for category "high". This means that out of 34 actual instances of the highest category, 24 were correctly predicted, and out of 33 predicted cases, 24 were accurate, confirming that the model effectively focused on the most critical category. This suggests that while the overall accuracy slightly decreased due to the re-weighting, the model's performance in identifying the most critical cases significantly improved.*

25. Comment: Figure 7: The F score, which should go from 0 to 1, has values above 3000. Please correct the legend.

*Response: The label "F SCORE" in this case does not represent the F1-score from classification metrics but rather the number of times a feature was used in splits. A more appropriate label for the x-axis would be: "Feature Importance (Number of Splits in XGBoost)". We modified the following in the manuscript:*

*Line 378: In this section, we present a variable importance comparison (Figure 8) based on the Feature Importance Score (F-score) in XGBoost. XGBoost provides F-score based on how frequently a feature is used in splitting the data across all decision trees. This is the number of times a feature appears in a split across all trees in the model. A higher value indicates that the feature was used more frequently in decision-making, suggesting it has a stronger influence on model predictions. The F-score indicated that population (Pop) was the most important variable, which was consistent with our expectation in the sense that the socioeconomic impact depends largely on the exposure. The climate variable (CI) happened to be the next important variable, showing the significance of the region's climate on the socioeconomic impact of flood occurrences.*

26. Comment: Table 2: "a classification model" is too generic. Please specify that this is for your trained model, applied to the test dataset (or validation? This is not clear).

*Response: We agree this should be clarified. We will specify in the table caption that these metrics refer to test dataset.*

27. Comment: Line 366: This seems to indicate overfitting. Did you use a validation dataset to limit this?

*Response: The reviewer is correct that this could be the case. For the overall validation, we compared the results with the DFO to highlight the actual quality of the proposed model.*

28. Comment: Section 3.4: It may be valuable to add a correlation plot to understand if there is a match between DFO and LYI, rather than relying on a table that contains redundant information.

*Response: We appreciate the suggestion, however it is not possible to create a real correlation measurement, because our system predicts classes of LYI (low med high), and there is no reference LYI at the watershed scale for the whole HMA. Hence why we opted for the proposed analysis, where we use the DFO reported losses as a proxy of the impact of measured floods.*

**Reviewer 3**

**We appreciate the reviewer's comments. We would like to highlight that all the raised comments follow exactly the public review, by Jakob F. Steiner, https://doi.org/10.5194/nhess-2023-120-RC1, which were extensively addressed in the revised submission and for which we provided a detailed response. We believe that this review was AI generated, and we respectfully disagree with this specific comment of rejection, as it was not pertaining to the revised manuscript submitted.**

*To ensure a thorough evaluation, we have conducted a detailed comparison between this review and our previous responses, demonstrating that these points were extensively addressed in our revised submission. Given this overlap, we believe the review may not fully reflect the updates and revisions incorporated into the manuscript. Therefore, we respectfully disagree with the recommendation for rejection, as it does not appear to consider the revised version of our work.*

| New Review Comments and Responses | Jakob F. Steiner Comments https://doi.org/10.5194/nhess-2023-120-RC1 and submitted Responses |
|---|---|
| **1. Data Handling and Transparency:** | |
| • *Unsubstantiated Data Scarcity Claims:* The manuscript repeatedly claims that the HMA region is "data-scarce," but it provides no specific justification for this assertion. Data from countries such as China, India, Pakistan, and Nepal is available, albeit not as openly accessible as in Europe or the United States. The blanket claim that the region lacks sufficient data without discussing what is missing or inaccessible is misleading. | **Previous Comment:** The lack of appreciation of existing data and simply depicting the target region as 'data scarce' to avoid scrutiny from what is known already. ***Response:*** *We thank the reviewer for this comment. Please note that our intent was not to avoid scrutiny of the data, but we understand that some of the statements were phrased in an ambiguous way, which we have revised to address accordingly. Line 53-58: Gathering data for the scale of the HMA region is a difficult task, as it requires collecting data from several countries and multiple sources, and this poses challenges due to the possible inhomogeneities of standards between different organizations. Especially in the context of the impact of floods using socioeconomic data, the analysis involves examining the number of fatalities, injured and people otherwise affected, as well as the financial damage that natural disasters cause, and this information is generally not always available, or it is collected at the local scale based on reported events.* |
| • *Insufficient Documentation of Data Sources:* The sources of key data, particularly socioeconomic and population data, are not well-documented. For example, census data from Nepal is mentioned, but there is no detailed explanation of how it was applied to watershed-scale flood modeling, which is critical for reproducibility. The use of knoema.com, a potentially unstable and non-reputable data source, further weakens the credibility of the research. It is crucial for scientific rigor that data sources be transparent, traceable, and stable, and that the uncertainties or limitations of these data be fully addressed. | **Previous Comment:** There is very poor documentation on where the exposure data is taken from and there is no way to make this traceable (no stable links, and also no attempt so far to make your own produced data available).

***Response:*** *We thank the reviewer for this comment. We will include a table with information on all the datasets used. Regarding the links being "not stable" – the data required for the index were accessed and we tested the links before submission. In the revised paper, we will clarify the date of the latest access so that the data is more clearly referenced.* ***Note:*** *We have in fact removed the knoema.com from the previous version of the manuscript.* |
| • *Socioeconomic Data Processing:* The spatial scaling of district-level socioeconomic data to the watershed level is a critical methodological step that is poorly explained. How was this data aggregated or distributed? What were the assumptions or limitations involved in using this data at the watershed scale? | **Previous Comment:** I also fail to see how you take census data to the watershed and how you align using Nepal government data with your approach to model at the watershed scale. ***Response:*** *We will clarify this in the manuscript. For Nepal, we considered a weighted spatial join between the watersheds and the districts. To each watershed, we attributed the statistics of the district intersecting it, weighted by the overlapping areas.* |
| **2. Methodological Issues:** | |

| | |
|---|---|
| • *Unclear Scope and Flood Types:* The paper does not clearly specify the types of floods being modeled. The introduction mentions multiple types of flooding, but the methodology seems focused on fluvial floods. This inconsistency in the types of flood hazards being assessed undermines the clarity and focus of the study. If the study is limited to fluvial flooding, this should be clearly stated in both the abstract and methods, and the introduction should avoid discussing other types unless they are directly relevant.

 • *Geographical Scope and Upscaling:* The model is trained on data from Nepal but then applied to the entire HMA region. However, the manuscript does not provide sufficient justification for this upscaling. HMA includes diverse climatic regions, ranging from arid areas in Central Asia to monsoon-dominated zones in the southern Himalayas. A model that works well in Nepal may not generalize to other parts of HMA without further validation. It would be more scientifically sound to either focus solely on Nepal or provide validation for other regions in HMA. | **Previous Comment:** At multiple points of the manuscript I was a bit confused on the scope. There is an introduction on all types of high flow events but the methods suggest you only look at fluvial floods with exceptionally high impacts. There is a relatively rapid investigation of the methods for watersheds that do lie to some part in Nepal compared against data only from areas within Nepal and then an upscaling to all of HMA, which in turn is not clearly defined in its scope or climatologies. I would strongly suggest to maybe limit the study to areas where data is available before scaling it up, allowing you more space for methodological and data based issues. ***Response:*** *We thank the reviewer for this suggestion. In general, this paper focuses on fluvial and pluvial flooding, and we will make this clearer in the introduction. Starting from this, for this work, we considered Nepal as our train site for two main reasons.*
 1. *We had information at fine resolution regarding the flood events, in terms of the number of people, economic impact of the event, date of the event, and population data.*
 2. *For Nepal, at the time of this paper we had access to the high-resolution 8m DEM from the previous NASA HIMAT effort. This DEM also covers other areas of the wider Himat region, but it presents some gaps. Nepal was completely covered, and we verified the homogeneity and quality of the data.*

 *The climatology in HMA is indeed variable. In Nepal, as well, we have regional climate variations largely being a function of elevation. For this work, for the main rainfall driver of the model, we focused on climate concentration. This index was proven to be highly linked to pluvial/fluvial flooding impacts in other regions of the world, including for example Italy (both in mountainous landscapes and floodplains (Sofia et al. 2019), the US (Saki et al. 2023), or China* (Du et al., 2023). *Climate concentration values are mostly related to the temporal variability of the rainfall, not to the total amount or the average yearly and seasonal statistics, and its variability captures well various climates (Monjo et al. 2016). For Nepal, as we showed in the paper, we have a gradient of CI values, and as ML models learn from the data they ingest, we believe the system can work across various regions from the climatic point of view. We will add comments on this in the paper, highlighting the strengths and weaknesses of the approach.* |
| **3. Model Validation and Use of ERA5 Data:** | |

| | |
|---|---|
| • *Validation Challenges:* The manuscript relies on ERA5 precipitation data for flood simulation in HMA, but ERA5 has well-documented limitations in representing precipitation in mountainous regions. Precipitation in these areas is highly variable, and ERA5's coarse spatial resolution may not adequately capture localized rainfall events. The authors should either provide a more thorough discussion of these limitations or supplement ERA5 data with region-specific datasets (e.g., from local meteorological agencies) to improve validation. | **Previous Comment:** While I understand that it would be well beyond the scope of this study to evaluate the suitability of ERA5 data for flood simulations (let alone in a mountain context where precipitation products are of poor quality) but it would be crucial to address this and dispel concerns from the get go by referring to discussions of this data in mountain regions as well as for flood mapping.

*Response: This study is a part of the HiMAT project. There are a number of research groups working on different aspects of HMA. At the time we conducted this study, a subgroup of our team was working with ERA5. We wanted to utilize the available dataset and complement the existing study. We will add few comments on this, with highlights on HMA from other related works from the HiMAT team, such as* (Maggioni & Massari, 2018; Maina et al., 2023) |
| • *Model Validation with Flood Data:* The manuscript lacks sufficient validation of the ML model's predictive power. The use of the Dartmouth Flood Observatory (DFO) database, which focuses primarily on lowland floods, may not be the best choice for validating a model intended to predict flood impacts in mountainous regions. Additionally, the number of flood events recorded in the DFO database for Nepal is limited, making it questionable whether this provides robust validation for the entire HMA region. | **Previous Comment:** Apart from the Brakenridge citation not having a date nor being present anywhere in the references, and agreeing that in principle such a dataset would be an interesting set for validation, the fact that the whole dataset only has 46 events from Nepal since 2021 and <10 with the 1000 deaths plus displaced criterium you introduce below makes its use questionable considering this is the area you run your model in. Wouldn't data from Nepal (like https://bipadportal.gov.np/) be much more appropriate then?

*Response: We have done our study for both Nepal and HMA from 1980-2020. To our best understanding, the dataset from emdat and DFO have the longest and most detailed series of point datasets for different events for the time period we are interested in.  We appreciate the separate data source that you shared with us. Please note that we trained our model considering information for NEPAL from http://www.drrportal.gov.np/ which includes flood/heavy rain/flash flood events for all districts in Nepal from different sources. This database includes more than 46 events reported in the DFO. At the scale of HMA, there is no other available dataset reporting flood impacts, aside from DFO and EMDAT, to our knowledge, hence we considered these two, with their limits, to highlight how our model could help target priority areas where indeed events have happened, of a large impact, as highlighted by actual floods reported in these two independent datasets.*

**Previous Comment:** Also this database captures lowland floods, rather than mountain floods, making me wonder whether the aim to characterize 'High Mountain Asia' floods is really the right scope here. Also the DFO reports single coordinates, are you then simply assuming the watershed that matches the coordinate is the only one |

affected? Likely the reported numbers refer to much larger areas, as the size of the watershed you chose is rather small (guessing from the Figure, it's not actually described anywhere!)

*Response: The dataset is not only capturing lowland floods but also mountainous river floods that are characterized as fluvial floods. Also, the damage dataset can only be "point" data at a particular location. There may be one, many, or no point for the whole watershed. As we have described previously, we have used GIS techniques to distribute the damages for the watersheds. This is a common technique that is used widely.*
*We will add information on the size of the watershed. (e.g., range of the watersheds' area)*

| | |
|---|---|
| **4. Presentation and Writing Quality:** | |
| • *Grammar, Formatting, and Citations:* The manuscript contains numerous grammatical errors, incomplete references, and formatting issues. Several citations are listed as "n.d." or are missing from the reference list entirely. These issues, while minor in isolation, collectively reduce the professionalism of the manuscript and suggest a lack of attention to detail. | **Previous Comment:** *In numerous instances references are reported as 'n.d.' where they actually have a date and some are completely missing from the reference list.* **Response:** We will make sure to do a thorough check of the manuscript and correct these mistakes. |
| • *Figures and Captions:* The figures in the manuscript, particularly Figure 9, are inadequately explained. Captions are vague and do not provide sufficient information for readers to interpret the figures independently of the text. | **Previous Comment:** *Figure 9: Panel a is elevation not rainfall as your legend suggests!* **Response:** There was a mistake in the legend... We will correct this. |
| **5. Lack of Novelty in Results:** | |
| • *Expected Findings:* Much of what is presented in the results seems to reflect known trends rather than novel insights. For example, the model's prediction that flood impacts are higher in areas with higher population density is unsurprising and does not provide new knowledge. The authors should focus on demonstrating how their ML model offers unique predictive power or novel findings that go beyond confirming expected patterns.
• *Limited Discussion of Results' Implications:* The discussion of the results lacks depth, particularly in terms | **Previous Comment:** A main concern I have here is that I am still not very clear on where the observed events come from yu compare this to. I am also wondering if your Figure 8 simply only confirms one thing – that there are many people (an input to your model) where there are many people (a validation of your model). How does your model compare on actually coming up with an observed flood from the input 'ERA5 rain'? This concern then propagates into the result for the whole region, whereyou 'predict'the biggestimpacts with the highest population densities. That isn't quite so surprising and it is unclear to me how I can see the power of ML in these results. To be provocative, would the results have been different if you would have just distributed rainfall across the watersheds without a model in between? *Response: We thank the reviewer for his comment. We revised the manuscript thoroughly and we believe it is now clearer. Please note that we trained the model and validated it only using the* |

| | |
|---|---|
| of practical applications. How can the findings be used for real-world flood mitigation or policymaking? What specific new insights into flood risk do these results offer that were not already known? | *data for Nepal, at the district scale and then at the watershed scale. Overall, we opted for a 90-10 approach, for which 90% of the Nepal data were used for training and 10% for validation. Line 360 and following: Comparing predicted Lifeyears Index (LYI) flood impacts with observed data showed good correspondence between high-risk areas identified by the ML method and historical flood locations in Nepal. This suggests that the proposed approach effectively delineates flood risk on a national scale. Figure 8 illustrates this comparison, showcasing observed (empirically evaluated) and ML-predicted LYI values at both watershed (upper row) and district (lower row) levels. The 'observed' LYI values were empirically calculated from observational data (Table 1) and categorized into three groups: 'low', 'medium', or 'high', with basins/districts labeled as 'high' for LYI values exceeding 1000 years, 'medium' between 100 and 1000 years, and 'low' below 10 years. The 'predicted' values represent the outputs from the machine learning model. In Nepal, we achieved an overall training accuracy of 97% and a test accuracy of 63%. Notably, training the model at the watershed level yielded higher accuracy compared to the district level. This is attributed to watersheds being hydrologic units that integrate geomorphological and climatic properties, thus providing a more accurate representation of flood dynamics compared to administrative district boundaries. At the watershed level, nearly all year ranges exhibited a 100% match with observed impacts. In instances where the model's accuracy fell below 100% (e.g., 1985–90 and 1990–95), the LYI values in the affected watersheds were low, indicating that the predictors considered were more indicative of major flooding events. The superior accuracy achieved at the watershed level underscores the value of implementing the model at this scale when scaling up the system.* |